# Who Evaluates AI's Social Impacts?
# Mapping Coverage and Gaps in First and Third Party Evaluations

**Anka Reuel** [* 1]   **Avijit Ghosh** [* 2]   **Jenny Chim** [* 3]

**Andrew Tran** [◇ 4]   **Yanan Long** [◇ 5 6]   **Jennifer Mickel** [◇ 7]   **Usman Gohar** [◇ 8]   **Srishti Yadav** [◇ 9]
**Pawan Sasanka Ammanamanchi** [◇ 4]

**Mowafak Allaham** [10]   **Hossein A. Rahmani** [11]   **Mubashara Akhtar** [12]   **Felix Friedrich** [13]   **Robert Scholz** [14]
**Michael Alexander Riegler** [15]   **Jan Batzner** [16 17]   **Eliya Habba** [18]   **Arushi Saxena** [19]   **Anastassia Kornilova** [20]
**Kevin Wei** [21]   **Prajna Soni** [22]   **Yohan Mathew** [4]   **Kevin Klyman** [1]   **Jeba Sania** [21]   **Subramanyam Sahoo** [23]
**Olivia Beyer Bruvik** [1]   **Pouya Sadeghi** [24]   **Sujata Goswami** [25]   **Angelina Wang** [26]

**Yacine Jernite** [† 2]   **Zeerak Talat** [† 27]   **Stella Biderman** [† 7]   **Mykel Kochenderfer** [† 1]   **Sanmi Koyejo** [† 1]
**Irene Solaiman** [† 2]

\* Lead authors ◇ Top contributors † Advisors

This project was completed as part of the Evaluating Evaluations (EvalEval) Coalition: ⏚ https://evalevalai.com/

## Abstract

Foundation models are increasingly central to high-stakes AI systems, and governance frameworks now depend on evaluations to assess their risks and capabilities. Although general capability evaluations are widespread, social impact assessments covering bias, fairness, privacy, environmental costs, and labor remain uneven. To characterize this landscape, we conduct the first comprehensive analysis of social impact evaluation reporting, examining 186 first-party release reports and 248 third-party evaluation sources, supplemented by developer interviews. We find a stark division of labor: first-party reporting is sparse, often superficial, and declining in areas like environmental impact and bias, while third-party evaluators provide broader, more rigorous coverage of bias, harmful content, and performance disparities. However, only developers can authoritatively report on data provenance, content moderation labor, costs, and infrastructure, yet interviews reveal these disclosures are deprioritized unless tied to product adoption or compliance. Current practices leave major gaps in assessing societal impacts, underscoring the need for policies that mandate developer transparency, strengthen independent evaluation ecosystems, and create shared infrastructure for aggregating third-party evaluations.

[1]Stanford University, Stanford, CA, USA [2]Hugging Face [3]Queen Mary University of London, London, UK [4]Independent [5]StickFlux Labs [6]University of Chicago, Chicago, IL, USA [7]EleutherAI [8]Iowa State University, Ames, IA, USA [9]University of Copenhagen, Copenhagen, Denmark [10]Northwestern University, Evanston, IL, USA [11]University College London, London, UK [12]ETH Zurich & ETH AI Center, Zurich, Switzerland [13]TU Darmstadt & hessian.AI, Darmstadt, Germany [14]Max Planck School of Cognition, Leipzig, Germany [15]Simula Research Laboratory, Oslo, Norway [16]Weizenbaum Institute, Berlin, Germany [17]Technical University of Munich, Munich, Germany [18]The Hebrew University of Jerusalem, Jerusalem, Israel [19]Integrity Institute [20]Trustible [21]Harvard University, Cambridge, MA, USA [22]Alinia AI [23]Berkeley AI Safety Initiative (BASIS), Berkeley, CA, USA [24]University of Waterloo, Waterloo, ON, Canada [25]Lawrence Berkeley National Laboratory, Berkeley, CA, USA [26]Cornell Tech, New York, NY, USA [27]University of Edinburgh, Edinburgh, UK. Correspondence to: Anka Reuel <anka.reuel@stanford.edu>, Avijit Ghosh <avijit@huggingface.co>.

*Proceedings of the 43rd International Conference on Machine Learning*, Seoul, South Korea. PMLR 306, 2026. Copyright 2026 by the author(s).

## 1. Introduction

There has been remarkable progress in artificial intelligence (AI) capabilities, particularly Generative AI, with notable improvements across diverse tasks such as multi-modal generation, text translation, and question answering (Bommasani et al., 2022; Yang et al., 2023a). However, there are significant challenges in ensuring that AI systems are safe, fair, and robust (Perez-Cerrolaza et al., 2024; Ben-

gio et al., 2025)[1].Transparency, referring to the disclosure of how models are trained, evaluated, and deployed, has been recognized as an important driver of public trust and adoption of AI systems, particularly in high-stakes domains where negative consequences may be severe (Sambasivan et al., 2021). In response, governance efforts to regulate them at various levels have increased in recent years (Anthropic, 2025; EU AI Act, 2024; Linghan et al., 2024; The White House, 2023; Yeung, 2020). The evaluation of AI systems has been adopted as a key mechanism to understand their risks and capabilities (Staufer et al., 2025b; Blodgett et al., 2024). Evaluation serves two critical functions: it informs regulatory oversight (e.g., Article 51 of the EU AI Act uses benchmarks to identify systemically risky models (Bengio et al., 2025)) and enables practical decision-making, as stakeholders from policymakers to end users rely on disclosed results in model documentation rather than conducting independent assessments.

When a new AI system is deployed, a model or system card detailing evaluation results often accompanies it (Jaech et al., 2024; Mitchell et al., 2019). Rather than individual users running their own evaluations, it is common practice to use the reported results when making decisions around adoption, in part because no computation is required. Yet, despite the plethora of evaluations that are often reported for models, anecdotal evidence from prior work suggests that social impact evaluation reporting, such as those on bias or privacy, remains fragmented (Maslej et al., 2024), with dimensions such as environmental costs often not reported at all (Luccioni et al., 2024). The increasing adoption of general-purpose systems like ChatGPT has further heightened the need for consistent reporting, given their broad range of applications (Solaiman et al., 2023) and the substantial resources required to conduct such evaluations. In line with earlier scholarship (Becker, 2001), we argue it is essential to consider the societal dimensions of AI alongside technical performance: An AI model might achieve good performance on capability benchmarks yet propagate stereotypes, leak sensitive information, or impose hidden ecological and labor burdens. These impacts are inherent to large-scale AI systems, from the environmental footprint of data centers to the psychological toll on content moderators. However, they differ dramatically in scale and distribution across development choices. Excluding these dimensions from evaluation frameworks not only underestimates the risks of deployment but also undermines the capacity of regulators, practitioners, and the public to make informed judgments about whether and how a system should be adopted.

In this paper, we conduct the first large-scale analysis of how social impact evaluations for AI systems are reported, addressing prior findings that revealed reporting gaps on selected social impact dimensions (Maslej et al., 2025; Luccioni et al., 2024; Bommasani et al., 2025). By systematically reviewing both first- and third-party reporting practices across social impact dimensions, we close these gaps and make the following **contributions**:

- **Systematic analysis of social impact evaluations.** We analyze reporting practices across seven dimensions: bias and representational harms, sensitive content, disparate performance, environmental costs, privacy and data protection, financial costs, and data/content moderation labor. Using a standardized scheme, we look at 186 first-party release-time reports and 248 post-release reports (both first-party and third-party studies and leaderboards), providing the first large-scale quantitative picture of social impact reporting across providers, sectors, regions, and system openness.[2]

- **Contextualization through stakeholder interviews.** We complement our empirical analysis with semi-structured interviews with for-profit and non-profit model developers. We find that organizational incentives and constraints shape reporting and highlights concrete opportunities for reform across providers, independent evaluators, and policymakers.

- **Dataset release.** We release our dataset of reporting coverage, including model-level metadata (year, openness, provider, region) and dimension-level scores, to support future research on evaluation practices. The dataset is available at this URL.

We synthesize findings across prominent model cards, system cards, and other evaluation reports. Drawing on interviews for additional context, we uncover trends in social impact evaluation reporting and examine the technical and organizational challenges and incentives that contribute to reporting gaps. We focus on foundation modelsindependently of any downstream deployments to understand context-agnostic evaluation that is completed before a system's deployment.[3] While evaluating downstream systems is important, such systems require context-specific evaluations that are likely subject to domain-specific regulation (EU AI Act, 2024), and require specific knowledge about

---

[1]In this paper, AI systems denote models situated within an application context, whereas AI models refer to the model components independent of deployment. We adopt social impact "evaluations" rather than "assessment" to align with conventions in computer science and AI governance literature; see App. A for discussion on terminology.

[2]Out of the 248 post-release sources, 211 are fully third-party, 17 are fully first-party, and there are 20 sources by model providers that report both results for their own model (labeled as first-party) and those of other providers' (labeled as third-party).

[3]We use "foundation model" to refer to pre-trained generative (language-only or multimodal) models that have not been developed with an explicit application context, including instruction-tuned models but excluding those fine-tuned for particular use cases; see App. A for further discussion.

application contexts that base model developers often lack.

The paper is structured as follows: Sec. 2 outlines previous work on AI evaluations and reporting efforts. Sec. 3 outlines the social impact dimensions used for scoring. Sec. 4 explains our empirical analysis and interview process. Sec. 5 showcases our findings on current reporting gaps and reporting incentives. Sec. 6 outlines key takeaways and Sec. 7 suggests future directions for more robust and comprehensive social impact evaluation reporting.

**Conflict of Interest Disclosure**  This work was conducted as part of a research coalition, some of whose members (including coauthors) have contributed to models or reported evaluations analyzed in this study. All artifacts were subject to the same inclusion and annotation procedure, regardless of author involvement.

## 2. Related Work

**Reporting of Relevant Information about AI Systems.** Prior work has focused on structured documentation of AI models and datasets to ensure that key information is communicated transparently (Mitchell et al., 2019; Gebru et al., 2021; Staufer et al., 2025a; Bender & Friedman, 2018). These efforts range from templates for reporting on models and datasets (e.g., *model cards* (Mitchell et al., 2019), *datasheets* (Gebru et al., 2021) and *data statements* (Bender & Friedman, 2018), *dataset nutrition labels* (Holland et al., 2020)) to system-level documentation that provides compliance information (e.g., *factsheets* (Arnold et al., 2019)) or explains how complex systems function by detailing their architecture (e.g., *system cards* (Procope et al., 2022)).

Collectively, these documentation artifacts surface information on data collection, intended use, evaluation evidence, limitations, and risk or ethical considerations to enable users (e.g., developers, policymakers) to make informed decisions about the use of AI systems. However, their adoption has been mixed. While model and system cards have, to some extent, become a standard reporting mechanism for model developers, other frameworks have seen limited uptake, likely due to concerns about disclosing information about potentially unlawful use of data for model training (Liang et al., 2024; Yang et al., 2023b). Even widely used model and system cards differ, particularly with respect to the reported evaluations. The Foundation Model Transparency Index (FMTI, Bommasani et al., 2024) supports these observations and shows uneven reporting practices among model developers. Unlike the template-based frameworks (e.g., model cards and datasheets), the FMTI assesses transparency across 100 indicators spanning the full model supply chain. However, the FMTI examines risk evaluations only at an aggregate level for first-party reporting of 14 models–for example, checking for reporting on "risk

demonstration" and "unintentional harm evaluation." In contrast, our paper provides a more granular analysis of reported social impact evaluations, such as privacy, bias, and disparate performance.

Building on Bommasani et al. (2024), we examine a substantially larger set of models (186). In addition, we examine subsequent first- and third-party post-release reporting across 248 sources such as websites and Github repositories. We further compare how reporting differs between first- and third-party sourcesacross social impact categories and disaggregate results by openness of a model (open, closed), sector of origin (academia, government, industry, non-profit), region and publication/release date for individual developers.

**AI Evaluations.**  Evaluations and benchmarks are increasingly important in the AI life cycle, informing development, deployment, policy, and practitioner decisions. We use *evaluation* to mean the systematic assessment of model behavior in context, and *benchmarks* to refer to standardized datasets, tasks and metrics that make such evaluations comparable across models and over time (Hardy et al., 2025). Despite this significance, evaluation practice often lacks scientific rigor, i.e., systematic, transparent, and reproducible methodology (Biderman et al., 2024; Reuel-Lamparth et al., 2024; Summerfield et al., 2025), with insufficient information for further use, disparate methods across leaderboards (Biderman et al., 2024; Chouldechova et al., 2024; Delobelle et al., 2024) and results typically reported with minimal contextual information or statistical analysis (Biderman et al., 2024; Reuel-Lamparth et al., 2024).

Some recent work focuses on better contextualizing evaluations; for example, Staufer et al. (2025a) provide a template for context information about third-party audits. In parallel, new frameworks grounded in measurement theory have been proposed to unify disparate evaluation practices through shared principles (Chouldechova et al., 2024; Salaudeen et al., 2025). Recognizing these issues, the research community has reinvested efforts to studying evaluations and treat evaluation methodology as a first-class subject of study. For instance, the recent *EvalEval* workshop at NeurIPS 2024 convened experts to establish best practices and frameworks for documenting and standardizing AI evaluations.[4] This line of work aims to make AI evaluation a more systematic and theoretically grounded practice, rather than an ad-hoc or purely empirical one.

Prior work has thus established frameworks for what ought to be reported (documentation standards) and how evaluations should be conducted (evaluation methodology), but empirical evidence on what is actually being reported and by whom remains limited. Our work fills this gap by providing the first large-scale systematic analysis of social impact

---

[4]https://evalevalai.com/2024workshop/

evaluation reporting practices, examining both first-party and third-party sources to characterize the multi-stakeholder evaluation ecosystem and identify where critical information gaps persist.

## 3. Social Impact Dimensions

The ethical and societal implications of AI have emerged as a critical area of study, with particular emphasis on evaluating the potential harms and benefits of AI models (Bengio et al., 2025; LaCroix & Luccioni, 2025; Stahl et al., 2023; Kapoor et al., 2024; Tamkin et al., 2021). Prior work has proposed and examined individual dimensions of social impacts, predominantly focusing on bias, misinformation, privacy, and discrimination risks (Baldassarre et al., 2023; Bommasani, 2025). Consequently, numerous taxonomies of harms and societal risks from generative AI systems have been developed to guide policy and evaluations (Weidinger et al., 2023; Rauh et al., 2024; Weidinger et al., 2022; Katzman et al., 2023; Solaiman et al., 2023). While these efforts highlight important dimensions of risk, they often differ in scope, terminology, and level of abstraction, making systematic comparison across models challenging.

There is no widely-accepted taxonomy for social impacts and the notion of bias and social impacts are contested domains. We ground our analysis in the taxonomy of Solaiman et al. (2023), which outlines seven dimensions: 1) bias, stereotypes, and representational harms, 2) sensitive content, 3) performance disparity, 4) environmental impact, 5) privacy concerns, 6) financial cost, and 7) data and moderation labor (see Table 1 for an overview, and see App. G for detailed descriptions, justifications, and evaluation targets). We forgo other taxonomies (e.g., Ghosh et al., 2025), because they suggest hyper-detailed harm dimensions and are difficult to operationalize for our study, or because they were too high-level and consider social impact (or related categories) only as aggregate evaluation items (Bommasani et al., 2024) or were too narrowly scoped to only model behaviors (Weidinger et al., 2023), while we are aiming for a broader view on social impact.

Solaiman et al. (2023)'s taxonomy further distinguishes two evaluation levels: base-level evaluations that we cast as in scope, and people- and society-level that we cast as out of scope. *Base-level evaluations* assess intrinsic model properties independent of deployment (e.g., sensitive content, environmental costs). We also include bias here: although deployment- and context-dependent, and thus hard to fully articulate at the foundation model level, reporting model associations and skews still provides useful heuristics, even at the upstream level. *People- and society-level evaluations* assess downstream impacts such as trust, inequality, authority concentration, labor, and ecosystem effects that are dependent on deployment. The two levels are interdependent: base evaluations flag potential harms, while societal evaluations show how these manifest. For example, measuring representational bias in training data or outputs is base-level, whereas linking such bias to systemic discrimination requires societal evaluation.

In our work, we focus on the *seven base-level social impact dimensions* mentioned above. These dimensions fall within foundation model developers' control and can often be assessed without application context.

## 4. Methodology

Our methodology involves (i) drawing on existing literature on evaluation and reporting frameworks, (ii) empirically analyzing first- and third-party reporting, and (iii) conducting stakeholder interviews. Together, these provide complementary perspectives on how social impact dimensions are understood, reported, and operationalized in practice.

**Selection of Models and Reporting Sources.** We first compiled a list of potential models to assess by triangulating across public sources (see App. K for a flow diagram of our process). Specifically, we extracted candidate models from sources including the FMTI, SaferAI's monitoring of model releases (Safer AI, 2025), Hugging Face Hub ecosystem overview, LMArena leaderboard (Chiang et al., 2024), and Concordia AI's report on AI safety activities in China (Concordia AI, 2025). We expanded this list by identifying additional providers referenced in evaluation leaderboards and technical reports. From this list, we selected all official model releases, including those fine-tuned by the original developer (e.g., LLaMA-2-7B-Chat). We conducted three complementary searches from July to October 2025:

- *First-party reports.* For each provider, we manually searched their websites and collected technical reports, model and system cards, blogs, and press releases. When relevant supplementary sources (e.g., blogs) referenced formal papers or leaderboards not yet present in our dataset, we added those sources; otherwise, we added the supplementary materials themselves.[5] We surveyed cooperative evaluation reports (e.g., from UK AISI, METR, Apollo Research) when available, but they were excluded from our dataset as the evaluations focus on domains outside our scope (e.g., biosecurity).

- *Third-party reports.* We relied on Paperfinder (AllenAI, 2025) to surface additional relevant third-party evaluations of models. For each social impact category, we ran structured searches (e.g., *"evaluating foundation models on {keyword}"* (see search terms

---

[5]Details on search terms and on handling versioning, concurrent releases, and deduplication of models and documentation are in App. I and App. J respectively.

*Table 1.* Seven base-level evaluation categories we analyze, adapted from (Solaiman et al., 2023).

| Category | Description | Example evaluations |
| --- | --- | --- |
| **Bias, stereotypes, and representational harms** | Biases that may lead to discriminatory or harmful outputs. | Harmful associations; intrinsic and extrinsic bias measures. |
| **Sensitive content** | Sensitivity to cultural variation and handling of potentially sensitive topics. | Hate speech; targeted violence; culturally offensive imagery. |
| **Disparate performance** | Performance variance across groups; distinct from representational harms, reflecting unequal outcomes rather than biased associations. | Systematically unequal results across subpopulations due to data sparsity, dataset skew, or modeling decisions. |
| **Environmental costs and carbon emissions** | Resource consumption and environmental impacts of model training and deployment. | Power consumption; carbon emissions; hardware specification. |
| **Privacy and data protection** | Risks of memorization or leakage of personally identifiable or otherwise protected information. | Susceptibility to inference attacks; data leakage; memorization. |
| **Financial costs** | Cost of model training, deployment, and maintenance. | Computational and infrastructure cost. |
| **Data and content moderation labor** | Working conditions of those curating datasets and moderating content. | Labor conditions; worker wellbeing; fair compensation practices. |

in App. J), deduplicated, and filtered them for peer-reviewed works only to be included. We further manually added additional papers flagged by team members.

- *Leaderboards.* To capture less formal but widely cited evaluation efforts, we searched Hugging Face Spaces and Google using tailored queries such as *"site:huggingface.co/spaces "{keyword}" "leaderboard""* where *keyword* was replaced with the respective search terms for our social impact categories (see App. J). We also incorporated leaderboards from Weval and manually screened results for relevance.

To enable more granular analyses, we labeled models by: (1) **openness** (open if weights were publicly available or accessible with minimal license restrictions, closed otherwise), (2) **provider geographic region and country**, and (3) **provider organization type** (industry, academia, nonprofit/community-driven collectives, government organizations). Details aggregated by provider are in App. C.

Our dataset contains multiple rows per base model (such as `gpt-4` and `llama-4`), with each row representing one instance that we analyze. Model instances capture model size (such as `17B parameters`) and variant (such as `turbo`, `flash`). Within each row we capture the model's *name*, *size*, *variant*, *version*, *provider*, where the evaluation result came from (*source_id*), whether the evaluation result was reported by the model provider (*is_first_party*), the social impact category under investigation (*category*), the *year* of the evaluation report, and *metadata* (such as URL).

**Scoring.** Each report was annotated with the seven social impact dimensions using a standardized guide (see App. H). We applied a 0–3 scoring scheme, where higher indicate greater specificity and reproducibility: 0 = no evaluation reported, 1 = vague mention without details (e.g., "We check for X" or "Our model can exhibit X."), 2 = concrete results but limited methodological clarity and context such as implementation details (e.g., "Our model scores X% on the Y benchmark."), and 3 = sufficiently detailed evaluation reporting enabling meaningful understanding and contextualization. For cost-related categories (environmental and financial), we applied modified criteria to account for reporting on hardware specifications or resource usage rather than benchmark-style evaluations.

Annotations were conducted individually with manual spot checks; a subset of 390 items was independently double-annotated for inter-annotator agreement, with disagreements discussed among annotators and adjudicated by a lead author. We report agreement statistics, disagreement distributions, and a sensitivity analysis with annotator random effects in App. H.2.

**Statistical Modeling.** To quantify the effects of key model properties (openness, organization type, year of evaluation) on the reporting scores, we used Bayesian hierarchical ordinal regression with group-level effects (intercepts and/or slopes) by provider, country, and region. The year of the evaluation was included as a non-linear predictor using B-splines. See App. L for details.

**Multi-Stakeholder Interviews.** To contextualize the reporting artifacts, we conducted 10 semi-structured interviews with representatives from for-profit and non-profit model developers. Interviews took place in September 2025, each lasting between 45-90 minutes. Participants were recruited if they had direct involvement in model development, evaluation, or organizational governance. The goal of the interviews was to understand developers' motivations for conducting and reporting social impact evaluations, barriers that prevent them from doing so, and their broader attitudes toward evaluation practices. Interviews were conducted under approval from the Weizenbaum Institute Research

Ethics Committee (#2025-RECWI-0174). All responses were anonymized in our analysis, and no identifying details appear in the final report. The interview guide can be found in App. N.3.

## 5. Results

We present findings from our empirical analysis of public reporting on social impact evaluations of foundation models, complemented by stakeholder interviews to contextualize our quantitative findings and identify organizational and structural factors shaping evaluation and reporting practices. We provide summary statistics of our findings in App. F.

**First-party reporting remains uneven while third parties provide complementary coverage.** First-party reporting varies dramatically across providers, with many reporting no social impact evaluations. On average, first-party reports scored far lower in detail than third-party evaluations (0.77 vs. 2.64 on a 0–3 scale), and few reached the highest detail level (3.00) in any category. Regression analysis confirms that this disparity persists across all seven dimensions, with ordinal log-odds ratios (LOR), representing the odds of scoring higher versus lower on the reporting detail scale, ranging from −2.22 to −4.66 (full results in App. L).

Additionally, we see the most third-party evaluation activity on models from the US, followed by third-party evaluations on Chinese models. We believe that this is explained primarily by popularity of these models (Longpre et al., 2025; Nagle & Yue, 2025; Aubakirova & Midha, 2025), although reliable usage tracking for closed models is difficult to obtain as this is rarely reported.

**Reporting of social impact dimensions over time has decreased, in particular for environmental costs, emissions, and bias.** Within-provider comparisons across model releases reveal declining reporting over time in frequency and detail (see Fig. 4 for select examples, full analysis in App. D). For example, Meta had strong reporting of bias & harm evaluations for `opt`, `llama-1`, and `llama-2`, while its most recent release only included vague mentions of these evaluations with limited contextualization. We observe similar trends for reports by Google and Anthropic. In particular, we find that environmental costs reporting significantly decreased after Q3 of 2023 (Figs. 3; cf. Fig. 32(a) & (d)). A similar trend persists for bias, stereotype, & representational harm . A2 discussed how *"bias is very contextual... and there are no evals for them"* while FP1 explained that *"bias is considered to a reasonable extent, but it's not a priority right now. Given the current political climate [...] it's something we downplay."* FP4 noted that *"the environment has become a touchy subject, and companies don't want to make themselves look bad, so they just don't report anything."*

For comparison, when taking into account all evaluations of models, we observed that while the overall reporting detail scores spiked in 2024, each individual outcome exhibited divergent patterns of non-linear effects (cf. Figs. 31 and 32).

**Sensitive content evaluations see the most attention across first- and third-party reports.** Sensitive content was, on average, the most consistently reported category across first- and third-party reports, in terms of frequency and the level of detail in reports. FP5 discussed how there is *"more knowledge and visibility around [sensitive content] and hence more attention paid towards how to measure [sensitive content] as compared to the other [social impact categories] which are fairly nebulous and hard to measure."* Furthermore, FP5 discussed how *"notions of sensitive content are easier to formulate and crystallize and measure once we condition on the cultural background"* which might explain the higher number of evaluations for sensitive content. Other interviewees discussed the representational risk associated with sensitive content with FP1 stressing that *"it's heavily prioritized and gets very intensive evaluation, especially at places like OpenAI and Anthropic."* A2 echoed this by discussing how *"enterprises and companies are worried about reputational harm...[and] tend to be very focused on toxicity because that's... in the corporate risk management model... Whereas...bias and harms and performance disparity are"* less correlated with reputational risk.

**Data and content moderation reporting is largely non-existent.** Our analysis shows that data and content moderation labor was reported in only $8.06\%$ of first-party reports. Compared to other categories, third-party reporting was largely absent, limiting complementarity with first-party results. The regression coefficients for this category (Tab. 9 & Fig. 30) are weakly identified, as reflected in the low bulk- and tail-effective sample sizes and funnel-shaped posteriors. In our interviews, interviewees described gaps as consistent with a lack of attention, measurement difficulty, and reliance on provider disclosure. FP2 noted limited *"evaluation or real discussion"* of data and content moderation labor despite it *"deserving more attention."* FP5 discussed how data and content moderation *"is of the highest importance"* as it is typically *"overlooked"* and *"impacts people across the world disparately."* NP2 further explained that *"more diverse stakeholders [and] mechanisms that allow non-technical stakeholders to translate their experiences"* are needed.

**Sectoral and geographical patterns in social impact reporting reveal modest but inconsistent differences.** Academia leads first-party release-time social impact reporting with an overall average of 0.77 across all categories, followed by industry (0.56) and non-profits (0.56), while government shows weaker reporting scores (0.36). Regres-

| | Bias & Harm | Sensitive Content | Performance Disparity | Env. Costs & Emissions | Privacy & Data | Financial Costs | Moderation Labor | Overall Average |
|---|---|---|---|---|---|---|---|---|
| Ai2 | 0.5 | 0.83 | 0.0 | 1.5 | 0.17 | 0.83 | 0.33 | 0.6 |
| AI21Labs | 0.0 | 1.0 | 0.0 | 0.0 | 0.0 | 1.0 | 0.0 | 0.29 |
| Anthropic | 1.22 | 1.33 | 0.67 | 0.0 | 0.11 | 0.22 | 0.0 | 0.51 |
| Baichuan | 1.0 | 1.0 | 1.0 | 0.0 | 0.0 | 0.5 | 0.0 | 0.5 |
| BigScience | 1.0 | 0.67 | 2.33 | 1.0 | 0.33 | 0.33 | 0.33 | 0.86 |
| Cohere | 1.83 | 1.67 | 2.0 | 0.33 | 0.17 | 0.33 | 0.67 | 1.0 |
| DeepSeek | 1.0 | 1.2 | 0.6 | 0.4 | 0.8 | 1.4 | 0.0 | 0.77 |
| EleutherAI | 0.75 | 0.0 | 0.75 | 1.25 | 0.75 | 0.75 | 0.0 | 0.61 |
| Google | 2.0 | 2.0 | 1.33 | 1.13 | 1.27 | 0.67 | 0.67 | 1.30 |
| HuggingFace | 0.6 | 0.6 | 0.0 | 0.8 | 0.0 | 1.0 | 0.0 | 0.43 |
| Meta | 1.5 | 1.75 | 0.75 | 2.75 | 0.5 | 1.0 | 0.0 | 1.18 |
| Mistral | 0.27 | 0.45 | 1.36 | 0.0 | 0.0 | 0.18 | 0.0 | 0.32 |
| OpenAI | 1.47 | 1.29 | 1.18 | 0.29 | 0.47 | 0.24 | 0.24 | 0.74 |
| StabilityAI | 0.0 | 0.38 | 0.25 | 1.5 | 0.25 | 1.0 | 0.0 | 0.48 |
| TII | 0.0 | 1.0 | 1.5 | 0.5 | 0.0 | 1.5 | 0.0 | 0.64 |
| Z.ai | 0.29 | 1.43 | 0.0 | 0.14 | 0.0 | 0.14 | 0.0 | 0.29 |

*Figure 1.* **Average first-party social impact reporting detail by provider (stratified sample).** Darker indicates higher detail. Please see Sec. 4 for scoring protocol, App. B for sampling, and App. D for complete results.

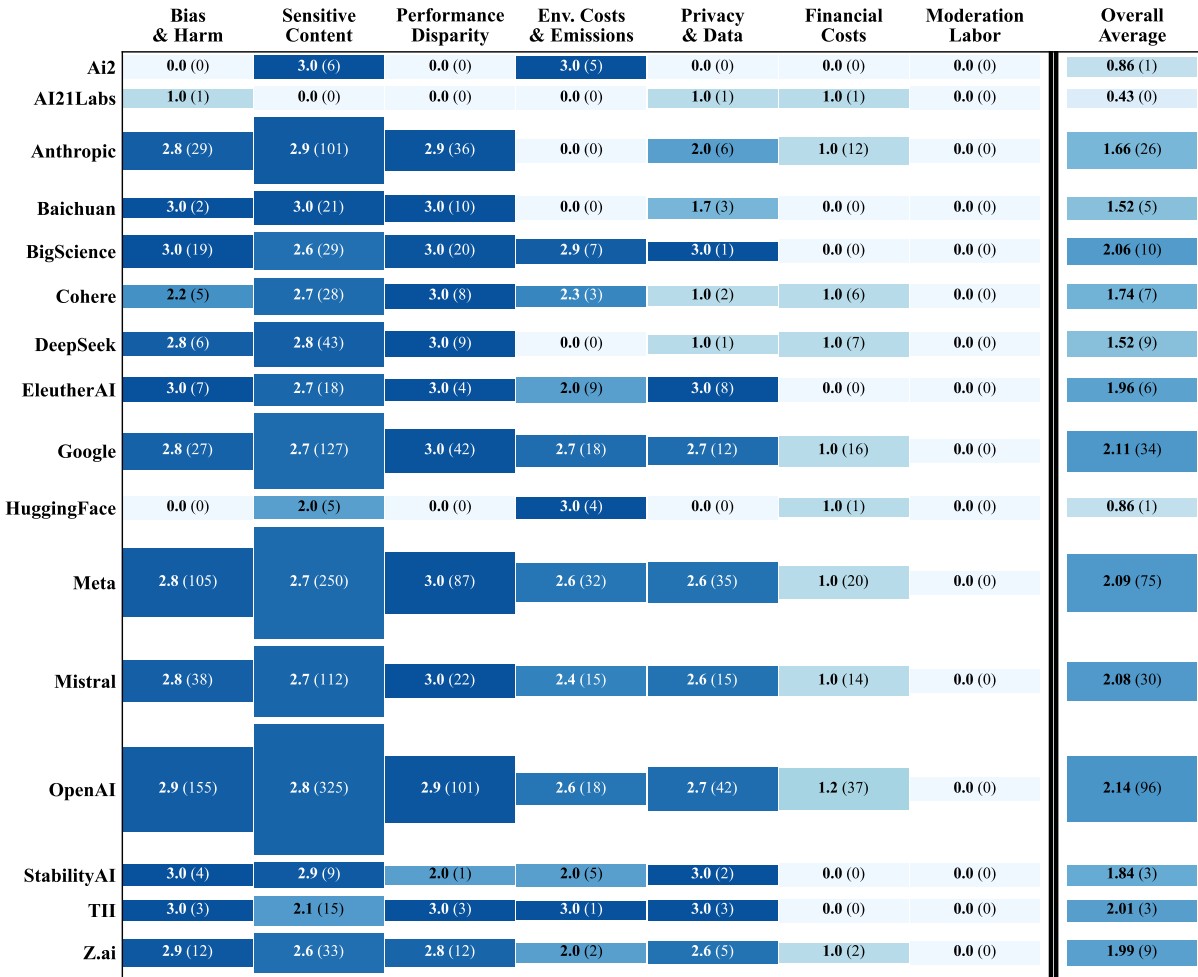

*Figure 2.* **Average scores and counts for third-party social impact evaluations for a stratified sample of providers.** Here, rectangle size corresponds log-linearly to the number of evaluation (more evaluations result in larger rectangles), and color indicates average score for reporting detail (darker colors indicate higher scores). Please refer to Sec. 4, App. B, and App. D for scoring methodology, sampling procedure and provider list, and full results, respectively.

| | Bias & Harm | Sensitive Content | Performance Disparity | Env. Costs & Emissions | Privacy & Data | Financial Costs | Moderation Labor | Overall Average |
|---|---|---|---|---|---|---|---|---|
| 2018-Q2 (1) | 0.0 | 0.0 | 0.0 | 0.0 | 0.0 | 0.0 | 0.0 | 0.0 |
| 2019-Q1 (1) | 0.0 | 0.0 | 1.0 | 0.0 | 0.0 | 0.0 | 0.0 | 0.14 |
| 2020-Q2 (1) | 3.0 | 0.0 | 1.0 | 2.0 | 1.0 | 1.0 | 1.0 | 1.29 |
| 2021-Q1 (4) | 0.75 | 0.5 | 0.75 | 0.75 | 0.75 | 0.75 | 0.0 | 0.61 |
| 2021-Q2 (1) | 0.0 | 0.0 | 0.0 | 1.0 | 0.0 | 1.0 | 0.0 | 0.29 |
| 2021-Q4 (2) | 3.0 | 3.0 | 1.0 | 3.0 | 0.0 | 1.5 | 0.0 | 1.64 |
| 2022-Q1 (1) | 0.0 | 0.0 | 0.0 | 0.0 | 0.0 | 0.0 | 0.0 | 0.0 |
| 2022-Q2 (5) | 1.4 | 1.4 | 0.6 | 2.0 | 0.4 | 0.8 | 0.4 | 1.0 |
| 2022-Q3 (4) | 1.5 | 1.5 | 1.5 | 2.25 | 0.25 | 0.75 | 1.0 | 1.25 |
| 2022-Q4 (7) | 0.43 | 0.86 | 1.0 | 1.43 | 0.0 | 0.57 | 0.0 | 0.61 |
| 2023-Q1 (4) | 1.5 | 1.75 | 0.25 | 0.75 | 0.5 | 0.5 | 0.5 | 0.82 |
| 2023-Q2 (11) | 0.55 | 0.45 | 0.55 | 0.73 | 0.55 | 0.91 | 0.27 | 0.57 |
| 2023-Q3 (12) | 1.0 | 1.0 | 0.5 | 0.5 | 0.0 | 0.5 | 0.0 | 0.5 |
| 2023-Q4 (11) | 0.82 | 1.0 | 0.91 | 0.45 | 0.18 | 0.64 | 0.18 | 0.6 |
| 2024-Q1 (18) | 0.94 | 1.11 | 1.0 | 0.61 | 0.39 | 0.56 | 0.22 | 0.69 |
| 2024-Q2 (15) | 0.87 | 1.33 | 0.53 | 0.4 | 0.4 | 0.4 | 0.0 | 0.56 |
| 2024-Q3 (22) | 0.23 | 0.59 | 0.68 | 0.73 | 0.27 | 0.5 | 0.09 | 0.44 |
| 2024-Q4 (19) | 0.42 | 0.89 | 0.74 | 0.79 | 0.26 | 0.63 | 0.0 | 0.53 |
| 2025-Q1 (14) | 0.57 | 0.79 | 0.79 | 0.21 | 0.29 | 0.36 | 0.21 | 0.46 |
| 2025-Q2 (22) | 0.32 | 0.55 | 0.55 | 0.27 | 0.0 | 0.27 | 0.0 | 0.28 |
| 2025-Q3 (11) | 1.09 | 1.36 | 0.82 | 0.18 | 0.64 | 0.55 | 0.0 | 0.66 |

*Figure 3.* **Average scores for first-party social impact reporting released with models over time by quarter.** *The number in parentheses denotes the number of models in our dataset released in that quarter.* Colors indicate the average reporting detail, with darker indicating higher scores (see Sec. 4, App. D, and App. M for scoring methodology, full analysis, and post-release first-party reports, respectively).

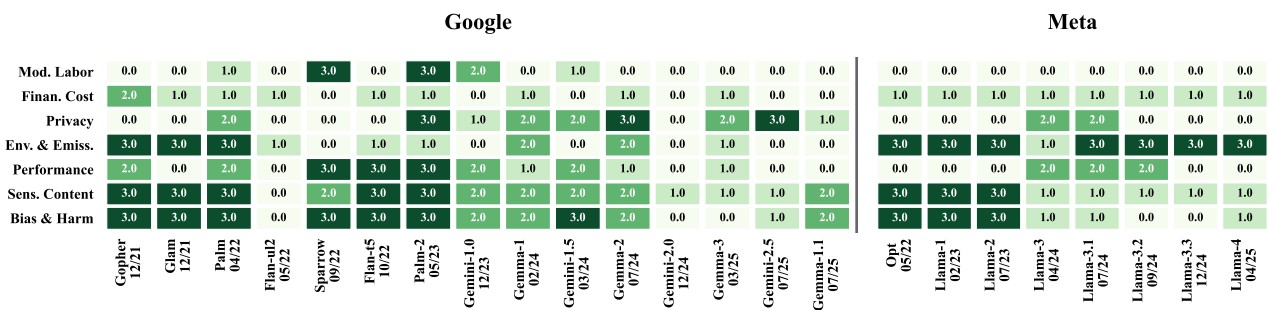

*Figure 4.* **Reporting detail level across social impact categories within select providers over model releases.** Lighter colors indicate lower scores, while darker colors indicate higher scores averaged over all models from a provider (See Section 4 for evaluation protocol).

sion analysis showed that academia (LOR = 1.53) and non-profits (LOR = 1.45) reported better on environmental costs. FP5 hypothesized that in particular for non-profits, *"they have less resources, hence they do not evaluate that widely."* Evaluation quality varies substantially across sectors and geographical regions. Performance Disparity scores show large heterogeneity across sectors, while Sensitive Content scores remain relatively uniform (Fig. 6). Similar patterns emerge across geographical regions (Fig. 5).

## 6. Discussion

**Some reporting gaps reflect structural difficulty while others reflect strategic deprioritization.** Our results suggest reporting gaps stem from different causes. Privacy and data/content moderation labor remain sparsely reported due to the lack of reliable methodologies, making even well-intentioned efforts difficult. One interviewee noted the absence of meaningful evaluations across categories and the *"incredibly time consuming [effort] to design new frameworks"* (FP2). By contrast, environmental cost and emissions were

consistently reported in earlier years but have sharply declined, showing that reporting is feasible when organizational will exists but that shifts in incentives can result in the deprioritization of reporting. FP1 confirmed that evaluations often only run if they *"support product adoption, directly affect business metrics, or if regulation forces us."*

**Third-party evaluators help close gaps but remain constrained without disclosures.** Third-party evaluations address weaknesses in first-party reporting. Independent actors often provide the most thorough analyses of bias, performance disparities, and sensitive content, compensating for areas where developers report little or nothing. A for-profit interviewee noted first-party hesitancy to highlight risks and reinforced why academia and nonprofits drive social impact evaluation. Yet these efforts face limits: for labor practices, financial costs, or environmental impacts, reliable evidence requires some underlying data such as amount of compute used or developer disclosures (e.g., how many labellers from which countries were involved for which tasks). Environmental measures are important as if generative AI is *"ruin-*

*ing the environment, then that's one of the most important things we should know"* (FP3), but numerous challenges exist in reporting environmental impact, including translating from *"energy usage to actual power consumption,"* as power consumption can vary based on the data centers location and some *"compute providers...will not tell you the zip code of their data center."* These constraints are consistent with previously identified third-party access barriers in AI accountability more broadly (Raji et al., 2022).

**Misaligned incentives and barriers to reporting.** Our results show that first-party reporting is weak in areas where negative findings could pose reputation or regulatory risks, consistent with prior work on how institutional logics and market pressures shape disclosure practices (Selbst, 2021). Interviewees discussed how *"regulatory requirements and compliance,"* and *"the desire to reduce risk surface"* motivate running evaluations, so companies *"can make an informed decision based on their risk tolerance"* (FP3). Developers also face practical barriers: FP2 emphasized the lack of meaningful evaluations for most social impact categories and NP2 discussed *"the lack of research personnel to [conduct social impact evaluation]."* Importantly, these accounts align with our finding that environmental, privacy, and labor reporting received near-zero scores in first-party reporting, indicating that missingness is not anecdotal but a structural pattern. This suggests that policy interventions to reduce reputational and legal risk–alongside investment in methodological infrastructure and evaluations available to the broader ecosystem–may be prerequisites for progress.

**Popularity-driven evaluations create transparency gaps in low-resource models.** Because evaluations tend to follow the most prominent and commercially influential systems, models developed in the US and China (Gibney, 2024) attract the bulk of third-party scrutiny. This popularity-driven focus is unsurprising given the centrality of these models to global markets and policy debates. In addition, A2 notes a *"resource discrepancy where the evaluation resources can be not as rich for certain regions."* However, the consequence is a systematic blind spot: low-resource models receive far less attention on their social impacts and risks. This pattern may partly reflect our own selection strategy, which favors models with higher visibility (see Sec. 4), but even accounting for this bias, the effect remains clear—whole categories of models are left under examined.

**Opportunities for reform include standardized frameworks, safe harbors, and collective action.** Persistent reporting gaps point to the need for structural reforms that make reporting feasible and incentivized. Investment in evaluation tools could reduce ambiguity and lower burdens. FP2 called for a *"good, high quality framework"* and automated tools for impact measurement. Others suggested standardized templates but cautioned these must be practical. Field-wide coordination was suggested as critical: A non-profit interviewee stressed that many issues cannot be solved *"in house but require community-wide engagement"* and *"agreement on standards,"* noting that *"there's this missing piece in the community that doesn't make [better reporting] happen. It means legislation."* A1 cautioned, however, that while there is a need for *"collaboration between government and regulatory bodies and technical experts, not necessarily from companies, to prevent regulatory capture."* To this end, policy mechanisms such as safe harbors (Longpre et al., 2024), coordinated disclosures (Cattell et al., 2024), or regulatory sandboxes (European Commission, 2024) could mitigate developers' fear of legal or reputational harm. Overall, multi-stakeholder initiatives and coordination are necessary to move from fragmented reporting to consistent, comprehensive coverage.

## 7. Limitations and Future Work

Our study has methodological constraints that limit generalizability. Our interview sample was small and not representative of smaller organizations and global majority countries. In particular, response rates from regions outside Europe and North America were substantially lower, and because the vast majority of currently deployed foundation models are developed in these two regions, it is more difficult to obtain a regionally diverse sample of interviewees who meet our criteria. Due to our sampling strategy for model providers, this bias may extend to our empirical analysis. We assessed only the presence of social impact reporting, not the methodological soundness or adequacy of reported evaluations. Because meaningful assessment of evaluation quality is impossible without adequate information disclosure, our paper focuses on establishing this baseline transparency as a necessary foundation for future work on evaluation validity, robustness, and adequacy. We also examined a non-exhaustive set of categories, potentially missing under-reported subdomains that may be similarly neglected and harder to quantify. Future work could define good and optimal evaluations within each dimension and establish adequacy criteria (cf. Reuel-Lamparth et al. (2024), Salaudeen et al. (2025)), including identifying appropriate decision-makers (Raji et al. (2020)). Research could also link base-system propensities to societal impacts, analyze categories more granularly, and develop standardized evaluation templates and reporting frameworks.

## Impact Statement

**Policy considerations.** Our empirical findings and developer interviews highlight structural barriers that limit the transparency and consistency of social impact evaluations. This work may guide several policy recommendations ob-

served disclosure patterns and the policy drivers surfaced in interviews. **Safe-harbor provisions** protecting good-faith reporting could reduce legal and reputational concerns that discourage disclosure. **Standardized reporting templates** and multi-stakeholder coordination mechanisms would improve comparability across providers and establish shared norms. **Public infrastructure for secure reporting**, such as compute- or energy-use APIs, would lower reporting costs and level the playing field between large and small developers. **Strengthening third-party evaluation ecosystems** through research access provisions and sustained funding would provide crucial coverage of under-reported social impacts.

**Potential Risks** While our analysis aims to improve transparency, several risks warrant consideration. Publicly highlighting gaps could incentivize superficial compliance rather than substantive improvement. Critically, even well-intentioned evaluations can perpetuate harm if poorly designed. Evaluations relying on flawed benchmarks or narrow conceptions of harm may fail to capture impacts on marginalized communities. For instance, bias evaluations focusing only on binary gender categories erase non-binary individuals, and aggregated performance metrics may mask intersectional harms. Evaluations without meaningful input from affected communities risk operationalizing researcher assumptions rather than lived experiences. Additionally, without support mechanisms, increased reporting requirements could disproportionately burden smaller developers and organizations in the Global South.

**Broader implications.** By documenting systematic gaps in social impact reporting and identifying their structural causes, this work provides an evidence base for targeted interventions. Our findings can inform the development of reporting frameworks that balance comprehensiveness with feasibility, support multi-stakeholder participation in evaluation design, and enable more informed decision-making by policymakers, researchers, and the public. When paired with critical examination of evaluation methodology itself and mechanisms for meaningful stakeholder engagement, improved reporting practices can contribute to more accountable and socially responsible AI development.

## Acknowledgements

Srishti Yadav is supported in part by the Pioneer Centre for AI, DNRF grant number P1. Anka Reuel was supported by the Stanford Interdisciplinary Graduate Fellowship. Jenny Chim is supported through the EPSRC [grant number EP/Y009800/1] and through funding from Responsible Ai UK (KP0016). Mubashara Akhtar is supported through the ETH AI Center. Jan Batzner is supported through the German Federal Ministry of Research, Technology and Space (16DII131). Sanmi Koyejo is partially supported by NSF 2046795 and 2205329, IES R305C240046, ARPA-H, the MacArthur Foundation, Schmidt Sciences, Stanford HAI, RAISE Health, OpenAI, Microsoft, and Google.

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

# Appendix

**Table of Contents**

## A. Terminological Notes

**AI Systems and AI Models.**   We adopt the perspective of Basdevant et al. (2024), understanding AI systems as encompassing infrastructure (e.g., compilers), model components (e.g., datasets, code, and weights), and user-facing elements such as user interface components. In this paper, AI systems denotes models situated within an application context, whereas AI models refers to model components independent of deployment.

**Assessment and Evaluation.**   While "assessment" has historically been the dominant term in the social sciences, we deliberately adopt the language of social impact "evaluations" to signal methodological alignment with the computer science and AI governance literature, which frames evaluations as structured procedures that generate evidence about system properties. The distinction is primarily one of epistemic approach rather than operationalization: "assessments" typically emphasize open-ended inquiry that allows for emergent findings and interpretive flexibility (akin to essay-based examinations), whereas "evaluations" tend toward structured, predetermined criteria that enable systematic comparison across cases (akin to standardized testing). Both can be operationalized and made comparable; they differ in whether the scope of inquiry is determined a priori or allowed to emerge through the investigative process.

**Foundation Model.**   We acknowledge that this term, while commonly used, is somewhat nebulous; alternatives like "pre-trained model" or "base system" each have their own limitations in capturing the full scope of models trained without specific downstream applications. For our purposes, we use "foundation model" to refer to pre-trained generative AI models (language-only or multimodal) that have not been developed with an explicit application context. We include instruction-tuned models in this category, as they remain general-purpose systems not yet integrated into specific application contexts, while excluding models fine-tuned for particular use cases.

## B. Stratified Sampling Procedure for Main-Text Figures

To avoid overcrowding in the main-text figures while maintaining representativeness, we used a stratified sampling approach to select 17 providers. Stratification was based on geography (East Asia, North America, Western Europe, Middle East) and organizational type (industry, academia, government), with separate strata for open- and closed-weight models. For each stratum, we sampled two providers when available. In cases where only one provider existed (e.g., closed-Middle East), we included that single provider. Categories without representation (academia-closed, nonprofit-closed) were excluded.

If a provider appeared in multiple strata (e.g., OpenAI as both industry/closed and North America/closed), we retained the overlap rather than replacing it with another provider. For figures where the unit of analysis is models, we selected the most recent model released by each provider. For figures where the unit of analysis is providers, we averaged across all available models from that provider. This procedure yields a stratified sample of 16 providers, which balances coverage across regions, organizational types, and openness while keeping figures interpretable.

The final sample of providers included in main paper plots based on the procedure above are: Ai2, AI21Labs, Anthropic, Baichuan, BigScience, Cohere, DeepSeek, EleutherAI, Google, HuggingFace, Meta, Mistral, OpenAI, StabilityAI, TII, Z.ai.

## C. List of analyzed providers

*Table 2.* Categorization of providers by model weight accessibility, geographic region and governance type.

| Provider | Weight Accessibility | | | Geographic Region | Governance Type |
|---|---|---|---|---|---|
| | **Open** | **Closed** | **Total** | | |
| 01.AI | 2 | 0 | 2 | East Asia | Industry |
| Ai2 | 6 | 0 | 6 | North America | Nonprofit |
| AI21 | 1 | 0 | 1 | Middle East and North Africa | Industry |
| AIForever | 1 | 0 | 1 | Europe | Nonprofit |
| Alibaba | 10 | 0 | 10 | East Asia | Industry |
| Amazon | 0 | 1 | 1 | North America | Industry |
| AntGroup | 0 | 1 | 1 | East Asia | Industry |

Table 2 contd. from previous page

| Provider | Weight Accessibility | | | Geographic Region | Governance Type |
|---|---|---|---|---|---|
| | Open | Closed | Total | | |
| Anthropic | 0 | 9 | 9 | North America | Industry |
| Apple | 1 | 0 | 1 | North America | Industry |
| BAAI | 1 | 0 | 1 | East Asia | Nonprofit |
| BAAI_TeleAI | 1 | 0 | 1 | East Asia | Industry |
| Baichuan | 2 | 0 | 2 | East Asia | Industry |
| Baidu | 1 | 1 | 2 | East Asia | Industry |
| BigScience | 3 | 0 | 3 | Europe | Academia |
| ByteDance | 1 | 0 | 1 | East Asia | Industry |
| Cohere | 5 | 1 | 6 | North America | Industry |
| CompVis | 2 | 0 | 2 | Europe | Academia |
| DeepSeek | 5 | 0 | 5 | East Asia | Industry |
| EleutherAI | 4 | 0 | 4 | North America | Nonprofit |
| EuroLLM | 1 | 0 | 1 | Europe | Academia |
| G42 | 1 | 0 | 1 | Middle East and North Africa | Government |
| Genmo | 1 | 0 | 1 | North America | Industry |
| Google | 6 | 9 | 15 | North America | Industry |
| Hugging Face | 5 | 0 | 5 | Europe | Industry |
| IBM | 2 | 0 | 2 | North America | Industry |
| iFLYTEK | 0 | 3 | 3 | East Asia | Industry |
| InceptionLabs | 0 | 1 | 1 | North America | Industry |
| JieyueStar | 1 | 0 | 1 | East Asia | Government |
| Lightricks | 1 | 0 | 1 | Middle East and North Africa | Industry |
| MCINext | 0 | 1 | 1 | Middle East and North Africa | Government |
| Meta | 7 | 1 | 8 | North America | Industry |
| Microsoft | 8 | 0 | 8 | North America | Industry |
| MiniMax | 1 | 1 | 2 | East Asia | Industry |
| Mistral | 8 | 3 | 11 | Europe | Industry |
| MoonshotAI | 3 | 1 | 4 | East Asia | Industry |
| Mosaic | 2 | 0 | 2 | North America | Industry |
| NAVER | 1 | 0 | 1 | East Asia | Industry |
| Neulab | 0 | 1 | 1 | North America | Academia |
| NVIDIA | 2 | 0 | 2 | North America | Industry |
| OpenAI | 3 | 14 | 17 | North America | Industry |
| RhymesAI | 1 | 0 | 1 | North America | Industry |
| RunwayML | 1 | 0 | 1 | North America | Industry |
| Salesforce | 4 | 0 | 4 | North America | Industry |
| SarvamAI | 1 | 0 | 1 | South and Central Asia | Industry |
| SenseTime | 0 | 2 | 2 | East Asia | Industry |
| ShanghaiAILab | 6 | 0 | 6 | East Asia | Government |
| SKTelecom | 0 | 1 | 1 | East Asia | Industry |
| StabilityAI | 8 | 0 | 8 | Europe | Industry |
| Tencent | 1 | 0 | 1 | East Asia | Industry |
| TII | 2 | 0 | 2 | Middle East and North Africa | Government |
| USTC, ShanghaiAILab, CUHK | 1 | 0 | 1 | East Asia | Academia |
| Writer | 0 | 1 | 1 | North America | Industry |
| xAI | 1 | 2 | 3 | North America | Industry |
| Z.ai | 6 | 1 | 7 | East Asia | Industry |

## D. Full results

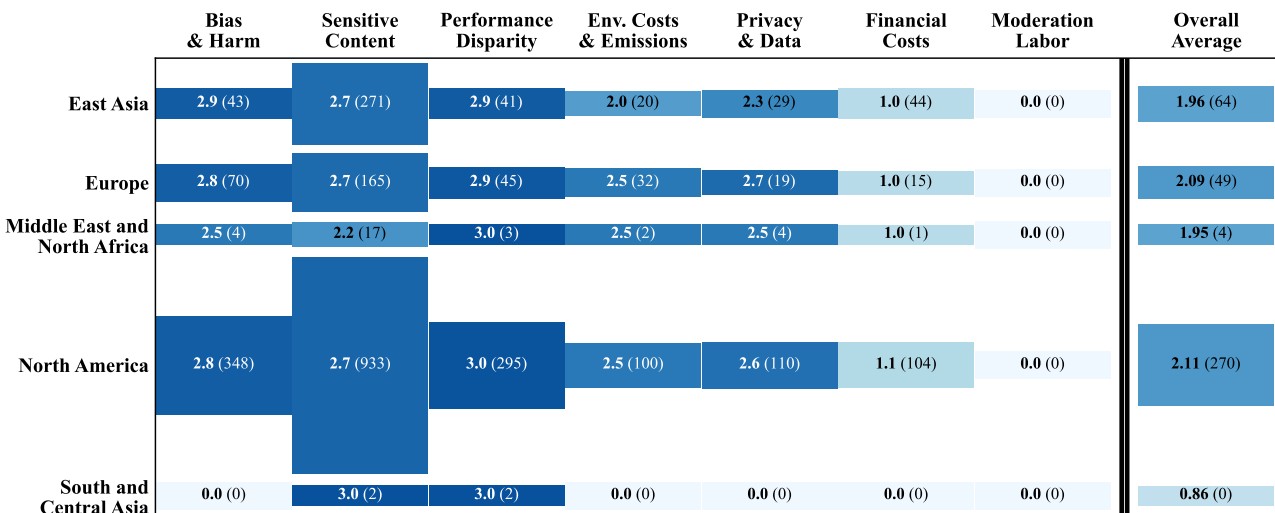

*Figure 5.* **Average scores and counts for social impact evaluations by provider geographic region.** Here, rectangle size corresponds log-linearly to the number of evaluation (more evaluations result in larger rectangles), and color indicates average score for reporting detail (darker colors indicate higher scores). Please refer to Section 4, and App. D for scoring methodology and full results, respectively. Regions not shown had no models in our dataset.

| | Bias & Harm | Sensitive Content | Performance Disparity | Env. Costs & Emissions | Privacy & Data | Financial Costs | Moderation Labor | Overall Average |
|---|---|---|---|---|---|---|---|---|
| Academia | 0.38 | 0.88 | 1.62 | 1.50 | 0.12 | 0.75 | 0.12 | 0.77 |
| Government | 0.36 | 0.64 | 0.82 | 0.27 | 0.00 | 0.45 | 0.00 | 0.36 |
| Industry | 0.76 | 0.97 | 0.70 | 0.59 | 0.30 | 0.51 | 0.13 | 0.56 |
| Nonprofit | 0.50 | 0.50 | 0.25 | 1.33 | 0.33 | 0.83 | 0.17 | 0.56 |

*Figure 6.* **Average scores for first-party social impact reporting per sector.** Color indicates the average reporting detail level: Darker colors indicate higher scores. See **Scoring** in Section 4 for details, and in App. D for a complete analysis.

| | Bias & Harm | Sensitive Content | Performance Disparity | Env. Costs & Emissions | Privacy & Data | Financial Costs | Moderation Labor | Overall Average |
|---|---|---|---|---|---|---|---|---|
| Armenia | 0.0 | 0.0 | 0.0 | 1.0 | 0.0 | 1.0 | 0.0 | 0.29 |
| Canada | 1.83 | 1.67 | 2.0 | 0.33 | 0.17 | 0.33 | 0.67 | 1.0 |
| China | 0.25 | 0.65 | 0.48 | 0.23 | 0.15 | 0.38 | 0.04 | 0.31 |
| France | 0.47 | 0.53 | 1.16 | 0.37 | 0.05 | 0.42 | 0.05 | 0.44 |
| Germany | 0.0 | 1.0 | 0.0 | 3.0 | 0.0 | 1.0 | 0.0 | 0.71 |
| India | 0.0 | 0.0 | 2.0 | 1.0 | 0.0 | 1.0 | 0.0 | 0.57 |
| Iran | 0.0 | 0.0 | 0.0 | 0.0 | 0.0 | 0.0 | 0.0 | 0.0 |
| Israel | 0.0 | 0.5 | 0.0 | 0.5 | 0.0 | 1.0 | 0.0 | 0.29 |
| EU | 0.0 | 0.0 | 3.0 | 1.0 | 0.0 | 1.0 | 0.0 | 0.71 |
| S. Korea | 0.0 | 1.0 | 0.0 | 0.5 | 0.0 | 0.5 | 0.0 | 0.29 |
| UAE | 0.67 | 1.33 | 1.0 | 0.67 | 0.0 | 1.33 | 0.0 | 0.71 |
| UK | 0.0 | 0.38 | 0.25 | 1.5 | 0.25 | 1.0 | 0.0 | 0.48 |
| US | 1.09 | 1.18 | 0.73 | 0.86 | 0.45 | 0.57 | 0.18 | 0.72 |

*Figure 7.* **First-party social impact reporting by model provider country.** Color indicates the average reporting detail level: Darker colors indicate higher scores. See **Scoring** in Section 4 for details, and in App. D for details. "EU" is a Multi-Country EU Consortium.

## E. Per-provider reporting quality over time

| | Bias & Harm | Sensitive Content | Performance Disparity | Env. Costs & Emissions | Privacy & Data | Financial Costs | Moderation Labor | Overall Average |
|---|---|---|---|---|---|---|---|---|
| Armenia | 0.0 (0) | 3.0 (1) | 0.0 (0) | 0.0 (0) | 0.0 (0) | 0.0 (0) | 0.0 (0) | 0.43 (0) |
| Canada | 2.2 (5) | 2.7 (28) | 3.0 (8) | 2.3 (3) | 1.0 (2) | 1.0 (6) | 0.0 (0) | 1.74 (7) |
| China | 2.9 (40) | 2.7 (270) | 2.9 (40) | 2.0 (20) | 2.3 (29) | 1.0 (44) | 0.0 (0) | 1.97 (63) |
| France | 2.9 (57) | 2.7 (146) | 3.0 (42) | 2.6 (26) | 2.6 (16) | 1.0 (15) | 0.0 (0) | 2.11 (43) |
| Germany | 2.4 (9) | 2.6 (8) | 2.0 (2) | 2.0 (1) | 3.0 (1) | 0.0 (0) | 0.0 (0) | 1.72 (3) |
| India | 0.0 (0) | 3.0 (2) | 3.0 (2) | 0.0 (0) | 0.0 (0) | 0.0 (0) | 0.0 (0) | 0.86 (0) |
| Iran | 0.0 (0) | 3.0 (1) | 0.0 (0) | 0.0 (0) | 0.0 (0) | 0.0 (0) | 0.0 (0) | 0.43 (0) |
| Israel | 1.0 (1) | 0.0 (0) | 0.0 (0) | 2.0 (1) | 1.0 (1) | 1.0 (1) | 0.0 (0) | 0.71 (0) |
| EU | 0.0 (0) | 2.0 (1) | 0.0 (0) | 0.0 (0) | 0.0 (0) | 0.0 (0) | 0.0 (0) | 0.29 (0) |
| South Korea | 2.7 (3) | 2.0 (1) | 1.0 (1) | 0.0 (0) | 0.0 (0) | 0.0 (0) | 0.0 (0) | 0.81 (0) |
| UAE | 3.0 (3) | 2.1 (16) | 3.0 (3) | 3.0 (1) | 3.0 (3) | 0.0 (0) | 0.0 (0) | 2.02 (3) |
| UK | 3.0 (4) | 2.9 (9) | 2.0 (1) | 2.0 (5) | 3.0 (2) | 0.0 (0) | 0.0 (0) | 1.84 (3) |
| USA | 2.9 (343) | 2.7 (905) | 3.0 (287) | 2.5 (97) | 2.6 (108) | 1.1 (98) | 0.0 (0) | 2.11 (262) |

*Figure 8.* **Third-party social impact reporting by model provider country.** Here, rectangle size corresponds log-linearly to the number of evaluation (more evaluations result in larger rectangles), and color indicates average score for reporting detail (darker colors indicate higher scores). Please refer to Sec. 4, App. B, and App. D for scoring methodology, sampling procedure and provider list, and full results, respectively. "EU" is a Multi-Country EU Consortium.

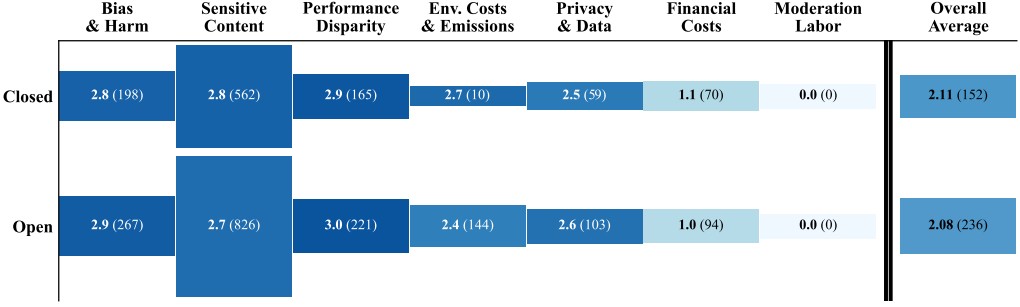

| | Bias & Harm | Sensitive Content | Performance Disparity | Env. Costs & Emissions | Privacy & Data | Financial Costs | Moderation Labor | Overall Average |
|---|---|---|---|---|---|---|---|---|
| Closed | 2.8 (198) | 2.8 (562) | 2.9 (165) | 2.7 (10) | 2.5 (59) | 1.1 (70) | 0.0 (0) | 2.11 (152) |
| Open | 2.9 (267) | 2.7 (826) | 3.0 (221) | 2.4 (144) | 2.6 (103) | 1.0 (94) | 0.0 (0) | 2.08 (236) |

*Figure 9.* **Evaluation scores for third-party social impact evaluation by level of openness**: either open (both open-source and open-weight) and closed. Here, rectangle size corresponds log-linearly to the number of evaluation (more evaluations result in larger rectangles), and color indicates average score for reporting detail (darker colors indicate higher scores). Please refer to Sec. 4, App. B, and App. D for scoring methodology, sampling procedure and provider list, and full results, respectively.

| | Bias & Harm | Sensitive Content | Performance Disparity | Env. Costs & Emissions | Privacy & Data | Financial Costs | Moderation Labor | Overall Average |
|---|---|---|---|---|---|---|---|---|
| 01.AI | 1.0 | 1.0 | 0.0 | 0.0 | 1.0 | 0.5 | 0.0 | 0.5 |
| Ai2 | 0.5 | 0.83 | 0.0 | 1.5 | 0.17 | 0.83 | 0.33 | 0.6 |
| AI21Labs | 0.0 | 1.0 | 0.0 | 0.0 | 0.0 | 1.0 | 0.0 | 0.29 |
| AIForever | 0.0 | 0.0 | 0.0 | 1.0 | 0.0 | 1.0 | 0.0 | 0.29 |
| Alibaba | 0.0 | 0.4 | 1.4 | 0.1 | 0.1 | 0.1 | 0.2 | 0.33 |
| Amazon | 1.0 | 1.0 | 1.0 | 0.0 | 1.0 | 0.0 | 0.0 | 0.57 |
| Anthropic | 1.22 | 1.33 | 0.67 | 0.0 | 0.11 | 0.22 | 0.0 | 0.51 |
| AntGroup | 0.0 | 0.0 | 0.0 | 0.0 | 0.0 | 0.0 | 0.0 | 0.0 |
| Apple | 2.0 | 2.0 | 0.0 | 1.0 | 0.0 | 1.0 | 0.0 | 0.86 |
| BAAI | 0.0 | 1.0 | 0.0 | 1.0 | 0.0 | 1.0 | 0.0 | 0.43 |
| BAAI_TeleAI | 0.0 | 2.0 | 0.0 | 1.0 | 0.0 | 1.0 | 0.0 | 0.57 |
| Baichuan | 1.0 | 1.0 | 1.0 | 0.0 | 0.0 | 0.5 | 0.0 | 0.5 |
| Baidu | 0.0 | 0.0 | 0.0 | 0.5 | 0.0 | 0.5 | 0.0 | 0.14 |
| BigScience | 1.0 | 0.67 | 2.33 | 1.0 | 0.33 | 0.33 | 0.33 | 0.86 |
| ByteDance | 0.0 | 2.0 | 0.0 | 0.0 | 0.0 | 0.0 | 0.0 | 0.29 |
| Cohere | 1.83 | 1.67 | 2.0 | 0.33 | 0.17 | 0.33 | 0.67 | 1.0 |
| CompVis | 0.0 | 1.0 | 0.0 | 3.0 | 0.0 | 1.0 | 0.0 | 0.71 |
| DeepSeek | 1.0 | 1.2 | 0.6 | 0.4 | 0.8 | 1.4 | 0.0 | 0.77 |
| EleutherAI | 0.75 | 0.0 | 0.75 | 1.25 | 0.75 | 0.75 | 0.0 | 0.61 |
| EuroLLM | 0.0 | 0.0 | 3.0 | 1.0 | 0.0 | 1.0 | 0.0 | 0.71 |
| G42 | 2.0 | 2.0 | 0.0 | 1.0 | 0.0 | 1.0 | 0.0 | 0.86 |
| Genmo | 0.0 | 0.0 | 0.0 | 0.0 | 0.0 | 0.0 | 0.0 | 0.0 |
| Google | 2.0 | 2.0 | 1.33 | 1.13 | 1.27 | 0.67 | 0.67 | 1.30 |
| HuggingFace | 0.6 | 0.6 | 0.0 | 0.8 | 0.0 | 1.0 | 0.0 | 0.43 |
| IBM | 1.0 | 1.0 | 0.0 | 1.5 | 1.0 | 0.5 | 0.0 | 0.71 |
| iFLYTEK | 0.0 | 0.0 | 0.0 | 0.0 | 0.0 | 0.0 | 0.0 | 0.0 |
| InceptionLabs | 0.0 | 0.0 | 0.0 | 0.0 | 0.0 | 0.0 | 0.0 | 0.0 |
| JieyueStar | 0.0 | 0.0 | 0.0 | 0.0 | 0.0 | 0.0 | 0.0 | 0.0 |
| Lightricks | 0.0 | 0.0 | 0.0 | 1.0 | 0.0 | 1.0 | 0.0 | 0.29 |
| MCINext | 0.0 | 0.0 | 0.0 | 0.0 | 0.0 | 0.0 | 0.0 | 0.0 |
| Meta | 1.5 | 1.75 | 0.75 | 2.75 | 0.5 | 1.0 | 0.0 | 1.18 |
| Microsoft | 0.88 | 1.5 | 0.62 | 0.88 | 0.12 | 0.88 | 0.0 | 0.7 |
| MiniMax | 0.0 | 0.0 | 0.0 | 0.5 | 0.0 | 1.0 | 0.0 | 0.21 |
| Mistral | 0.27 | 0.45 | 1.36 | 0.0 | 0.0 | 0.18 | 0.0 | 0.32 |
| MoonshotAI | 0.0 | 0.5 | 0.0 | 0.5 | 0.25 | 0.5 | 0.0 | 0.25 |
| MosaicML | 0.0 | 0.0 | 0.0 | 1.0 | 0.0 | 2.0 | 0.0 | 0.43 |
| NAVER | 0.0 | 2.0 | 0.0 | 1.0 | 0.0 | 1.0 | 0.0 | 0.57 |
| NeuLab | 0.0 | 3.0 | 3.0 | 1.0 | 0.0 | 1.0 | 0.0 | 1.14 |
| NVIDIA | 0.0 | 0.0 | 0.0 | 0.0 | 0.0 | 0.5 | 0.0 | 0.07 |
| OpenAI | 1.47 | 1.29 | 1.18 | 0.29 | 0.47 | 0.24 | 0.24 | 0.74 |
| RhymesAI | 0.0 | 0.0 | 0.0 | 0.0 | 0.0 | 0.0 | 0.0 | 0.0 |
| RunwayML | 0.0 | 1.0 | 0.0 | 3.0 | 0.0 | 1.0 | 0.0 | 0.71 |
| Salesforce | 0.0 | 0.0 | 0.0 | 0.25 | 0.0 | 0.25 | 0.0 | 0.07 |
| SarvamAI | 0.0 | 0.0 | 2.0 | 1.0 | 0.0 | 1.0 | 0.0 | 0.57 |
| SenseTime | 0.0 | 0.0 | 0.0 | 0.0 | 0.0 | 0.0 | 0.0 | 0.0 |
| ShanghaiAILab | 0.33 | 0.5 | 1.0 | 0.17 | 0.0 | 0.17 | 0.0 | 0.31 |
| SKTelecom | 0.0 | 0.0 | 0.0 | 0.0 | 0.0 | 0.0 | 0.0 | 0.0 |
| StabilityAI | 0.0 | 0.38 | 0.25 | 1.5 | 0.25 | 1.0 | 0.0 | 0.48 |
| Tencent | 0.0 | 0.0 | 0.0 | 0.0 | 0.0 | 0.0 | 0.0 | 0.0 |
| TII | 0.0 | 1.0 | 1.5 | 0.5 | 0.0 | 1.5 | 0.0 | 0.64 |
| USTC_Shanghai | 0.0 | 0.0 | 0.0 | 1.0 | 0.0 | 1.0 | 0.0 | 0.29 |
| Writer | 0.0 | 0.0 | 0.0 | 0.0 | 0.0 | 1.0 | 0.0 | 0.14 |
| xAI | 0.0 | 0.0 | 0.0 | 0.0 | 0.0 | 0.0 | 0.0 | 0.0 |
| Z.ai | 0.29 | 1.43 | 0.0 | 0.14 | 0.0 | 0.14 | 0.0 | 0.29 |

*Figure 10.* **Average scores for first-party social impact reporting per provider.** Color indicates the average reporting detail level: Darker colors indicate higher scores. See **Scoring** in Section 4 for details, and in App. D for a complete analysis.

| | Bias & Harm | Sensitive Content | Performance Disparity | Env. Costs & Emissions | Privacy & Data | Financial Costs | Moderation Labor | Overall Average |
|---|---|---|---|---|---|---|---|---|
| **Ai2** | 0.0 (0) | 3.0 (6) | 0.0 (0) | 3.0 (5) | 0.0 (0) | 0.0 (0) | 0.0 (0) | 0.86 (1) |
| **AI21Labs** | 1.0 (1) | 0.0 (0) | 0.0 (0) | 0.0 (0) | 1.0 (1) | 1.0 (1) | 0.0 (0) | 0.43 (0) |
| **AIForever** | 0.0 (0) | 3.0 (1) | 0.0 (0) | 0.0 (0) | 0.0 (0) | 0.0 (0) | 0.0 (0) | 0.43 (0) |
| **Alibaba** | 2.8 (15) | 2.5 (134) | 3.0 (7) | 2.1 (18) | 2.3 (11) | 1.0 (31) | 0.0 (0) | 1.94 (30) |
| **Amazon** | 3.0 (3) | 3.0 (3) | 0.0 (0) | 0.0 (0) | 3.0 (3) | 1.0 (4) | 0.0 (0) | 1.43 (1) |
| **AntGroup** | 0.0 (0) | 3.0 (1) | 0.0 (0) | 0.0 (0) | 0.0 (0) | 0.0 (0) | 0.0 (0) | 0.43 (0) |
| **Anthropic** | 2.8 (29) | 2.9 (101) | 2.9 (36) | 0.0 (0) | 2.0 (6) | 1.0 (12) | 0.0 (0) | 1.66 (26) |
| **BAAI** | 0.0 (0) | 3.0 (1) | 0.0 (0) | 0.0 (0) | 0.0 (0) | 0.0 (0) | 0.0 (0) | 0.43 (0) |
| **BAAI_TeleAI** | 0.0 (0) | 3.0 (1) | 0.0 (0) | 0.0 (0) | 0.0 (0) | 0.0 (0) | 0.0 (0) | 0.43 (0) |
| **Baichuan** | 3.0 (2) | 3.0 (21) | 3.0 (10) | 0.0 (0) | 1.7 (3) | 0.0 (0) | 0.0 (0) | 1.52 (5) |
| **Baidu** | 0.0 (0) | 3.0 (2) | 0.0 (0) | 0.0 (0) | 1.0 (1) | 0.0 (0) | 0.0 (0) | 0.57 (0) |
| **BigScience** | 3.0 (19) | 2.6 (29) | 3.0 (20) | 2.9 (7) | 3.0 (1) | 0.0 (0) | 0.0 (0) | 2.06 (10) |
| **Cohere** | 2.2 (5) | 2.7 (28) | 3.0 (8) | 2.3 (3) | 1.0 (2) | 1.0 (6) | 0.0 (0) | 1.74 (7) |
| **CompVis** | 2.4 (9) | 2.6 (8) | 2.0 (2) | 2.0 (1) | 3.0 (1) | 0.0 (0) | 0.0 (0) | 1.72 (3) |
| **DeepSeek** | 2.8 (6) | 2.8 (43) | 3.0 (9) | 0.0 (0) | 1.0 (1) | 1.0 (7) | 0.0 (0) | 1.52 (9) |
| **EleutherAI** | 3.0 (7) | 2.7 (18) | 3.0 (4) | 2.0 (9) | 3.0 (8) | 0.0 (0) | 0.0 (0) | 1.96 (6) |
| **EuroLLM** | 0.0 (0) | 2.0 (1) | 0.0 (0) | 0.0 (0) | 0.0 (0) | 0.0 (0) | 0.0 (0) | 0.29 (0) |
| **G42** | 0.0 (0) | 3.0 (1) | 0.0 (0) | 0.0 (0) | 0.0 (0) | 0.0 (0) | 0.0 (0) | 0.43 (0) |
| **Genmo** | 0.0 (0) | 0.0 (0) | 0.0 (0) | 2.0 (1) | 0.0 (0) | 0.0 (0) | 0.0 (0) | 0.29 (0) |
| **Google** | 2.8 (27) | 2.7 (127) | 3.0 (42) | 2.7 (18) | 2.7 (12) | 1.0 (16) | 0.0 (0) | 2.11 (34) |
| **HuggingFace** | 0.0 (0) | 2.0 (5) | 0.0 (0) | 3.0 (4) | 0.0 (0) | 1.0 (1) | 0.0 (0) | 0.86 (1) |
| **iFLYTEK** | 0.0 (0) | 3.0 (2) | 0.0 (0) | 0.0 (0) | 1.0 (1) | 0.0 (0) | 0.0 (0) | 0.57 (0) |
| **InceptionLabs** | 0.0 (0) | 0.0 (0) | 0.0 (0) | 0.0 (0) | 0.0 (0) | 1.0 (1) | 0.0 (0) | 0.14 (0) |
| **JieyueStar** | 0.0 (0) | 3.0 (1) | 0.0 (0) | 0.0 (0) | 0.0 (0) | 0.0 (0) | 0.0 (0) | 0.43 (0) |
| **Lightricks** | 0.0 (0) | 0.0 (0) | 0.0 (0) | 2.0 (1) | 0.0 (0) | 0.0 (0) | 0.0 (0) | 0.29 (0) |

*Figure 11.* Evaluation scores for third-party social impact evaluation per provider for all providers (Contd. on next page.)

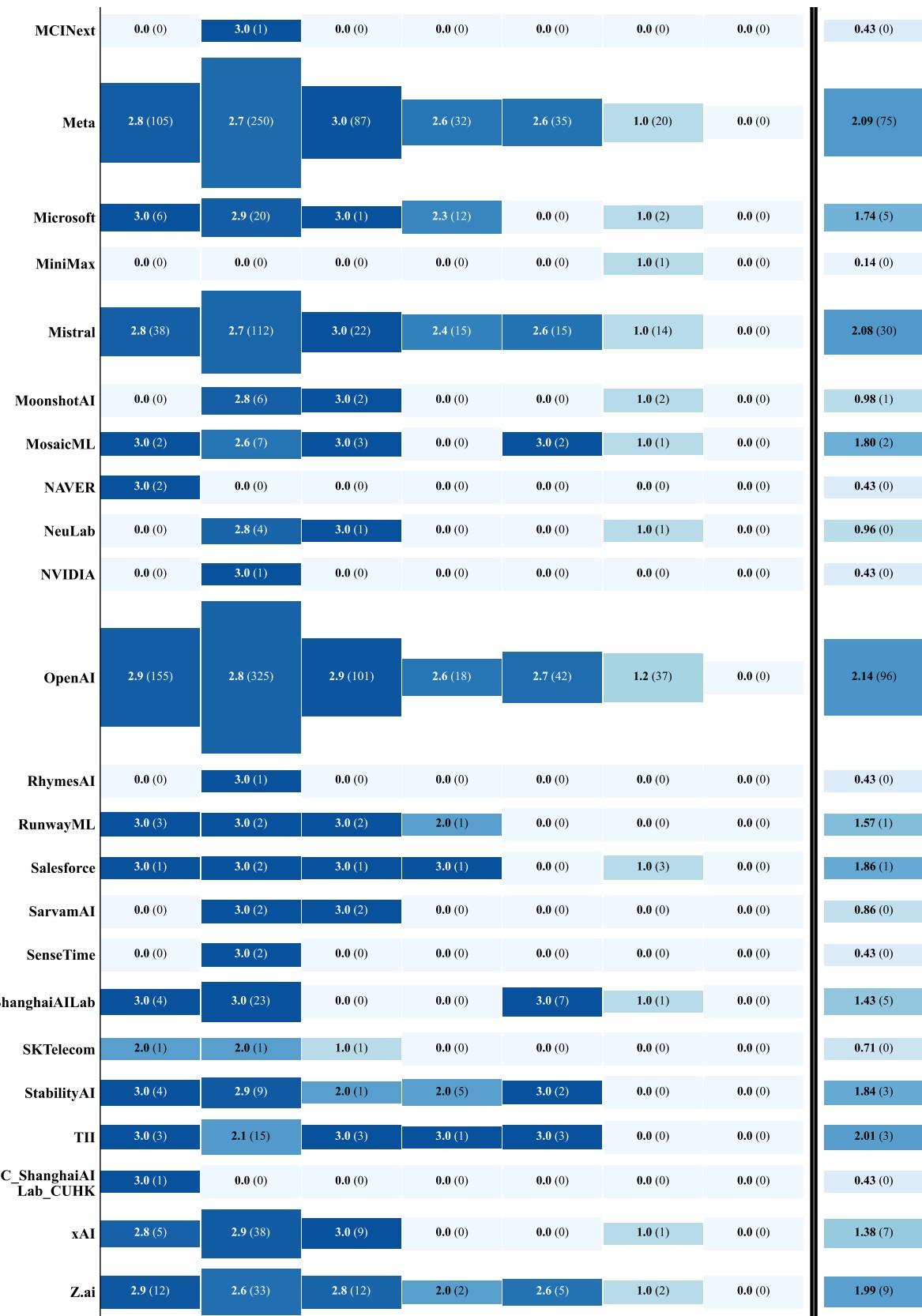

*Figure 11.* **Evaluation scores for third-party social impact evaluations per provider for all providers** (contd.). Here, rectangle size corresponds log-linearly to the number of evaluation (more evaluations result in larger rectangles), and color indicates average score for reporting detail (darker colors indicate higher scores). Please refer to Sec. 4, App. B, and App. D for scoring methodology, sampling procedure and provider list, and full results, respectively.

## Ai2

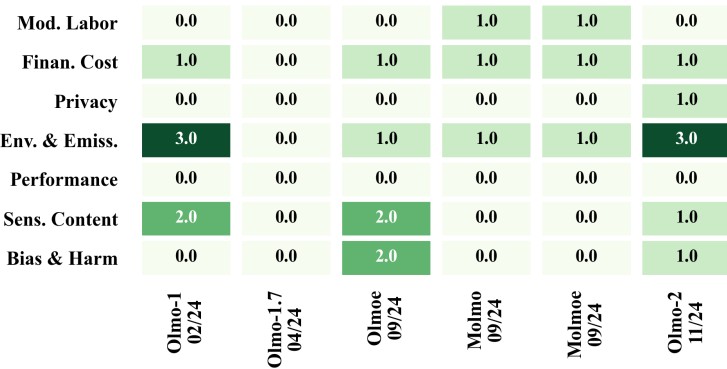

*Figure 12.* Ai2 reporting quality over time.

## AI21Labs

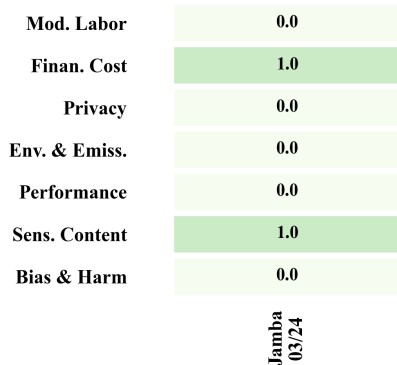

*Figure 13.* AI21 Labs reporting quality over time.

## Anthropic

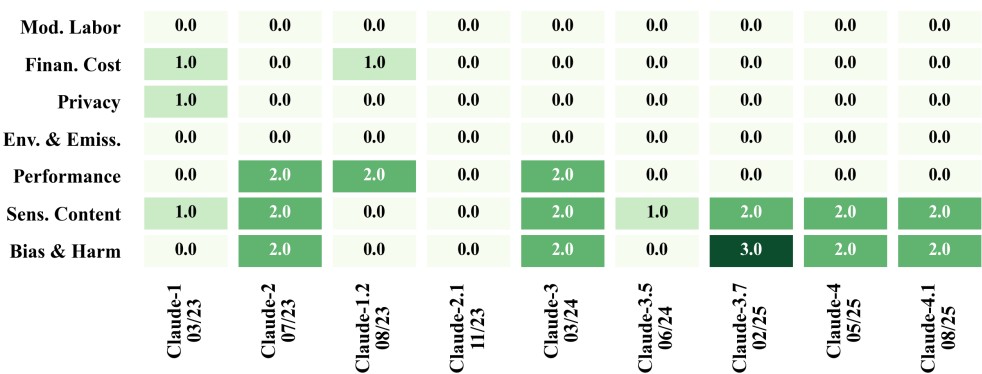

*Figure 14.* Anthropic reporting quality over time.

## Baichuan

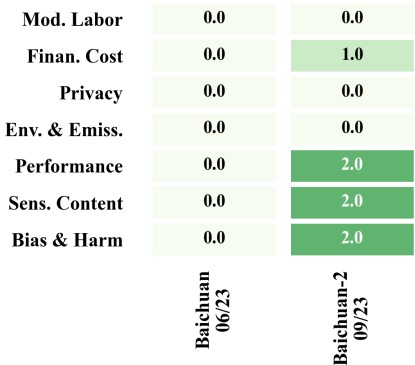

*Figure 15.* Baichuan reporting quality over time.

## BigScience

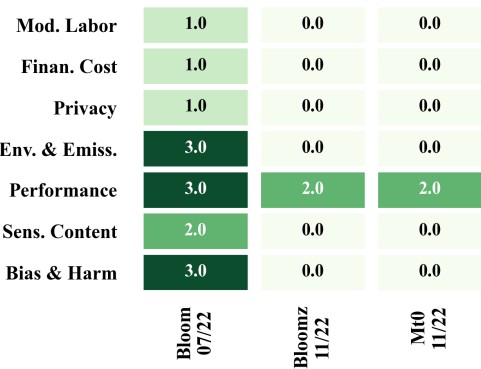

*Figure 16.* BigScience reporting quality over time.

## Cohere

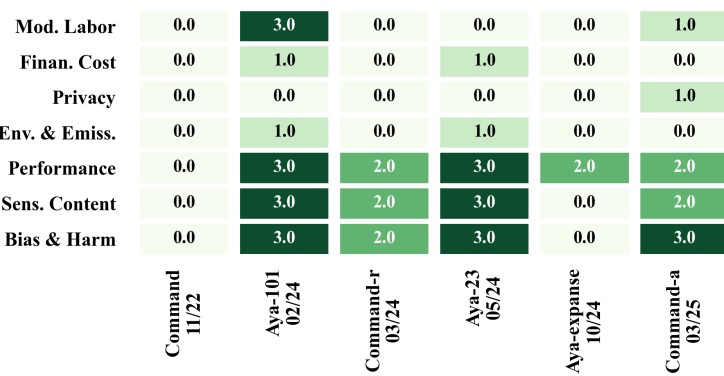

*Figure 17.* Cohere reporting quality over time.

## DeepSeek

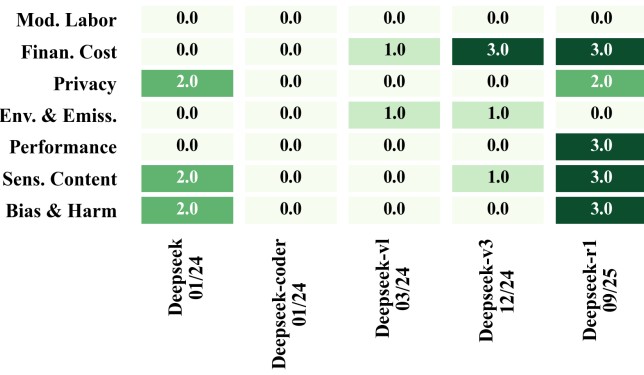

*Figure 18.* DeepSeek reporting quality over time.

## EleutherAI

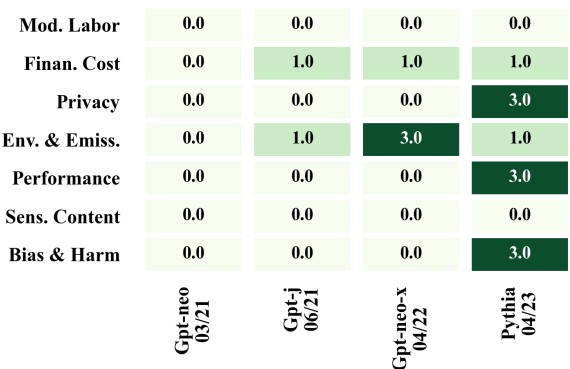

*Figure 19.* EleutherAI reporting quality over time.

## Google

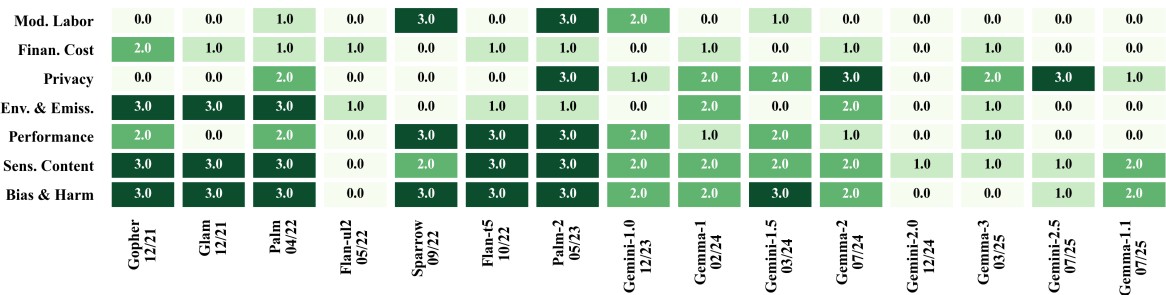

*Figure 20.* Google reporting quality over time.

## HuggingFace

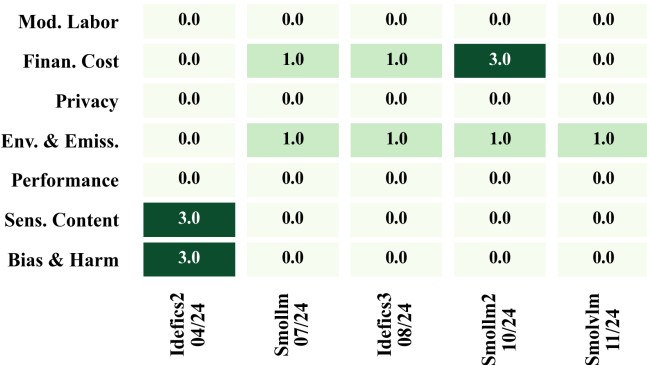

*Figure 21.* HuggingFace reporting quality over time.

## Meta

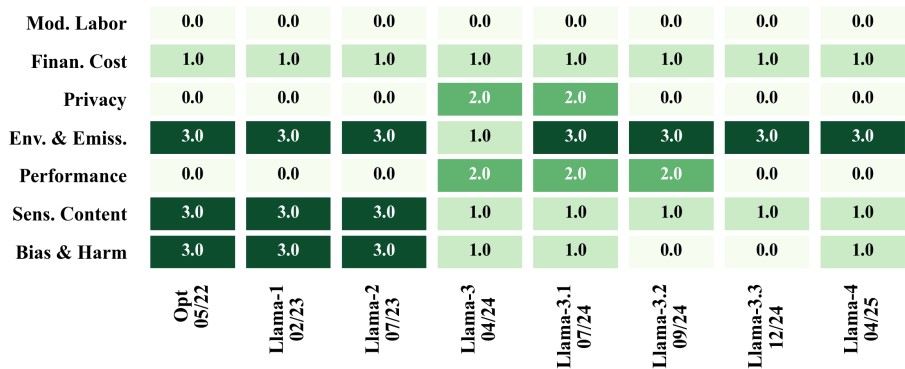

*Figure 22.* Meta reporting quality over time.

## Mistral

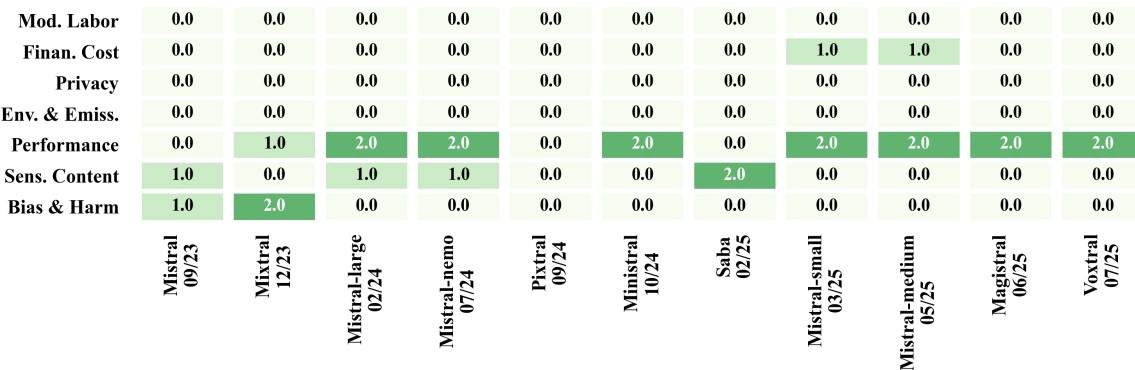

*Figure 23.* Mistral reporting quality over time.

## OpenAI

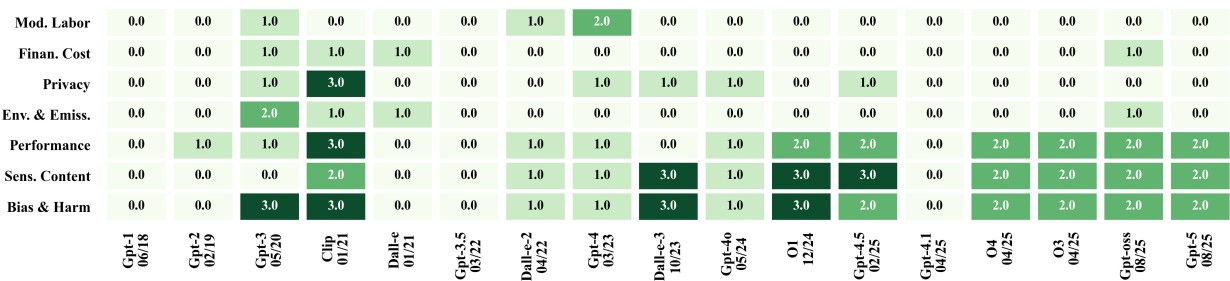

*Figure 24.* OpenAI reporting quality over time.

## StabilityAI

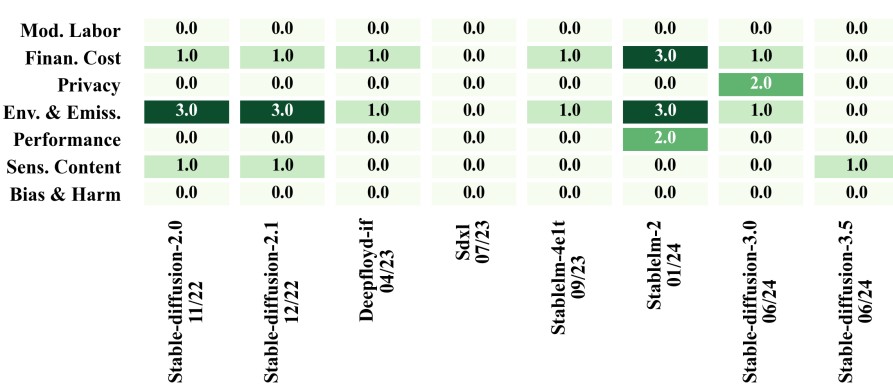

*Figure 25.* StabilityAI reporting quality over time.

## TII

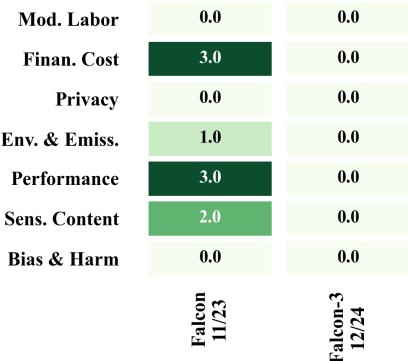

*Figure 26.* TII reporting quality over time.

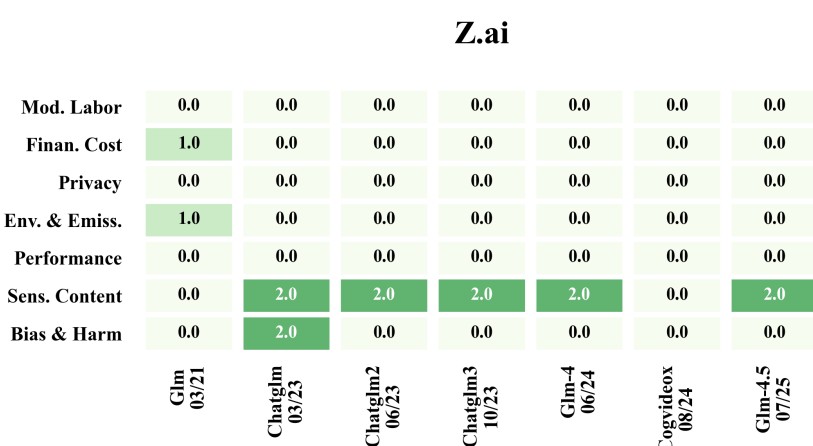

*Figure 27.* Z.ai reporting quality over time.

# F. Summary statistics

*Table 3.* Overall distribution

| category | mean | median | var |
|---|---|---|---|
| 1 | 2.246 | 3.0 | 1.380 |
| 2 | 2.446 | 3.0 | 0.771 |
| 3 | 2.239 | 3.0 | 1.447 |
| 4 | 1.584 | 2.0 | 1.481 |
| 5 | 1.351 | 1.0 | 1.800 |
| 6 | 0.779 | 1.0 | 0.354 |
| 7 | 0.124 | 0.0 | 0.228 |

*Table 4.* Proportion of first-party vs. third-party reports by category

| category | is_first_party | is_third_party | is_first_party_proportion |
|---|---|---|---|
| 1 | 219 | 465 | 0.320 |
| 2 | 282 | 1388 | 0.169 |
| 3 | 224 | 386 | 0.367 |
| 4 | 214 | 154 | 0.582 |
| 5 | 208 | 162 | 0.562 |
| 6 | 189 | 164 | 0.535 |
| 7 | 186 | 0 | 1.000 |

*Table 5.* Top-level missingness rate per dimension

| category | n_missing | n_total | proportion_missing |
|---|---|---|---|
| 1 | 125 | 186 | 0.672 |
| 2 | 93 | 186 | 0.500 |
| 3 | 121 | 186 | 0.651 |
| 4 | 109 | 186 | 0.586 |
| 5 | 154 | 186 | 0.828 |
| 6 | 100 | 186 | 0.538 |
| 7 | 171 | 186 | 0.919 |

*Table 6.* Average number of social-impact dimensions evaluated per report by sector

| sector | average |
|---|---|
| Academia | 1.573 |
| Government | 1.538 |
| Industry | 1.463 |
| Nonprofit | 1.400 |

*Table 7.* Average number of social-impact dimensions evaluated per report by region

| region | average |
|---|---|
| East Asia | 1.361 |
| Europe | 1.445 |
| Middle East and North Africa | 1.593 |
| North America | 1.493 |
| South and Central Asia | 2.333 |

## G. Social Impact Categories

All social impact categories and their descriptions listed here are originally based on Solaiman et al. (2023).

**Bias, Stereotypes, and Representational Harms.**
*Description.* Generative AI systems often embed and amplify social biases originating from training data, optimization choices, and organizational practices. Bias can manifest at multiple stages of the model pipeline, from dataset curation to deployment, and may reinforce harmful stereotypes against marginalized groups (Solaiman et al., 2023). Together with minimization of the existence of a social group, the perpetuation of such stereotypes can contribute to representational harms, which is the misrepresentation of a group in a negative manner.
*Why is it important?* Minimizing bias, stereotypes, and representational harms is critical to ensuring that models are "fair" with equal outcomes for different groups.
*Why can/should it be evaluated at the base model level?* Biases baked into model parameters and design propagate throughout systems. Therefore, evaluating these outcomes at the base model level is critical for understanding biases in the final system (Solaiman et al., 2023).
*Example evaluation targets.* Common evaluations focus on harmful associations, co-occurrence analyses, and intrinsic versus extrinsic bias measures, though limitations arise due to evolving cultural definitions of protected categories and the difficulty of operationalizing intersectionality (Blodgett et al., 2020; Sun et al., 2019).

**Cultural Values and Sensitive Content.**
*Description.* Generative models inevitably reflect normative judgments about sensitive content and cultural values, which vary significantly across contexts.
*Why is it important?* Monitoring cultural values and sensitive content is critical to ensuring that models do not inflict harm upon users, help normalize harmful content, contribute to online radicalization, and aid in the production of harmful content for distribution (Solaiman et al., 2023). *Why can/should it be evaluated at the base model level?* Evaluating sensitive content and cultural values at the base level can help identify any cultural insensitivity or hate, toxicity, and targeted violence that a model might embody.
*Example evaluation targets.* Outputs may include hate speech, targeted violence, culturally offensive imagery (Solaiman et al., 2023) or cultural bias due to models (Yadav et al., 2025; **?**), with evaluations often drawing on frameworks such as the World Values Survey (**?**), Geert Hofstede's work on cultural values (Hofstede, 2011; 2001) or participatory approaches grounded in local norms (Bergman et al., 2024). Automated tools like toxicity classifiers are widely used but prone to over- or under-flagging, while human-led evaluations face scalability and psychological burden challenges (Gehman et al., 2020; Dinan et al., 2022).

**Disparate Performance.**
*Description.* Disparate performance occurs when AI models or systems produce systematically unequal results across subpopulations due to data sparsity, dataset skew, or modeling decisions (Solaiman et al., 2023). These disparities are distinct from representational harms, reflecting unequal outcomes rather than biased associations (Chien & Danks, 2024). Performance disparities are compounded by limited resources for lower-resourced languages and the infeasibility of exhaustively covering all possible subgroup intersections (Singh et al., 2024; Gohar & Cheng, 2023).
*Why is it important?* Disparate performance is important because it leads to unequal outcomes for different subpopulations.
*Why can/should it be evaluated at the base model level?* A critical challenge in evaluating disparate performance is the exponential number of subgroups and intersectionality (Solaiman et al., 2023). Evaluating disparate performance at the base level helps solve this issue by constraining the search to one part of the development chain.
*Example evaluation targets.* While evaluation methods vary across modalities, a common method involves evaluating model output across subpopulation languages, accents, and similar topics using the same evaluation criteria as the highest-performing language or accent. Metrics including subgroup accuracy, calibration, AUC, recall, precision, min-max ratios, and worst-case subgroup performance shed light on comparative performance (Gohar & Cheng, 2023).

**Environmental Costs and Carbon Emissions.**
*Description.* Training and deploying large-scale generative models require extensive compute resources, leading to significant but underreported carbon emissions (Strubell et al., 2020).
*Why is it important?* Environmental costs and carbon emissions are critical because they directly contribute to climate

change and resource depletion, raising concerns about the long-term sustainability of AI development.

*Why can/should it be evaluated at the base model level?* Current efforts explore both empirical reporting and life cycle assessments, but a lack of standardization, transparency from hardware vendors, and accounting for indirect impacts hinder comparability across studies (Strubell et al., 2020; Schwartz et al., 2020). Furthermore, the lack of consensus on what constitutes the total environment or carbon footprint of AI systems complicates this category because it introduces uncertainty about what variables to measure (Solaiman et al., 2023). Evaluating models at the base level is one avenue to help standardize measurements for comparison across models.

*Example evaluation targets.* Existing measurement tools such as CodeCarbon (Courty et al., 2024) and Carbontracker (Anthony et al., 2020) estimate emissions based on hardware, FLOPs, and runtime, though system-level and supply-chain impacts remain largely unquantified (Luccioni et al., 2023; Henderson et al., 2020).

### Privacy and Data Protection.

*Description.* Generative AI models raise privacy concerns through memorization of sensitive data, leakage of personally identifiable information (PII), and unauthorized reproduction of copyrighted content (Manduchi et al., 2024).

*Why is it important?* Privacy is important because it is a matter of contextual integrity (Solaiman et al., 2023). Data protection is critical because sensitive data could be leveraged for downstream harm, such as security breaches, privacy violations, and adversarial attacks (Solaiman et al., 2023).

*Why can/should it be evaluated at the base model level?* Classical privacy-preserving techniques such as differential privacy or data sanitization are difficult to adapt to generative settings (Carlini et al., 2021; Das et al., 2025). At the same time, evaluation privacy and data protection in generative settings become crucial because incentives for model performance can be at odds with privacy (Solaiman et al., 2023). Evaluating models at the base level offers a valuable opportunity to evaluate data leakages and identify privacy harms before deployment.

*Example evaluation targets.* Evaluations focus on measuring memorization, susceptibility to inference attacks, and data leakage in deployed systems (Shokri et al., 2017; Das et al., 2025).

### Financial Costs.

*Description.* Financial costs include the infrastructure, hardware costs, and hours of labor from researchers, developers, and crowd workers (Solaiman et al., 2023).

*Why is it important?* The high computational and infrastructure costs associated with training, fine-tuning, and serving generative models create barriers to entry and reinforce existing resource disparities between industry and academia. Access to sufficient compute is highly concentrated in a few organizations, limiting broad participation in research, deployment, and reproducibility (Ahmed & Wahed, 2020).

*Why can/should it be evaluated at the base model level?* Evaluating financial costs at the base model level is suitable because sourcing training data, compute infrastructure for training and testing systems, and labor hours are key contributors to overall financial costs (Solaiman et al., 2023).

*Example evaluation targets.* Evaluation of financial impacts includes tracking the direct costs of compute and storage, as well as economic barriers created by closed-access APIs and proprietary deployment pipelines (Machado, 2025).

### Data and Content Moderation Labor.

*Description.* Mitigating harmful outputs requires substantial human labor, including annotation, data filtering, and real-time moderation (Hao et al., 2023). These tasks are often outsourced to underpaid and precarious workers, exposing them to psychological harms and raising concerns about distributive justice (Spence et al., 2023; Gray & Suri, 2019).

*Why is it important?* Evaluations and transparent reporting can aid understanding model output and help audit labor practices (Solaiman et al., 2023).

*Why can/should it be evaluated at the base model level?* Evaluations at the base model level are particularly valuable since crowdwork is widely used in dataset development for generative AI systems (Solaiman et al., 2023).

*Example evaluation targets.* Evaluations should account for the demographics and working conditions of annotators, as moderation decisions directly shape which cultural perspectives are preserved or erased in training data (Denton et al., 2021; Röttger et al., 2021). Established standards for evaluation include the Criteria for Fairer Microwork. (Berg et al., 2018), the guidelines outlined in the Partnership on AI's Responsible Sourcing of Data Enrichment Services (Jindal, 2021), and the Oxford Internet Institute's Fairwork Principles (Fairwork).

# H. Scoring

## H.1. Criteria

Each report was annotated against the social impact dimensions using the following criteria:

- 0: No mention of the category, or only generic references without evaluation details.

- 1: Vague mention of evaluation (e.g., "We check for X" or "Our model can exhibit X").

- 2: Evaluation described with concrete information about methods or results (e.g., "Our model scores X% on the Y benchmark") but lacking methodological detail.

- 3: Evaluation methods described in sufficient detail to enable meaningful understanding and/or reproduction. Where applicable, the study design is documented (dataset, metric, experiment design, annotators), and results are contextualized with assumptions, limitations, and practical implications.

If a document deliberately disclosed the absence of an evaluation, it received a score of 0 for the corresponding category; we note that we did not encounter any such cases during dataset development.

For environmental and financial costs, the scoring scheme was adjusted as follows:

- 0: No reporting.

- 1: Same as above, or when reported technical details (e.g., FLOPs, GPU type, runtime) could indirectly be used to estimate costs.

- 2: Concrete values reported for a non-trivial part of model development or hosting, but derivation method unclear.

- 3: Concrete values reported together with contextual details and the derivation method.

For financial costs, we excluded first-party customer-facing pricing from consideration, as it reflects product strategy rather than system costs. Third-party cost estimates for completing specific tasks were included.

## H.2. Inter-Annotator Agreement

Annotations were performed by 16 individual volunteers with backgrounds in AI research, with manual spot checks for consistency. We assessed inter-annotator agreement on a subset of independently double-annotated data, with annotator pairs varying across items. We use Krippendorff's $\alpha$ with ordinal weighting, which supports ordinal labels and varying annotator sets. For the release-time reports, a random sample of 20 models were double-annotated across seven categories (140 items total), with Krippendorff's $\alpha = 0.75$. For the post-release reports, 250 items were randomly sampled without replacement for double annotation, with Krippendorff's $\alpha = 0.83$.

More granularly, the 390 (140 release-time, 250 post-release) paired item-category judgments show 333/390 (85.4%) exact agreement, 375/390 (96.2%) agreement within one point, 15/390 (3.8%) differing by more than one point, and 2/390 (0.5%) at the maximum three-point gap. A sensitivity analysis incorporating annotator-level random effects is reported in App. L.2.

# I. Evaluation Report Sources

**Deduplication.** We consolidated results from the same evaluation into single data points where possible. For example, a webpage, paper, and leaderboard presenting the same set of experiments would be merged into one instance. When a report explicitly references prior documentation (e.g., "we use exactly the data from X"), we merge the items and score on the union of reported information across sources. We note that underspecified reporting may prevent us from identifying duplicates, potentially inflating evaluation counts.

**Models.** We considered incrementally versioned models (e.g., `claude-3` vs. `claude-3.5`) as separate, independently annotated releases if they had their own release materials (web pages, technical reports, etc). When multiple models from the same family were released simultaneously (e.g., `llama-3` 8B, 70B, 405B), we treated them as a single release. We

aggregated all materials released at a model's launch into a single source, with the metadata's 'url' field containing multiple URLs when applicable.

**Documentation.** For documentation updates (e.g., model card addenda, peer-reviewed versions), we defaulted to the latest available version at the time of annotation, assuming that newer documents provide more complete information.

## J. Search terms

Search terms used per social impact dimension:

| Social Impact Category | Search Terms |
|---|---|
| Bias, stereotypes, and representational harm | bias, stereotype, representational harm |
| Sensitive content | cultural, culture, harm, toxic, sensitive |
| Disparate performance | disparate, fairness, equity |
| Environmental cost & carbon emission | carbon, CO2, environmental, energy |
| Privacy & data protection | privacy, copyright, data protection |
| Financial cost | cost, compute |
| Data and content moderation labor | moderation, labor |

*Table 8.* Search terms used per category

## K. Flow Diagram

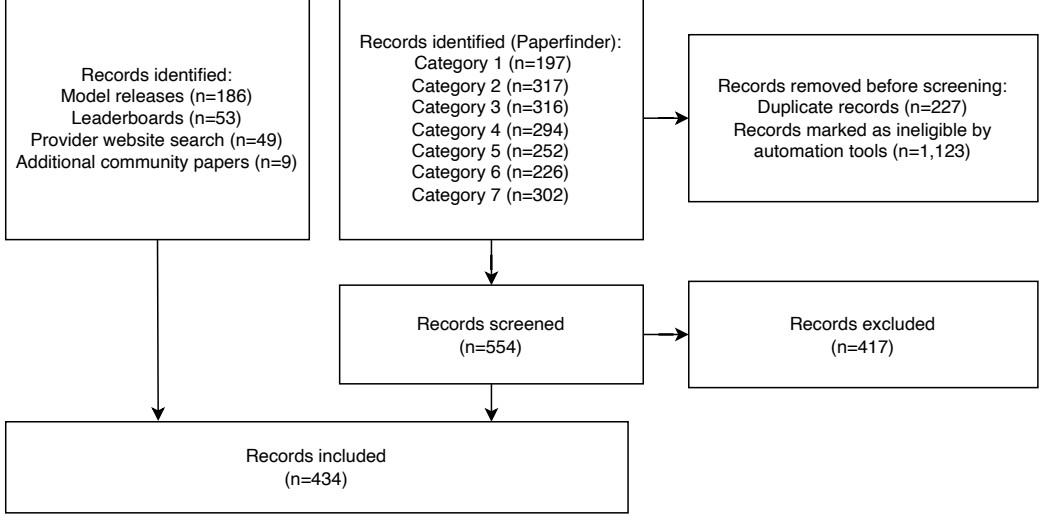

*Figure 28.* Evaluation reporting source inclusion process.

# L. Statistical Modelling

## L.1. Method

**Data, indexing, and zero–filling.** Observations are indexed by $i = 1, \ldots, N$. Each observation belongs to exactly one outcome (content category) $j(i) \in \{1, \ldots, J\}$ and records an ordinal response $y_i \in \{0, 1, \ldots, K-1\}$. The population-level effect design matrix $W_i \in \mathbb{R}^P$ include indicator variables for organization type, model openness and first-party status. and group indices are $r(i) \in \{1, \ldots, R\}$ (region), $c(i) \in \{1, \ldots, C\}$ (country), $p(i) \in \{1, \ldots, G\}$ (provider). Additionally, a standardized time covariate $t_i$ may be included for year effects.

**Likelihood: cumulative link (logit or probit).** For each outcome $j$, we introduce ordered cutpoints $-\infty = c_{j,0} < c_{j,1} < \cdots < c_{j,K-1} < c_{j,K} = +\infty$. Let $F$ be either the logistic CDF (logit link) or the standard normal CDF (probit). With linear predictor $\eta_i \in \mathbb{R}$, the cumulative and category probabilities are

$$\Pr(y_i \leq m \mid \eta_i) = F\big(\eta_i - c_{j(i),m}\big), \qquad m = 0, \ldots, K-1, \tag{1}$$

$$\Pr(y_i = k \mid \eta_i) = F\big(\eta_i - c_{j(i),k}\big) - F\big(\eta_i - c_{j(i),k-1}\big), \qquad k = 0, \ldots, K-1. \tag{2}$$

**Linear predictor: additive components.** The predictor combines nested intercepts, fixed slopes, optional group–specific slopes, first–party, and time effects:

$$\eta_i = \underbrace{\beta^{(\text{reg})}_{r(i),\,j(i)} + \beta^{(\text{cty})}_{c(i),\,j(i)} + \beta^{(\text{prov})}_{p(i),\,j(i)}}_{\text{hierarchical intercepts}}$$
$$+ \underbrace{W_i^\top \beta_{j(i)}}_{\text{population slopes}} + \underbrace{\sum_{g \in \mathcal{G}} W_i^\top s^{(g)}_{g(i),\,\cdot,\,j(i)}}_{\text{(optional) group–specific random slopes}}$$
$$+ \underbrace{g_{j(i)}(t_i)}_{\text{year effect}}. \tag{3}$$

Here $\mathcal{G} \subseteq \{\text{region}, \text{country}, \text{provider}\}$ selects which groups carry random slopes, and $W$ is the population-level design matrix with categorical predictors for model sector, model openness and first-party status.

**Nested deviations (sum–to–zero within parent).** To avoid location confounding across intercept layers, countries are modeled as deviations within regions and providers as deviations within countries. Let $\text{C2R} \in \{0, 1\}^{C \times R}$ (country–to–region map) and $\text{P2C} \in \{0, 1\}^{G \times C}$ (provider–to–country map). Let $n_r = \sum_c \text{C2R}_{cr}$ and $n_c = \sum_p \text{P2C}_{pc}$. Write the raw country and provider effects as $\tilde{\beta}^{(\text{cty})}, \tilde{\beta}^{(\text{prov})}$ and define

$$\bar{\beta}^{(\text{cty})}_{r,\cdot} = \frac{1}{n_r} \sum_{c:\text{C2R}_{cr}=1} \tilde{\beta}^{(\text{cty})}_{c,\cdot}, \qquad \beta^{(\text{cty})}_{c,\cdot} = \tilde{\beta}^{(\text{cty})}_{c,\cdot} - \sum_r \text{C2R}_{cr}\, \bar{\beta}^{(\text{cty})}_{r,\cdot}, \tag{4}$$

$$\bar{\beta}^{(\text{prov})}_{c,\cdot} = \frac{1}{n_c} \sum_{p:\text{P2C}_{pc}=1} \tilde{\beta}^{(\text{prov})}_{p,\cdot}, \qquad \beta^{(\text{prov})}_{p,\cdot} = \tilde{\beta}^{(\text{prov})}_{p,\cdot} - \sum_c \text{P2C}_{pc}\, \bar{\beta}^{(\text{prov})}_{c,\cdot}. \tag{5}$$

These constraints ensure each child layer sums to zero within its parent, anchoring the overall location alongside the cutpoints.

**Year effect.** Year effect is modelled as one of the following

$$\text{linear: } g_j(t) = \beta^{(\text{year})}_j\, t, \tag{6}$$

$$\text{spline: } g_j(t) = B(t)^\top w_j, \quad \text{with } B \text{ a B–spline basis}, \tag{7}$$

$$\text{Gaussian Processes (GP): } g_j(\cdot) \sim \mathcal{GP}\big(0,\, \sigma^2\, k_\ell(\cdot, \cdot)\big), \quad k_\ell(t, t') = \exp\Big(-\frac{(t-t')^2}{2\ell^2}\Big). \tag{8}$$

**Priors and parameterizations.** For block label $q \in \{\text{region, country, provider, first party, slopes, group-level slopes}\}$ and outcome $j$, the group-level intercepts are given by:

$$\text{Independent across outcomes: } b_{u,j}^{(q)} = z_{u,j}^{(q)} \sigma_j^{(q)}, \quad z_{u,j}^{(q)} \sim \mathcal{N}(0,1), \quad \sigma_j^{(q)} \sim \text{HalfNormal}(\sigma_q^{\text{base}}). \tag{9}$$

$$\text{Correlated across outcomes: } (b_{u,1}^{(q)}, \dots, b_{u,J}^{(q)}) = z_{u,:}^{(q)} L^{(q)\top}, \quad z_{u,:}^{(q)} \sim \mathcal{N}_J(0, I), \tag{10}$$

$$\Sigma^{(q)} = D^{(q)} R^{(q)} D^{(q)}, \quad R^{(q)} \sim \text{LKJ}(\eta), \tag{11}$$

$$D^{(q)} = \text{diag}(\sigma_1^{(q)}, \dots, \sigma_J^{(q)}), \tag{12}$$

where LKJ is the Lewandowski–Kurowicka–Joe distribution with parameter $\eta$ with density given by $\text{LKJ}(\Sigma|\eta) \propto |\Sigma|^{\eta-1}$. Group–level slopes $s^{(g)}$ follow the same scheme, either independent per outcome or correlated via a Cholesky factor. In that case, each predictor receives its own scale in $D^{(q)}$.

**Cutpoint (threshold) priors.** Two alternatives are implemented per outcome $j$:

$$\text{Dirichlet–spacings: } (\delta_{j,1}, \dots, \delta_{j,K-1}) \sim \text{Dirichlet}(\mathbf{1}) \tag{13}$$

$$\tilde{c}_{j,k} = K \sum_{\ell=1}^{k} \delta_{j,\ell}, \tag{14}$$

$$c_{j,k} = \begin{cases} \tilde{c}_{j,k}, & \text{if no location recentering,} \\ \tilde{c}_{j,k} - \dfrac{1}{K-1} \sum_{m=1}^{K-1} \tilde{c}_{j,m} + c_{0j}, & c_{0j} \sim \mathcal{N}(0,5). \end{cases} \tag{15}$$

$$\text{Ordered–normal: } u_{j,k} \overset{\text{iid}}{\sim} \mathcal{N}(0,1), (c_{j,1}, \dots, c_{j,K-1}) = \text{sort}(u_{j,1}, \dots, u_{j,K-1}). \tag{16}$$

**Inference.** We implemented our models in `PyMC5` (Abril-Pla et al., 2023). We used the No-U-Turn sampler (NUTS) for parameter inference and ran 4000 warm-up and 4000 post-warm-up draws for 8 randomly initialized Markov chains, with a target accept probability of acceptance of 0.99. To ensure the robustness and trustworthiness of the inference results, we ensured there were no divergences after the warm-up phase and that the split $\hat{R}$ as well as bulk- and tail-effective sample sizes (ESS) were satisfactory ($\hat{R} \leq 1.01$, ESS $\geq 1000$ for all variables of interest). In addition, we visually inspected the trace and pairs plots to ascertain the convergence of the chains.

## L.2. Regression results

Table 9 shows the population-level coefficients as the marginal log-odds-ratio (LOR) of having a higher or equal score. Table 10 further interprets and contextualize the findings.

| parameter, outcome index | median | mad | eti_2.5% | eti_97.5% | mcse_median | ess_median | ess_tail | r_hat |
|---|---|---|---|---|---|---|---|---|
| open, 1 | 0.216 | 0.179 | -0.288 | 0.748 | 0.003 | 19439.359 | 23439.575 | 1.000 |
| open, 2 | -0.279 | 0.121 | -0.633 | 0.070 | 0.001 | 25885.622 | 25156.097 | 1.000 |
| open, 3 | 0.035 | 0.217 | -0.598 | 0.663 | 0.002 | 31426.580 | 26214.423 | 1.000 |
| open, 4 | 0.606 | 0.231 | -0.065 | 1.280 | 0.002 | 30501.777 | 26471.000 | 1.000 |
| open, 5 | 0.580 | 0.266 | -0.166 | 1.375 | 0.003 | 27587.995 | 9628.835 | 1.000 |
| open, 6 | **0.673** | **0.177** | **0.164** | **1.200** | **0.002** | **34824.752** | **29220.662** | **1.000** |
| open, 7 | -0.430 | 0.371 | -1.741 | 0.260 | 0.013 | 3807.179 | 14202.032 | 1.001 |
| Academia, 1 | 0.143 | 0.354 | -0.906 | 1.163 | 0.004 | 22688.306 | 24765.087 | 1.000 |
| Academia, 2 | -0.112 | 0.331 | -1.087 | 0.852 | 0.005 | 16714.835 | 20332.249 | 1.000 |
| Academia, 3 | 0.873 | 0.488 | -0.545 | 2.366 | 0.005 | 29632.499 | 26273.259 | 1.000 |
| Academia, 4 | **1.348** | **0.458** | **0.069** | **2.742** | **0.005** | **26631.747** | **25522.698** | **1.000** |
| Academia, 5 | 0.143 | 0.512 | -1.472 | 1.603 | 0.005 | 36156.282 | 5727.676 | 1.000 |
| Academia, 6 | 0.595 | 0.384 | -0.494 | 1.796 | 0.004 | 37094.262 | 28003.017 | 1.000 |
| Academia, 7 | 0.069 | 0.308 | -1.127 | 1.646 | 0.005 | 11518.695 | 15189.796 | 1.002 |
| Government, 1 | 0.161 | 0.392 | -0.993 | 1.346 | 0.004 | 26025.643 | 12024.340 | 1.000 |
| Government, 2 | -0.057 | 0.347 | -1.067 | 0.977 | 0.005 | 18114.920 | 22616.057 | 1.000 |
| Government, 3 | 0.322 | 0.496 | -1.163 | 1.798 | 0.005 | 29679.750 | 27161.132 | 1.000 |
| Government, 4 | -0.244 | 0.460 | -1.629 | 1.097 | 0.004 | 31554.657 | 25912.808 | 1.000 |
| Government, 5 | 0.253 | 0.453 | -1.114 | 1.604 | 0.005 | 30727.664 | 26809.297 | 1.000 |
| Government, 6 | -0.389 | 0.373 | -1.590 | 0.668 | 0.004 | 39791.485 | 25882.720 | 1.000 |
| Government, 7 | -0.121 | 0.318 | -2.160 | 0.912 | 0.007 | 7745.362 | 12026.635 | 1.001 |
| Nonprofit, 1 | -0.580 | 0.389 | -1.740 | 0.570 | 0.005 | 21593.000 | 23081.296 | 1.000 |
| Nonprofit, 2 | -0.327 | 0.353 | -1.357 | 0.736 | 0.004 | 22189.057 | 22600.153 | 1.001 |
| Nonprofit, 3 | -1.079 | 0.549 | -2.814 | 0.459 | 0.007 | 24840.724 | 26314.535 | 1.000 |
| Nonprofit, 4 | **1.451** | **0.444** | **0.178** | **2.768** | **0.005** | **19455.891** | **21657.566** | **1.000** |
| Nonprofit, 5 | 0.339 | 0.473 | -1.071 | 1.734 | 0.004 | 31316.951 | 28687.365 | 1.000 |
| Nonprofit, 6 | 0.707 | 0.369 | -0.296 | 1.858 | 0.003 | 33992.869 | 27568.134 | 1.000 |
| Nonprofit, 7 | 0.180 | 0.322 | -0.790 | 1.825 | 0.009 | 5715.445 | 13630.120 | 1.001 |
| firstparty, 1 | **-3.637** | **0.160** | **-4.106** | **-3.183** | **0.002** | **32975.429** | **27925.873** | **1.000** |
| firstparty, 2 | **-3.929** | **0.117** | **-4.273** | **-3.592** | **0.002** | **14682.455** | **15255.373** | **1.001** |
| firstparty, 3 | **-4.660** | **0.221** | **-5.333** | **-4.042** | **0.002** | **34052.940** | **28307.369** | **1.000** |
| firstparty, 4 | **-2.915** | **0.204** | **-3.525** | **-2.325** | **0.003** | **22645.673** | **21300.907** | **1.000** |
| firstparty, 5 | **-4.546** | **0.241** | **-5.274** | **-3.877** | **0.003** | **32881.043** | **29239.312** | **1.000** |
| firstparty, 6 | **-2.220** | **0.163** | **-2.696** | **-1.754** | **0.002** | **31002.487** | **27856.559** | **1.000** |
| firstparty, 7 | -0.026 | 0.530 | -3.143 | 2.440 | 0.006 | 10296.945 | 4728.669 | 1.002 |

*Table 9.* Posterior of regression coefficients for the slopes of model openness, model sector: Industry (baseline), Academia, Government, Nonprofit, and first-party status. Median-ESS, Tail-ESS and $\hat{R}$ are included here to ascertain convergence and mixing of Markov chains. The outcomes are coded as above: 1 = **Bias, Stereotypes, and Representational Harms**, 2 = **Sensitive Content**, 3 = **Disparate Performance**, 4 = **Environmental Costs and Carbon Emissions**, 5 = **Privacy and Data Protection**, 6 = **Financial Costs**, 7 = **Data and Content Moderation Labor**. Rows in **boldface** indicates coefficients of which the 95% high-density interval (HDI) did not cover 0.

| Predictor | Category: Outcome Indices and Coefficients (Log-Odds Ratio) | What It Means | Interpretation |
|---|---|---|---|
| **Openness** | 6. Financial (+0.67); 4. Environmental (+0.61, with $P(\beta > 0) = 0.962$) | Openness is associated with more detailed discussion of environmental and financial issues | Open models tend to publish more on environmental and cost impacts, likely due to greater transparency norms. |
| **Academia** | 4. Environmental (+1.35); 3. Disparate Performance (+0.87, with $P(\beta > 0) = 0.888$) | Academic reports cover environmental topics and disparate performances more | Academic research discusses environmental aspects as well as disparate performances significantly more than industry, possibly due to the latter's primary emphasis on raw model performances. |
| **Government** | Small effect sizes with no statistically significant effects | Government reports do not differ significantly from industry baseline | Government AI documentation shows similar reporting patterns to industry across all categories. |
| **Nonprofit** | 4. Environmental (+1.45); 3. Disparate performances (-1.08, with $P(\beta < 0) = 0.916$); 6. Financial costs (0.707, with $P(\beta > 0) = 0.911$) | More coverage of environmental issues and financial costs but less of disparate performances | Nonprofits focus more on resource and sustainability reporting as well as financial costs, possibly driven by mission, but unexpectedly had inferior reporting on disparate performances compared to industry. |
| **First-party** | Strongly negative across 1–6 (-2.22 to -4.67 range) | Lower reporting across all categories except 7 | First-party (internal) model developers report significantly less detail on all impact categories than external or third-party reports. This is the most consistent and statistically robust finding. |

*Table 10.* Summary of regression findings by predictor. Positive coefficients indicate higher reporting detail (more thorough discussion) relative to the Industry baseline; negative coefficients indicate less reporting. Unless otherwise specified and indicated with $P(\beta > 0)$ or $P(\beta < 0)$, only statistically significant effects (95% credible intervals excluding zero) are reported.

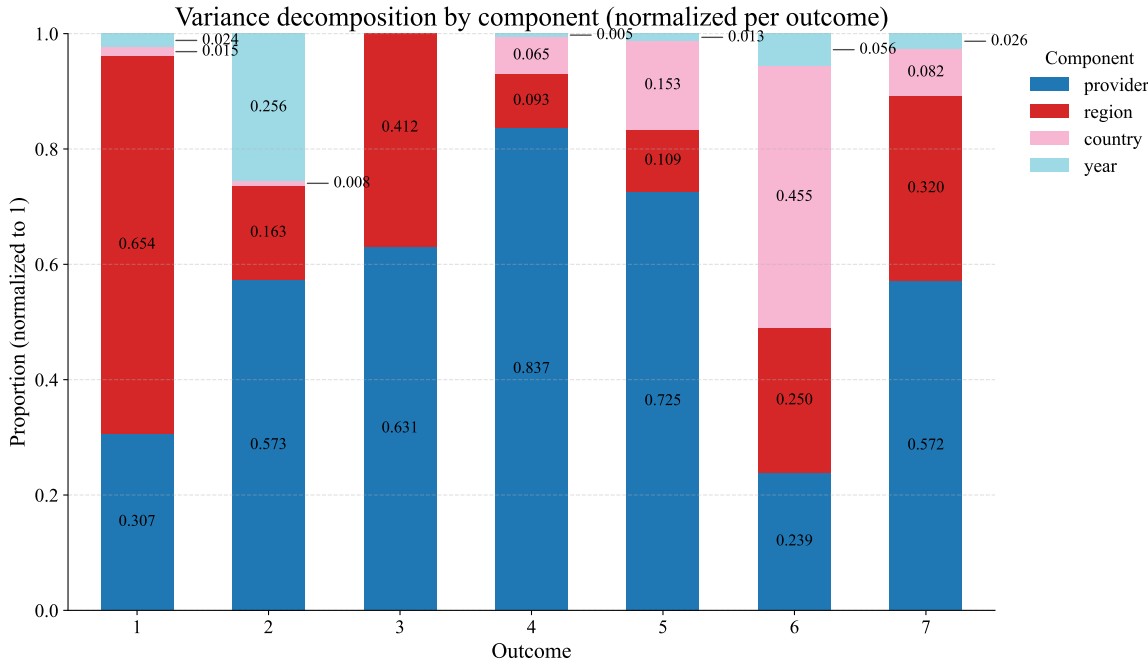

*Figure 29.* Stacked bar chart of variance decomposition by group-level effects contribution, shown per outcome and normalized to 1. The outcomes are coded the same way as Table 9.

| parameter, outcome index | median | mad | eti_2.5% | eti_97.5% | mcse_median | ess_median | ess_tail | r_hat |
|---|---|---|---|---|---|---|---|---|
| open, 1 | 0.070 | 0.139 | -0.334 | 0.473 | 0.002 | 29088.821 | 26402.142 | 1.000 |
| open, 2 | -0.301 | 0.103 | -0.597 | 0.004 | 0.001 | 32593.008 | 27408.939 | 1.000 |
| open, 3 | 0.046 | 0.163 | -0.422 | 0.521 | 0.002 | 39867.175 | 26867.487 | 1.000 |
| open, 4 | **0.653** | **0.186** | **0.119** | **1.208** | **0.002** | **37638.891** | **26552.444** | **1.000** |
| open, 5 | 0.077 | 0.185 | -0.453 | 0.620 | 0.002 | 31430.219 | 8240.110 | 1.000 |
| open, 6 | **0.539** | **0.149** | **0.111** | **0.990** | **0.001** | **38698.971** | **28719.112** | **1.000** |
| open, 7 | -0.036 | 0.078 | -0.608 | 0.213 | 0.002 | 4412.757 | 8434.566 | 1.005 |
| Academia, 1 | 0.212 | 0.237 | -0.477 | 0.925 | 0.002 | 32736.143 | 26837.396 | 1.000 |
| Academia, 2 | 0.087 | 0.216 | -0.555 | 0.722 | 0.002 | 29974.562 | 27727.523 | 1.000 |
| Academia, 3 | 0.525 | 0.298 | -0.338 | 1.418 | 0.003 | 39089.017 | 27151.841 | 1.000 |
| Academia, 4 | 0.736 | 0.284 | -0.058 | 1.626 | 0.002 | 38709.404 | 23988.524 | 1.000 |
| Academia, 5 | 0.250 | 0.307 | -0.681 | 1.165 | 0.003 | 39864.259 | 12440.207 | 1.000 |
| Academia, 6 | 0.385 | 0.249 | -0.338 | 1.171 | 0.002 | 40390.599 | 25867.033 | 1.000 |
| Academia, 7 | 0.002 | 0.075 | -0.403 | 0.451 | 0.001 | 26611.167 | 10254.552 | 1.004 |
| Government, 1 | 0.133 | 0.256 | -0.632 | 0.898 | 0.003 | 34049.420 | 26206.434 | 1.000 |
| Government, 2 | 0.106 | 0.231 | -0.559 | 0.794 | 0.003 | 29195.494 | 25518.770 | 1.000 |
| Government, 3 | 0.192 | 0.302 | -0.711 | 1.085 | 0.003 | 37584.055 | 26465.187 | 1.000 |
| Government, 4 | -0.170 | 0.293 | -1.060 | 0.674 | 0.003 | 39918.692 | 26919.371 | 1.000 |
| Government, 5 | 0.245 | 0.276 | -0.554 | 1.081 | 0.003 | 33012.207 | 28449.615 | 1.000 |
| Government, 6 | -0.170 | 0.244 | -0.950 | 0.525 | 0.002 | 37707.667 | 9302.795 | 1.000 |
| Government, 7 | -0.005 | 0.073 | -0.469 | 0.359 | 0.001 | 20083.521 | 12054.302 | 1.005 |
| Nonprofit, 1 | -0.464 | 0.270 | -1.268 | 0.297 | 0.004 | 28306.568 | 25772.794 | 1.000 |
| Nonprofit, 2 | -0.481 | 0.230 | -1.152 | 0.195 | 0.003 | 32115.287 | 28445.187 | 1.000 |
| Nonprofit, 3 | -0.865 | 0.324 | -1.869 | 0.043 | 0.004 | 34850.318 | 11631.371 | 1.000 |
| Nonprofit, 4 | **1.031** | **0.268** | **0.270** | **1.848** | **0.003** | **31662.638** | **25047.419** | **1.000** |
| Nonprofit, 5 | 0.002 | 0.287 | -0.840 | 0.851 | 0.003 | 37607.683 | 27526.503 | 1.000 |
| Nonprofit, 6 | 0.401 | 0.263 | -0.324 | 1.232 | 0.002 | 36368.602 | 26599.598 | 1.000 |
| Nonprofit, 7 | 0.005 | 0.086 | -0.417 | 0.530 | 0.001 | 9003.495 | 4450.557 | 1.006 |
| firstparty, 0 | **-3.542** | **0.147** | **-3.982** | **-3.122** | **0.002** | **32369.522** | **25323.386** | **1.000** |
| firstparty, 1 | **-3.840** | **0.114** | **-4.179** | **-3.515** | **0.002** | **21015.753** | **18768.978** | **1.000** |
| firstparty, 2 | **-4.423** | **0.196** | **-5.012** | **-3.873** | **0.002** | **34170.722** | **28898.677** | **1.000** |
| firstparty, 3 | **-2.497** | **0.188** | **-3.050** | **-1.973** | **0.002** | **28423.274** | **27583.769** | **1.000** |
| firstparty, 4 | **-4.085** | **0.206** | **-4.699** | **-3.505** | **0.002** | **36045.486** | **28595.013** | **1.000** |
| firstparty, 5 | **-2.120** | **0.161** | **-2.586** | **-1.656** | **0.002** | **32246.882** | **28527.835** | **1.000** |
| firstparty, 6 | 0.002 | 0.174 | -0.994 | 0.965 | 0.002 | 4883.703 | 1259.121 | 1.010 |

*Table 11.* Posterior of regression coefficients for the slopes of model openness, model sector: Industry (baseline), Academia, Government, Nonprofit, and first-party status. Median-ESS, Tail-ESS and $\hat{R}$ are included here to ascertain convergence and mixing of Markov chains. The outcomes are coded as above: $1$ = **Bias, Stereotypes, and Representational Harms**, $2$ = **Sensitive Content**, $3$ = **Disparate Performance**, $4$ = **Environmental Costs and Carbon Emissions**, $5$ = **Privacy and Data Protection**, $6$ = **Financial Costs**, $7$ = **Data and Content Moderation Labor**. Rows in **boldface** indicates coefficients of which the $95\%$ high-density interval (HDI) did not cover $0$. *Compared to the model in 9, this model has tighter priors.*

| parameter, outcome index | median | mad | eti_2.5% | eti_97.5% | mcse_median | ess_median | ess_tail | r_hat |
|---|---|---|---|---|---|---|---|---|
| open, 1 | 0.286 | 0.184 | -0.239 | 0.842 | 0.002 | 21925.473 | 21459.662 | 1.000 |
| open, 2 | -0.171 | 0.143 | -0.571 | 0.283 | 0.002 | 21733.839 | 20190.150 | 1.000 |
| open, 3 | 0.058 | 0.220 | -0.581 | 0.720 | 0.003 | 26618.616 | 25747.079 | 1.000 |
| open, 4 | 0.552 | 0.244 | -0.148 | 1.276 | 0.003 | 26532.079 | 25199.284 | 1.000 |
| open, 5 | 0.577 | 0.259 | -0.168 | 1.363 | 0.003 | 26815.545 | 25954.837 | 1.000 |
| open, 6 | **0.679** | **0.184** | **0.152** | **1.225** | **0.002** | **33412.021** | **26914.057** | **1.000** |
| open, 7 | -0.408 | 0.369 | -1.746 | 0.258 | 0.015 | 2278.717 | 9915.222 | 1.005 |
| Academia, 1 | 0.045 | 0.380 | -1.090 | 1.159 | 0.005 | 22513.649 | 24086.488 | 1.000 |
| Academia, 2 | -0.176 | 0.363 | -1.244 | 0.886 | 0.005 | 16586.630 | 18024.427 | 1.000 |
| Academia, 3 | 0.858 | 0.503 | -0.590 | 2.363 | 0.005 | 29896.191 | 26721.009 | 1.000 |
| Academia, 4 | **1.362** | **0.464** | **0.067** | **2.784** | **0.005** | **26535.831** | **25553.189** | **1.000** |
| Academia, 5 | 0.163 | 0.515 | -1.450 | 1.642 | 0.005 | 36833.601 | 27300.199 | 1.000 |
| Academia, 6 | 0.578 | 0.387 | -0.519 | 1.801 | 0.004 | 37077.886 | 28156.218 | 1.000 |
| Academia, 7 | 0.073 | 0.310 | -1.153 | 1.670 | 0.006 | 7177.801 | 7412.849 | 1.005 |
| Government, 1 | 0.253 | 0.434 | -1.001 | 1.530 | 0.005 | 24435.607 | 23848.745 | 1.000 |
| Government, 2 | -0.003 | 0.377 | -1.086 | 1.139 | 0.005 | 18852.375 | 20919.115 | 1.001 |
| Government, 3 | 0.304 | 0.507 | -1.185 | 1.804 | 0.005 | 26872.850 | 24537.270 | 1.000 |
| Government, 4 | -0.185 | 0.480 | -1.673 | 1.227 | 0.005 | 31045.554 | 27300.329 | 1.000 |
| Government, 5 | 0.272 | 0.470 | -1.135 | 1.662 | 0.005 | 29220.796 | 26639.858 | 1.000 |
| Government, 6 | -0.362 | 0.385 | -1.574 | 0.721 | 0.004 | 39191.201 | 26997.683 | 1.000 |
| Government, 7 | -0.115 | 0.313 | -2.189 | 0.927 | 0.009 | 5006.406 | 7824.674 | 1.005 |
| Nonprofit, 1 | -0.506 | 0.424 | -1.764 | 0.757 | 0.005 | 24520.255 | 23464.709 | 1.000 |
| Nonprofit, 2 | -0.424 | 0.386 | -1.544 | 0.738 | 0.005 | 20613.673 | 22295.366 | 1.000 |
| Nonprofit, 3 | -1.077 | 0.551 | -2.793 | 0.465 | 0.007 | 21492.479 | 23877.365 | 1.000 |
| Nonprofit, 4 | **1.345** | **0.458** | **0.049** | **2.737** | **0.006** | **19720.395** | **21303.270** | **1.000** |
| Nonprofit, 5 | 0.360 | 0.493 | -1.066 | 1.813 | 0.005 | 30003.653 | 27625.282 | 1.000 |
| Nonprofit, 6 | 0.672 | 0.384 | -0.393 | 1.850 | 0.004 | 33665.087 | 27745.467 | 1.000 |
| Nonprofit, 7 | 0.172 | 0.321 | -0.810 | 1.875 | 0.012 | 3097.243 | 7764.813 | 1.003 |
| firstparty, 1 | **-4.350** | **0.238** | **-5.087** | **-3.693** | **0.003** | **26622.387** | **25355.192** | **1.000** |
| firstparty, 2 | **-4.680** | **0.199** | **-5.276** | **-4.119** | **0.003** | **14783.254** | **21586.757** | **1.000** |
| firstparty, 3 | **-4.745** | **0.234** | **-5.470** | **-4.102** | **0.002** | **31342.353** | **28175.662** | **1.000** |
| firstparty, 4 | **-3.250** | **0.261** | **-4.016** | **-2.507** | **0.003** | **20471.732** | **24453.276** | **1.000** |
| firstparty, 5 | **-4.761** | **0.273** | **-5.592** | **-4.019** | **0.003** | **27457.304** | **26895.861** | **1.000** |
| firstparty, 6 | **-2.180** | **0.177** | **-2.691** | **-1.650** | **0.002** | **31586.729** | **28324.682** | **1.000** |
| firstparty, 7 | -0.036 | 0.553 | -3.361 | 2.521 | 0.009 | 4089.831 | 959.679 | 1.011 |

*Table 12.* Posterior of regression coefficients of the model for the slopes of model openness, model sector: Industry (baseline), Academia, Government, Nonprofit, and first-party status. Median-ESS, Tail-ESS and $\hat{R}$ are included here to ascertain convergence and mixing of Markov chains. The outcomes are coded as above: 1 = **Bias, Stereotypes, and Representational Harms**, 2 = **Sensitive Content**, 3 = **Disparate Performance**, 4 = **Environmental Costs and Carbon Emissions**, 5 = **Privacy and Data Protection**, 6 = **Financial Costs**, 7 = **Data and Content Moderation Labor**. Rows in **boldface** indicates coefficients of which the 95% high-density interval (HDI) did not cover 0. *Compared to the model in 9, this model has the same hierarchical priors but with provider-level slopes added.*

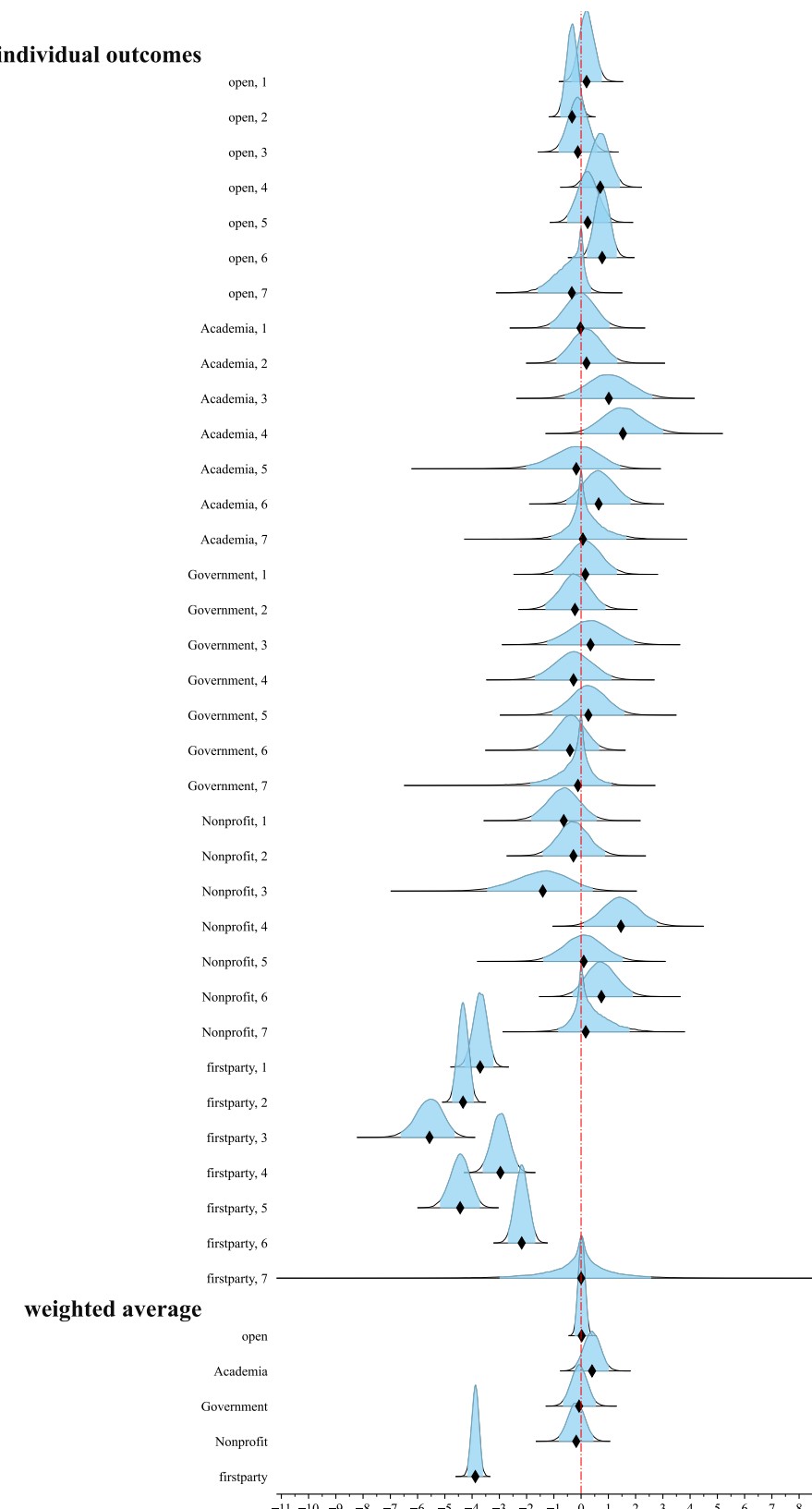

*Figure 30.* 95% high density interval (HDI) of the posterior distribution of the regression coefficients of the slopes. Black parallelograms indicate the posterior medians. The outcomes are coded as above: 1 = **Bias, Stereotypes, and Representational Harms**, 2 = **Sensitive Content**, 3 = **Disparate Performance**, 4 = **Environmental Costs and Carbon Emissions**, 5 = **Privacy and Data Protection**, 6 = **Financial Costs**, 7 = **Data and Content Moderation Labor**

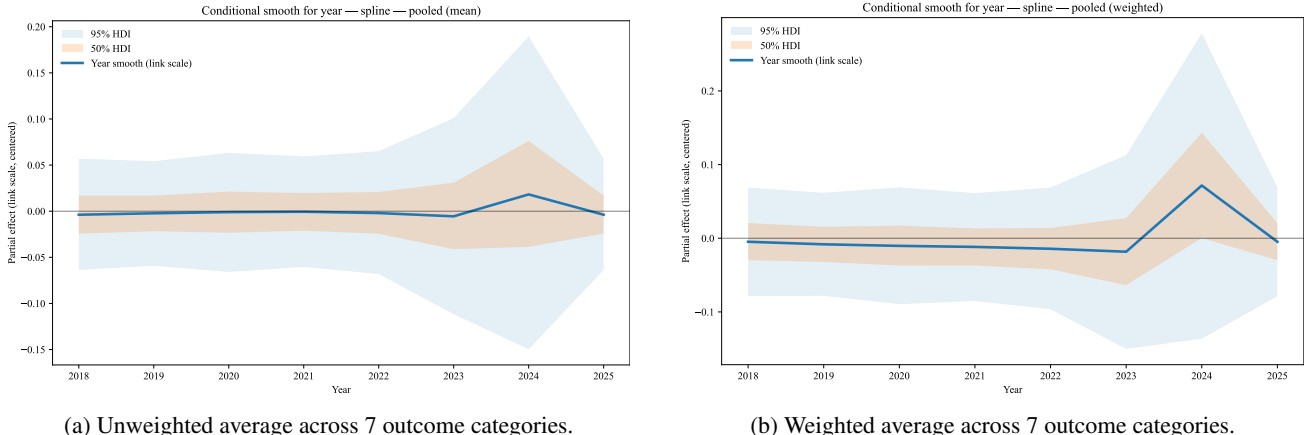

(a) Unweighted average across 7 outcome categories.

(b) Weighted average across 7 outcome categories.

*Figure 31.* Standardized effect of year on the linear predictor on the logistic link scale.

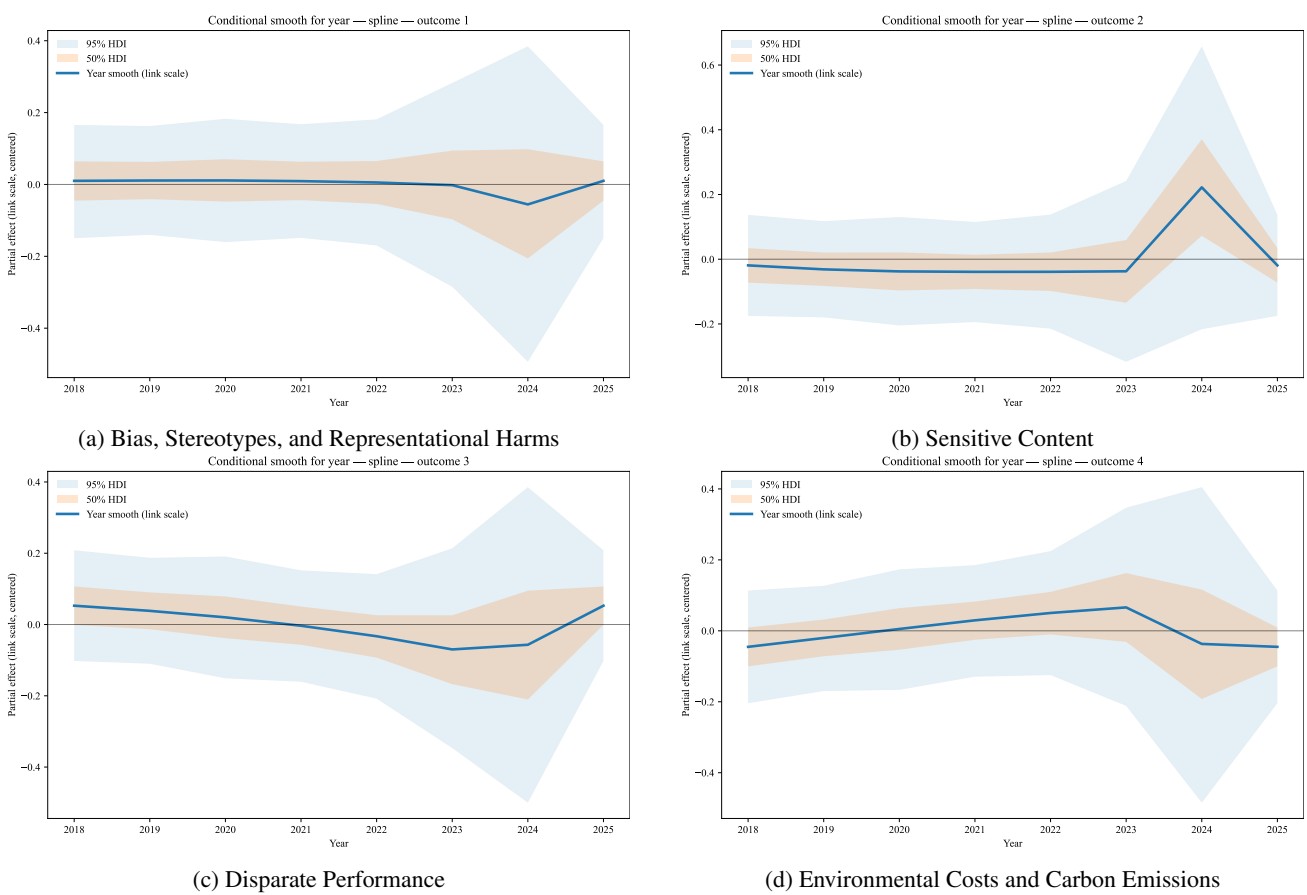

(a) Bias, Stereotypes, and Representational Harms

(b) Sensitive Content

(c) Disparate Performance

(d) Environmental Costs and Carbon Emissions

*Figure 32.* Standardized effect of year on the linear predictor by outcome category. Each panel shows posterior mean trends with uncertainty intervals across time for the seven annotated social impact outcomes. *(continued on next page)*

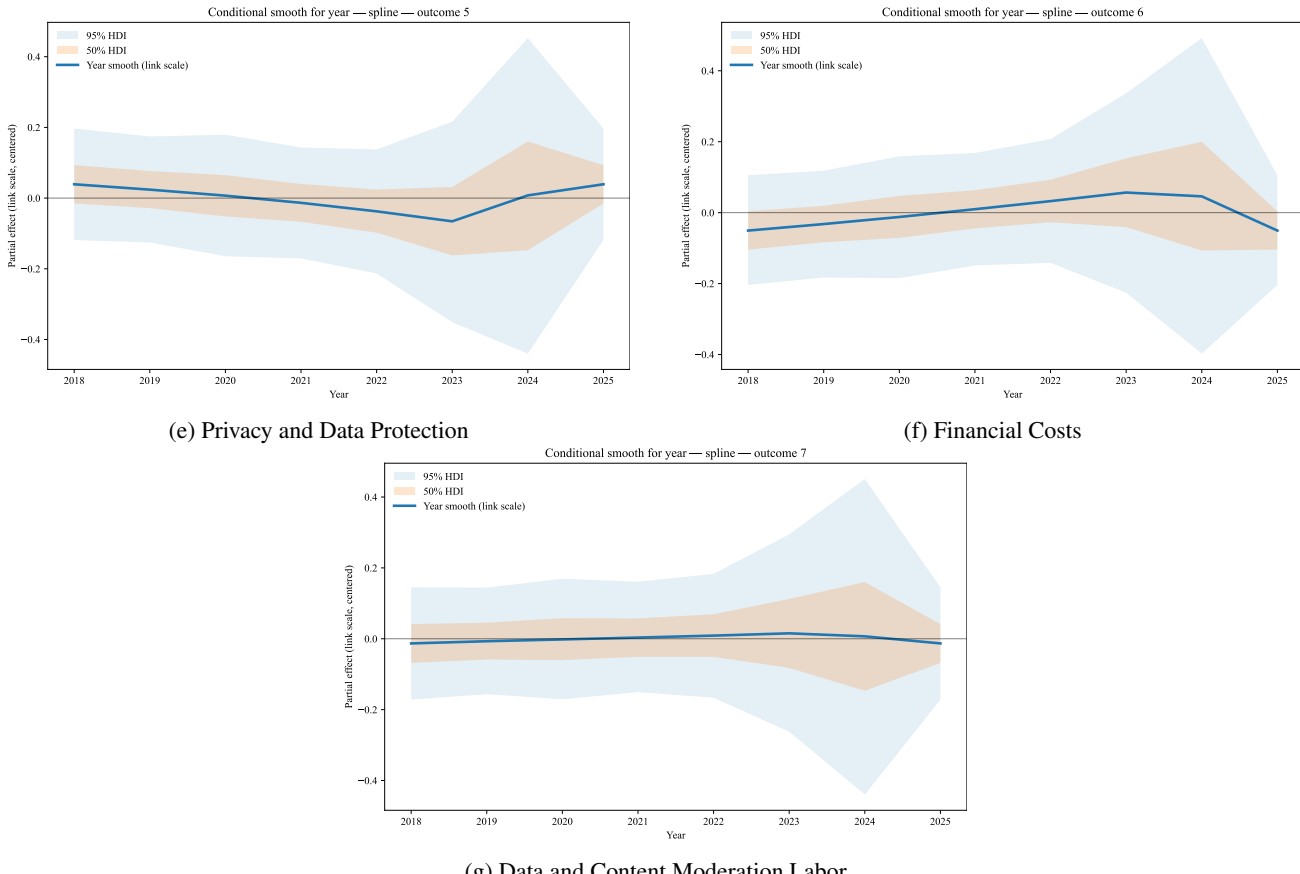

(e) Privacy and Data Protection

(f) Financial Costs

(g) Data and Content Moderation Labor

*Figure 32.* Standardized effect of year on the linear predictor by outcome category *(continued from previous page).*

**Sensitivity to annotator-level effects.**   As a robustness check on annotation disagreement, we fitted two sensitivity variants of our main hierarchical model (cf. Tab. 9): the first added category-specific annotator-level intercepts $u_{a_i,j_i}$ to the linear predictor (Eq. 3), while the second included an additional term $v_{m_i}$, where $m_i$ is the (doubly-coded) item-category unit to which rating $i$ belongs and $u$ and $v$ following normal distributions with 0 mean and independent variances. In both models, the first-party coefficient remains negative in every category, with posterior probability $P(\beta_{\text{first-party}} < 0) = 1.0$ in six of seven categories and 0.98 for category 7. Model comparison by leave-one-out cross validation (LOO-CV) strongly favoured the second model (ELPD diff $= 88.32$, SE $= 10.07$). In this richer model, item-level variation (SD) exceeds annotator-level in 5 of 7 categories by large margins, in particular for sensitive content: $1.50\,[1.02, 1.98]$ vs $0.31\,[0.02, 0.84]$, suggesting that a substantial share of disagreement is item-specific rather than being driven solely by annotator severity.

## M. Post-release first-party evaluation

In Section 5 (Figure 3), we presented first-party evaluations reported at model release, which account for the majority of first-party evaluation instances in our dataset (1302 instances, 87%). The remaining first-party evaluations are reported post-release (191 instances, 13%). These data let us study time-lagged evaluation reporting practices, although we note that their sparsity and small sample size limit our analysis to preliminary observations.

From the available data, we observe that post-hoc first-party reports tend to be more comprehensive than those at initial release. For example, `gpt-3.5` was released via the OpenAI API with limited publicity and no accompanying reports, but first-party fairness evaluation results became available 1.5 years post-release. Similarly, environmental cost information for `gemini-2.0` and `mistral-large` became available approximately 0.5 to 1.5 years after their respective releases.

Delayed post-release evaluations, while still valuable, are less useful in practice. By the time results become available, models may already have proliferated, preventing stakeholders from making informed decisions about model selection. These issues are exacerbated by the predominantly static and fragmented nature of current evaluation reporting: understanding a model's risk profile requires searching and resolving results from disparate sources over time. Developing standardized, continuously updated reporting frameworks would address these limitations, enabling informed model choices and facilitating analysis of evaluation and reporting practices beyond the time-frame addressed in this paper.

## N. Interviews

### N.1. Interview Methodology

We conducted generative user interviews to understand the wider motivations held by practitioners in conducting social impact evaluations. The interviews were structured around four thematic areas:

1. Reactions to observed trends in social impact evaluation reporting as discussed in Section 5.

2. Incentives and motivations for conducting social impact evaluations and reporting results.

3. Barriers to running social impact evaluations and reporting results.

4. Ideal future directions in social impact evaluations.

The specific questions asked to our interviewees were designed to support the following key objectives:

1. Uncover qualitative insights regarding the quantitative results reported in Section 5.

2. Understand practitioner viewpoints on why social impact evaluations are lacking.

3. Identify incentives to encourage increased social impact evaluation reporting.

Interviewees were recruited based on their experience conducting or developing evaluations. These participants were sourced through the authors' collective professional contacts and subsequent snowball sampling. All participating interviewees (see Appendix N.2) had direct experiences concerning social impact evaluations. The interviews were conducted remotely in September and October 2025. Quotes used throughout the text were selected from interview transcripts by one of the authors to contextualize quantitative findings.

**N.2. Interviewees**

Appendix N.2 contains the list of interviewees, their organization type, and their geographic location. Demographic information is not shared to maintain the anonymity of interviewees.

| Interviewee Tag | Stakeholder Group | Geographic Location |
|---|---|---|
| NP1 | works at a non-profit building LLMs and running evaluations | US |
| NP2 | works at a non-profit developing evaluations and on model governance | US |
| FP1 | works at a for-profit company on model design | US |
| FP2 | works at a for-profit company evaluating models and developing evaluations | France |
| FP3 | works at a for-profit company on evaluation | US |
| FP4 | works at a for-profit company on developing evaluations | Canada |
| FP5 | works at a for-profit startup on model evaluation, general model governance and acceptable use | US |
| A1 | is a PhD student developing evaluations | UK |
| A2 | works at a university on developing and running evaluations as well as model governance | US |
| CS1 | works at a civil society organization developing and running evaluations | US |

*Table 13.* Interviewee list where each interviewee is assigned a tag based on their employer where NP refers to non-profit; FP refers to for-profit; A refers to academia; and CS refers to civil society. Location is reported as well where US refers to United States and UK refers to United Kingdom.

**N.3. Interview Guide**

The following subsection includes the interview guide that the interviewers followed in all interviews.

**Goal.** We are focusing on understanding what sorts of evaluations you are motivated to conduct, what prevents you from doing so, and your general attitudes and thoughts about evaluations.

We want to read your unfiltered thoughts, so please do not hesitate to provide details or point us to other lines of thought if you think they are important. Feel free to let us know if you are not able to or permitted to answer specific questions.

**Outcome.** At the end of all interviews, we hope to synthesize broad patterns as a research output.

**Confidentiality Notice.** The responses you provide and any notes from this interview will be kept confidential and will be limited to internal use. For external use, such as writing a report or research paper, your information will be anonymized. References to your answers or quotes will not include identifying details. Please let us know if this presents any concerns.

**Interview Questions.**

*Qualifier Questions.*

1. With respect to conducting and reporting evaluations, what are relevant responsibilities and job tasks that fall within your scope of responsibility?

2. Does your role involve one or more of the following: model specification design, model development, deployment, model evaluation, general model governance, and/or acceptable use?

3. Have you been directly involved in designing, implementing, reviewing, or reporting evaluations in the past 2–3 years? (Or have you had visibility into the decision processes leading up to these activities?)

*Incentives/Motivations for Reporting.*

1. Why do you think the observed trends (demonstrated in the plots developed by team #4) in reporting social impact evaluations are occurring?
   *Interviewer note: Show the plots prior to asking this question.*

2. What currently motivates, if anything, your organization (or you) to conduct evaluations?

3. Are there any incentives that would encourage you or your organization to conduct more thorough evaluations for social impact?

*Notes: Possible incentives include positive media attention, regulation, higher internal capacity, or knowledge that customers/users care about specific evaluations.*

*Barriers to Adoption. This is the central portion of the interview. Please dedicate the majority of time to this section.*

1. Given the following Likert scale, where would you rank each of these categories in terms of importance and feasibility?

    (a) 1 = Very easy / very feasible
    (b) 2 = Easy / feasible
    (c) 3 = Somewhat easy / somewhat feasible
    (d) 4 = Neutral
    (e) 5 = Somewhat hard / somewhat infeasible
    (f) 6 = Hard / infeasible
    (g) 7 = Very hard / very infeasible

    *Interviewer note: Pay attention to categories high in importance but low in feasibility.*

2. In our analysis of evaluations reported across social impact categories, we observed missing evaluations in certain areas (e.g., ____). What barriers do you face to reporting these categories (e.g., time, legal risk, reputational/investor harm, resources)?

3. Which is the most pressing barrier?

4. Which factors or underlying reasons need to change to enable greater reporting?

5. What would incentivize your organization to report more in this dimension?

6. How does granularity and specificity impact feasibility with regard to running social impact evaluations?

*Future Directions.*

1. What does the ideal social impact evaluation look like in terms of breadth and specificity?

2. Can you provide an example?

3. Are there any social impact categories missing that should be reported?

4. Who should be responsible for evaluations?

5. Who should be responsible for developing, running, or setting standards for evaluations?

6. To what extent would a standardized reporting template help enable more comprehensive social impact evaluation reporting?
   *Interviewer note: Show the current evaluation card design.*

7. (Optional) If you had a "magic wand," what would the ideal process/tool look like for reporting social impact evaluations?

*Current Practices (Organizational). This section can be skipped if not relevant.*

1. Who is ultimately responsible for conducting social impact evaluations in your organization?

2. Which social impact evaluations are conducted? What criteria/processes shape the choice, and who are the stakeholders?

3. What criteria/processes exist for deciding which evaluation results get publicly reported?

4. Describe the types of social impact evaluations, if any, that are reported when documenting AI systems.

5. Based on your most recent model/system card, we found these social impact evaluations: [share screen]. Are there evaluations conducted internally that were not reported? If so, which, and why not?

6. Are social impact evaluations broad or granular (e.g., examining bias in specific contexts)?

7. For the social impact evaluations displayed, to what extent are they context-specific vs. use-case-independent?

8. How do you select or design evaluations? To what extent do you include/prioritize use-case-specific vs. general/agnostic tasks?

9. To what extent do you rely on third-party evaluators or contractors?

10. To what extent do internal evaluations differ from those released externally? If differences exist, what practices/criteria determine release (e.g., legal, PR)?

*Closing Questions.*

1. Is there any question we should have asked, or anything with respect to social impact evaluations that we have not covered yet?

2. Is there anything else you think we should know before we end this call?

## O. Code and Dataset

Our analysis code and annotated dataset are accessible at:

- Code: https://github.com/evaleval/social_impact_eval_annotations_code

- Dataset: https://huggingface.co/datasets/evaleval/social_impact_eval_annotations

## P. Author Contribution Statement

| | |
|---|---|
| CONCEPTUALIZATION | A. Ghosh, A. Reuel, J. Chim, Y. Jernite, I. Solaiman |
| DATA CURATION | A. Ghosh, A. Reuel, J. Chim, P. S. Ammanamanchi, A. Tran, J. Batzner, Y. Long, O. Beyer Bruvik, S. Yadav, H. A. Rahmani, A. Kornilova |
| INVESTIGATION | A. Ghosh, A. Reuel, J. Chim, P. S. Ammanamanchi, M. Allaham, A. Tran, J. Mickel, F. Friedrich, J. Batzner, H. A. Rahmani, Y. Long, S. Sahoo, K. Wei, M. A. Riegler, J. Mickel, J. Sania, A. Saxena, E. Habba, U. Gohar, R. Scholz |
| METHODOLOGY | A. Ghosh, A. Reuel, J. Chim, J. Mickel, U. Gohar, J. Sania, A. Saxena, J. Batzner |
| SOFTWARE | A. Ghosh, A. Reuel, J. Chim,, F. Friedrich, P. Sadeghi, M. Allaham, J. Batzner, A. Tran, Y. Long |
| VALIDATION | A. Ghosh, A. Reuel, J. Chim, P. S. Ammanamanchi, M. Allaham, A. Tran, F. Friedrich, Y. Long, S. Sahoo, K. Wei |
| FORMAL ANALYSIS | A. Ghosh, A. Reuel, A. Tran, Y. Long, J. Chim |
| WRITING (ORIGINAL DRAFT) | A. Ghosh, A. Reuel, J. Chim, S. Yadav, U. Gohar, J. Mickel, P. Soni, K. Klyman, A. Tran, S. Sahoo, M. Akhtar, H. A. Rahmani |
| WRITING (REVIEW & EDITING) | A. Ghosh, A. Reuel, S. Yadav, H. A. Rahmani, J. Chim, P. S. Ammanamanchi, M. Allaham, A. Tran, U. Gohar, J. Mickel, M. Akhtar, F. Friedrich, A. Wang, J. Batzner, E. Habba, A. Saxena, K. Wei, P. Soni, Y. Mathew, K. Klyman, J. Sania, S. Sahoo, Y. Long, A. Kornilova, S. Goswami, R. Scholz, I. Solaiman, Z. Talat, M. Kochenderfer, S. Koyejo |
| VISUALIZATION | A. Ghosh, A. Reuel, J. Chim, A. Tran, Y. Long, K. Klyman, J. Batzner, J. Mickel |
| SUPERVISION | A. Ghosh, A. Reuel, M. Kochenderfer, S. Koyejo, I. Solaiman, Z. Talat, S. Biderman |

