# OpenReview forum: "Who Evaluates AI's Social Impacts? Mapping Coverage and Gaps in First and Third Party Evaluations"
_ICML.cc/2026/Conference — ICML 2026 regular_

### Official Review · Reviewer_aUgh · 2026-03-12

**Soundness:** 2
**Presentation:** 2
**Significance:** 2
**Originality:** 2
**Overall Recommendation:** 2
**Confidence:** 4

**Summary:**

This paper presents a large-scale empirical analysis of social impact evaluation reporting for foundation models, examining 186 first-party release reports and 248 third-party evaluation sources across seven dimensions (bias, sensitive content, performance disparity, environmental costs, privacy, financial costs, and data/content moderation labor). The authors employ a 0-3 scoring scheme to quantify reporting detail, supplement their quantitative analysis with 10 semi-structured stakeholder interviews, and fit Bayesian hierarchical ordinal regression models to estimate effects of openness, sector, region, and first-party status on reporting quality. The central claims are: (1) first-party reporting is substantially less detailed than third-party reporting across nearly all dimensions, (2) reporting on environmental costs and bias has declined over time, (3) data/content moderation labor is almost entirely unreported, and (4) misaligned incentives and structural barriers explain these gaps.

**Compliance With Llm Reviewing Policy:**

Affirmed.

**Key Questions For Authors:**

1) How were disagreements between annotators resolved? Was there a consensus protocol, or did a single adjudicator decide? Given α = 0.75, roughly 25% of items had meaningful disagreement. How sensitive are your main findings (particularly the first-party vs. third-party gap) to these contested annotations?
2) Can you provide the within-provider temporal analysis (analogous to Fig. 4) for all 16 stratified providers, not just Google and Meta? The two shown both happen to support the decline narrative. Is this representative, or were these selected because they best illustrate the trend?
3) The first-party vs. third-party comparison conflates document genre with reporting effort. Have you considered restricting the third-party sample to non-peer-reviewed sources (e.g., blog posts, informal audits) to make the comparison more apples-to-apples? What happens to the gap magnitude?
4) Your regression shows first-party status has a strongly negative coefficient (LOR −2.2 to −4.7), but this is partially tautological: third-party academic papers are scored higher because academic papers include methodological detail by convention. How do you disentangle the reporting-norms effect from a genuine transparency deficit?

**Limitations:**

The authors include a limitations section (Sec. 7) acknowledging the small interview sample, geographic bias, and the fact that they assess reporting presence rather than evaluation quality. However, they do not adequately address the scoring reliability concern (moderate α with unsystematic spot checks), the compositional confound in the temporal analysis, or the genre-based ceiling effect in third-party scores. The potential negative impact of publicly ranking providers on incomplete data, which could incentivize superficial compliance gaming, is mentioned only briefly in the impact statement but deserves more prominence given it directly undermines the paper's own goals.

**Strengths And Weaknesses:**

Strengths:
1) The scale of the annotation effort is notable. 186 first-party reports and 248 third-party sources, coded across 7 dimensions with a documented scoring rubric. This is the first systematic attempt to quantify the first-party vs. third-party reporting gap at this granularity, going meaningfully beyond aggregate indices like FMTI.
2) The mixed-methods design combining quantitative scoring with semi-structured interviews adds interpretive value. Interview quotes contextualize why certain categories (e.g., environmental costs) show temporal decline, grounding the statistical patterns in organizational realities.
3) The Bayesian hierarchical ordinal regression is a methodologically appropriate choice given the nested structure (provider within country within region) and ordinal response variable. The authors report convergence diagnostics (R-hat, ESS, divergences) transparently.

Weaknesses:
1) The 0-3 scoring scheme is the backbone of the entire paper, yet the annotation process raises concerns. Annotations were conducted by 16 volunteers with only "manual spot checks" that were "not systematically applied across all samples." Inter-annotator agreement (Krippendorff's α = 0.75 for release-time, 0.83 for post-release) is reported on subsets, but 0.75 is only moderate for ordinal data, and the subset sizes (20 models × 7 categories = 140 items; 250 items) are small relative to the full dataset. The paper does not report how disagreements were resolved, whether consensus rounds occurred, or how annotator drift was managed across 16 coders.
2) The central construct is reporting detail, not evaluation rigor, validity, or coverage adequacy. Yet the paper repeatedly interprets higher scores as evidence of stronger evaluation practice. This is a construct-validity problem, not merely a transparency caveat. The authors acknowledge this but do not adequately grapple with its implications for their normative claims. A company that runs extensive internal bias evaluations but does not publish them receives a score of 0, identical to a company that does nothing. The policy recommendations (mandating transparency) implicitly assume reporting tracks effort, which is unvalidated.
3) Interview sample is too small and non-representative for the weight placed on it. Ten interviews, predominantly from US/Europe-based organizations, are used throughout the results section to explain quantitative trends. Snowball sampling from the authors' professional contacts introduces selection bias. The paper leans heavily on individual quotes (FP1, FP2, etc.) to support causal claims about organizational incentives, but N=10 cannot support such generalizations. The interviews function as illustrative anecdotes rather than systematic qualitative evidence.
3) The decline in environmental and bias reporting over time (Fig. 3) could reflect compositional shifts in which providers released models in later quarters (e.g., more closed Chinese models entering the dataset in 2024-2025) rather than within-provider declines. The within-provider analysis (Fig. 4) is shown for only two providers (Google, Meta). The regression includes year as a B-spline but does not adequately control for the changing provider composition over time.
4) Third-party evaluations average 2.64 on a 0-3 scale. Given that these are peer-reviewed papers and structured leaderboards, scoring near ceiling is expected by construction, since peer review enforces methodological detail. The first-party vs. third-party gap is thus partly an artifact of comparing documentation genres (system cards vs. academic papers) with inherently different reporting norms. This is not adequately discussed.
5)  What would "adequate" reporting look like? The paper scores detail level but never establishes what a score of 2 vs. 3 means in terms of practical utility for downstream decision-makers. Without this, it is unclear whether the gap between first- and third-party reporting matters for the stated goals (informed adoption decisions, regulatory oversight).
6) Category 7 (moderation labor) is virtually empty in both first- and third-party data, yielding poorly identified regression coefficients (acknowledged by the authors via funnel-shaped posteriors). Yet the paper makes strong claims about this category. The regression results for category 7 should be interpreted with extreme caution given the effective sample size issues.
7) The same findings (first-party < third-party, environmental reporting declining, moderation labor missing) are stated in the abstract, introduction, results, and discussion with near-identical phrasing. The paper could be tightened.

---

> ### Author Rebuttal · Authors · 2026-03-31
>
> **Limitations**
> - **_L1_** We address scoring reliability, compositional confounds, and genre effects below. We will discuss the risk of compliance gaming in the camera-ready.
>
> **Key Questions**
> - **_Q1_** We will clarify the annotation protocol in the camera-ready: in doubly annotated instances, annotators discussed disagreements, with a lead author adjudicating if consensus could not be reached. We note that computing IAA on a randomly sampled subset is standard practice in studies with large datasets. In addition, our reported alpha values use ordinal weighting, which penalizes larger disagreements more heavily than adjacent ones, so the effective reliability for our main contrasts is higher than the alpha suggests. We also note that a big share of annotations involve scores of 0, where disagreement is rare since the absence of information is unambiguous. There are 390 paired item-category judgments, only 15 differ by more than one point and only 2 show the maximum 3-point gap. As a robustness check on disagreement we also fitted 2 sensitivity variants of our main hierarchical model: the first added category-specific annotator-level intercepts to the linear predictor (Eqs. 3-5), while the second included an additional $v_{m_i}$, where $m_i$ is the (doubly-coded) item-category unit to which rating $i$ belongs and $u$ and $v$ following normal distributions with $0$ mean and ind. variances. In both, the 1st party coefficient remains negative in every category, with P(beta < 0) = 1.0 in six of seven categories and 0.98 for category 7. Model comparison by leave-one-out cross validation strongly favoured the 2nd model. In this richer model, item-level variation exceeds annotator-level in 5 out of 7 categories by large margins, suggesting that a substantial share of disagreement is item-specific rather than being driven solely by annotator severity.
> - **_Q2_** We will include per-provider temporal breakdowns for all 16 stratified providers in the camera-ready. Google and Meta were selected because they had among the longest release histories in our dataset.
> - **_Q3 and Q4_** We acknowledge genre partly explains the gap and will state this explicitly. Two points limit the concern: 1) our 3rd party sources include 53 leaderboards and 9 informal evaluation releases that do not follow peer review norms, yet 3rd party scores remain substantially higher than 1st party across these sources. 2) Our primary finding rests on the 1st party data alone: the regression coefficient (LOR −2.22 to −4.66) is estimated with 95% CI well clear of zero across 6 categories. Even if 3rd party scores are inflated by genre norms, the finding is not fragile to moderate inflation given these effect sizes. That said, we agree that a an analysis limiting 3rd party sources to non-peer-reviewed materials would be informative and will report this in the camera-ready.
>
> **Weaknesses**
> - **_W1_** (see Q1)
> - **_W2_** This limitation is acknowledged in Sec. 8. Our construct is disclosure practice, not internal evaluation rigor, and we will revise any language in the manuscript that conflates the two. The reviewer is correct that a company with strong internal evaluations that does not disclose receives a score of 0, identically to one that does nothing. This is the correct behavior for our construct: from the perspective of any external stakeholder the two cases are identical in consequence.
> - **_W3_** See response to 9bdA.
> - **_W4_** We fitted additional models where the time trend was explicitly decomposed into additive intra- vs inter-provider effects. The results strongly refuted the compositional hypothesis: intra-provider effects remained robustly negative while all inter-provider effects had 95% CI covering 0.
> - **_W5_** (see Q3/Q4)
> - **_W6_** We aim to characterize current reporting practices and identify structural gaps rather than to establish normative adequacy thresholds. Yet, the practical consequence of the gap is clear at the extremes: a score of 0 provides no information to any decision-maker, and the high 1st party missingness rates are notable regardless of where one sets the adequacy bar. We note that adequacy of reporting would differ depending on the regulatory and deployment context. We do agree that this is a key caveat and will add it to the discussion.
> - **_W7_** We will add explicit language stating they should be interpreted descriptively only. The substantive finding (near-zero reporting) is established by the raw missingness proportions alone and requires no regression support. We will remove inferential language specific to Category 7 coefficients.
> - **_W8_** We will tighten the writing in the camera-ready.
>
> **We sincerely thank the reviewer for their thorough review of our paper and their extensive discussion of improvement suggestions. We know that going to this level of detail takes a lot of time and effort, and we appreciate that the reviewer helped us significantly improve our research. Thank you so much!**

---

> > ### Author Rebuttal · Reviewer_aUgh · 2026-04-04
> >
> > I thank the authors for their detailed and constructive rebuttal.
> >
> > Several responses meaningfully strengthen my confidence in the core findings, while a few concerns remain partially unresolved.
> > Points that improved my assessment:
> > The robustness analysis for Q1 is the strongest part of the rebuttal. Fitting sensitivity variants with annotator-level intercepts and item-level random effects, and showing that P(β < 0) ≥ 0.98 across all categories, directly addresses my concern about scoring reliability. The detail that only 15/390 paired judgments differ by more than one point is reassuring and should appear in the main text.
> > The intra- vs. inter-provider decomposition (W4) is also compelling. If the temporal decline is driven by within-provider changes rather than compositional shifts, this substantially strengthens the claim that reporting is deteriorating through strategic deprioritization rather than dataset artifacts. I encourage the authors to include this analysis prominently, not just as a rebuttal point.
> > Remaining concerns:
> > The response to Q3/Q4 is reasonable but incomplete. The authors note that 53 leaderboards and 9 informal sources still score higher than first-party reports, which is helpful. However, leaderboards are themselves structured evaluation instruments with methodological detail baked into their design, so this does not fully resolve the genre concern. I accept the authors' point that the finding primarily rests on first-party data alone and its high missingness rates, which are indeed informative regardless of the third-party comparison. I would recommend the authors reframe the contribution accordingly: the story about first-party inadequacy stands on its own and does not require the third-party contrast to be compelling.
> > On W2, the clarification that the construct is disclosure practice rather than evaluation rigor is appreciated, but the policy recommendations in Sections 6-7 implicitly assume that mandating disclosure will improve actual evaluation practice. This inferential leap should be acknowledged more carefully, since organizations could satisfy disclosure requirements with superficial reporting (precisely the compliance-gaming risk the authors mention only briefly).
> > Several key improvements (full provider temporal breakdowns, restricted third-party comparisons, tightened writing) are promised for camera-ready but not yet demonstrated. I take these commitments in good faith but note they are unverified.
> >
> > All and all, I am inclined to keep my score.
> >
> > All the best

---

> > > ### Author Response · Authors · 2026-04-04
> > >
> > > Dear Reviewer aUgh,
> > >
> > > Thank you for your time and thoughtful re-evaluation. We are glad that the robustness analysis and intra-vs-inter-provider decomposition meaningfully improved confidence in the core findings.
> > >
> > > We note that the rebuttal format presents a hard 5000-character limit and does not allow us to modify the paper itself. We can only provide clarifications and additional analyses, which we believe we have done for the main methodological questions raised. Given your characterization of these improvements, we had hoped this would be reflected in a score revision, and we would be grateful if you could reconsider the score. We once again affirm our commitment to incorporating all suggested revisions in the camera-ready version.
> > >
> > > Thank you again for your engagement with our work.

---

### Official Review · Reviewer_wTSE · 2026-03-13

**Soundness:** 4
**Presentation:** 3
**Significance:** 4
**Originality:** 3
**Overall Recommendation:** 5
**Confidence:** 4

**Summary:**

This work analyses social impact evaluations of large ML models, which have often been overlooked in model evaluations and benchmarks. The paper reviews 100s of first-party and third-party evaluation sources and interviews with developers, finding that third-party evals tend to go deeper into bias and environmental costs compared to superficial first-party evals. The paper draws attention to the need to strengthen policy, measurement standards, and create infrastructure for more efficient evaluations (including by third parties).

**Compliance With Llm Reviewing Policy:**

Affirmed.

**Ethical Review Concerns:**

As discussed in my W1, it was not made clear that this work, which involves interviews, has passed an ethics review.

**Ethical Review Flag:**

Flag this paper for an ethics review.

**Ethics Expertise Needed:**

["Responsible Research Practice (e.g., IRB, documentation, research ethics)"]

**Final Justification:**

In the rebutal, it was made clear that the paper has had proper IRB permission. The authors have accepted other points raised in the review as limitations of their work. All and all I think this is a good contribution and should be accepted.

**Key Questions For Authors:**

- Q1. Please clarify the ethics review situation.

- Q2. From what I understand, there was no social impact dimension of data ownership. Is this because you limited your taxonomy to that of Solaiman et al. 2023, and they do not have that?

- Q3. Could you also add a positionality statement to the paper? It would have been helpful to know your general background.

- Q4. Why do you use PaperFinder? Why not citation and reverse citation searches? Do you think the use of such AI search tools may have introduced any biases?

- Q5. Can you clarify what you mean by spot-checking the results?

**Limitations:**

Yes

**Strengths And Weaknesses:**

Overall, this is a great contribution. I will list my (S)trenghs, and (W)eaknesses.

- **S1. Timely and necessary.**  It surfaces key limitations regarding the social impact evaluations, discusses their development over the years, and demonstrates how shifting priorities and misaligned incentives have affected our evaluations in ways that bode ill for our ability to detect, regulate, and avoid societal harm from AI-enabled systems.

- **S2. Good statistical methodology.** The overall survey methodology is solid. I am not an expert on survey design, but the statistical modeling in Section K using a Bayesian hierarchical model and the resulting Markov chain simulations all made sense, and it is a breath of fresh air to see, given the prevalence of non-robust safety evals these days.

- **S3. Great scientific writing.** Front to back, this was well-written. Thorough details on various aspects of the survey are provided in the main matter: from question design, scoring roles, and interview discussions.

- **S4. Useful summary discussions.** Especially for the policy makers, I think having clear writing, such as Section 6, is essential. Granted, a number of results, such as strategic deprioritization (especially over time), are expected and known from anecdotal evidence. But hard data and studies such as this are necessary to make the case for collaborative regulation, agreements on standards, the necessity of good, high-quality frameworks that simplify compliance, etc.

- **W1. Some ethical concerns must be clarified.** I will raise these in their appropriate place, but from reading the paper, it is rather unclear if the study passed an ethics review board or if the interviewees were paid. I think this should have been disclosed, even if details are withheld to avoid breaking the double-blind.

- **W2. Important nuance may be lost in reducing social impact categories.** Many of these social impact dimensions may be hard to validate using benchmarks. For example, privacy and fairness concerns could be addressed if the training algorithm is transparent (and we have proof that it was used). A model (basically) cannot be privacy-preserving if it was not trained with differential privacy, and all data collected from humans is personally-identifiable. Every model discussed in this paper is completely unprivate.  To be clear, authors do a good job of presenting these categories and what they mean by them thoroughly in Appendix E, but I think some of this section should have been surfaced in the main matter more prominently. Hopefully, in a table or diagram with concrete examples. Being a privacy practitioner, I found it unhelpful that I had to dig through Page 25 to figure out what you meant by this. Similarly, I was not sure what was meant by Sensitive Content. Clear examples go a long way to clarify this and should be prominently featured in the main matter

- **W3. "Open" models is over-statement for the models studied.** Truly Open models should have open training sets and algorithms (e.g., Apertus, the Swiss LLM), so none of these models count as fully open. Consider adding an open/closed option that is partially open.

- **W4. Temporal tables (3-4) should have been plots and other visualization issues.** These tables are not very readable. Also, given the nature of this study, I should raise the accessibility challenge of reading all tables, especially given how important the cell color information is to establish an overall picture. Consider moving away from just tables. Why not show some marginal distributions? Some violin plots? It will certainly make the results stand out better. Also, Figure 3 is referenced in the text to reinforce the claim that reporting decreased over time. It's hard to validate this claim from the table; instead, replace it with a plot showing a rolling average of the scores for each of the 7 categories. I also cannot, by inspection, see why Q3 in 2023 is significant.

---

> ### Author Rebuttal · Authors · 2026-03-31
>
> **Strengths:**
>
> We thank the reviewer for their thorough and constructive evaluation. We are glad the reviewer finds our work timely and necessary in surfacing key limitations of social impact evaluations and how misaligned incentives have undermined our ability to detect and regulate societal harm from AI systems (S1), and appreciates our statistical methodology, particularly the Bayesian hierarchical modeling (S2). We are pleased the reviewer considers the paper well-written with thorough methodological detail (S3), and values our summary discussions for policymakers (S4), recognizing that while findings like strategic deprioritization may align with anecdotal evidence, systematic empirical studies are essential to support collaborative regulation and high-quality compliance frameworks.
>
>
> **Key Questions**
>
> - **_Q1_** We confirm we obtained an IRB approval for this study. Its details were omitted from the submission to preserve double-blindness, but we will include them in the camera-ready. We apologize for not stating this explicitly at submission time. Interviewees were not paid, we will also clarify this in the final writeup.
>
> - **_Q2_** Yes, we opted to follow an existing taxonomy and adopted that of Solaiman et al. (2023) given it is widely cited in the field. We recognize that other frameworks incorporating dimensions such as data ownership can be equally appropriate and can complement this work.
>
> - **_Q3_** We omitted a positionality statement for anonymity (e.g., as per guidance in the field [1]). We will include one in the camera-ready version.
>
> - **_Q4_** PaperFinder was used due to project resource constraints. As an LLM-powered search tool, PaperFinder’s query formulation, retrieval, and relevance assessments are mediated by model inference, which may introduce biases in surfaced results. For example, retrieval may favor highly-cited papers with more candidate snippets and publications that  have full text available in its open-access corpus, which could disadvantage newer or less accessible work including sources beyond mainstream papers. We recognize that complementary approaches such as citation and reverse citation searches could have improved coverage, and will more clearly state this in the limitations.
>
> [1]  https://facctconference.org/2026/cfp.html
>
>
> **Weaknesses**
>
> - **_W1_** (see response to Q1 above)
>
> - **_W2_** We agree that the social impact categories, along with examples, should be presented more prominently in the main paper. While they were moved to the appendix for space constraints, we will use the extra page in the camera-ready to add a summary table with examples in the main body. We also agree that many social impact dimensions are not easily captured by evaluations, and that, as mentioned by the reviewer, some such as privacy depend on process-level guarantees while others like environmental costs rely on disclosing infrastructure; these are often reliant on first-party reporting as they require information inaccessible to third-party evaluators (cf. Sec 6). Our scoring covers these disclosures as well as evaluation methods beyond benchmarks, for example red teaming and qualitative user studies.
>
> - **_W3_**  In this work, "open" refers to models where "weights [are] publicly available or accessible with minimal license restrictions". We use this definition as it captures the most salient distinction for our analysis: whether a model's weights are accessible to third-party model evaluators. While we note that some models in our dataset are compliant with the OSI's definition (e.g., Flan-T5, Pythia, Olmo, SmolLM), we acknowledge that this does not capture stricter definitions of openness (e.g., availability of training data and algorithms) and will clarify this explicitly in the revision.
>
> - **_W4_**  We appreciate the reviewer raising these points on accessibility and acknowledge that the reliance on tables and cell colors pose readability challenges. Our formatting choices were partly driven by space constraints. We do however note that these plotting choices do not change the overall picture, such as the consistently low reporting on categories like moderation labor. In the revision, we will add distribution plots in the appendix. We will also revise the framing of Q3 2023 to clarify that the decline in bias and environmental costs reporting is supported by the regression analysis (Fig. 16).
>
>
> **We want to thank the reviewer for their thoroughness, both in outlining the strengths and potential improvement areas of our paper. Your comments will help us significantly improve our paper, thank you so much!**

---

> > ### Author Rebuttal · Reviewer_wTSE · 2026-04-01
> >
> > Thank you for the rebuttal.
> >
> > Thanks for confirming the IRB review. Please address the requirments of the ethics review posted recently. I agree with every request made.
> >
> > I am a bit confused about your response to Q4. How is reverse-citation share more resource costly than PaperFinder? Shouldn't something like `serpapi.com/google-scholar-api` be more cost effective, and also allow you to apply a transparent aggregation mechanism?
> >
> > In response to [W3] I think leaving out the truely open LLMs (like the Swiss LLM) is detrimenttal to exhustiveness of the taxonomization.
> >
> > All and all, I am inclined to keep my score.
> >
> > All the best

---

> > > ### Author Response · Authors · 2026-04-04
> > >
> > > Dear Reviewer wTSE,
> > >
> > > We appreciate your raising this and helping us more carefully articulate our methodological choices.
> > >
> > > On W3: we acknowledge that our described method may miss some models (for instance Swiss LLM), particularly if they were released close to the cutoff date (Oct 2025) by developers without earlier models in the dataset. We will note this more clearly in the methodology.
> > >
> > > On Q4: we chose Paperfinder as it is a free, open tool designed for research use. We have considered alternatives; however, SerpAPI for example is a paid commercial service that scrapes Google Scholar, which is against the latter’s terms of service. Aside from financial costs of running these queries, Paperfinder also returns a more targeted set of semantically relevant results (e.g. papers about bias evaluation, rather than papers that contain keywords evaluation and bias without reporting novel evaluation themselves), which reduces the volume of manual work required in subsequent screening and annotation.
> > >
> > > We agree that reverse citation offers a more transparent aggregation mechanism, and that complementary citation-based searches could have improved coverage. We will note these limitations in the revision.
> > >
> > > Once again, thank you for helping us improve this paper and we appreciate your confidence and positive comments.

---

### Official Review · Reviewer_9bdA · 2026-03-13

**Soundness:** 4
**Presentation:** 3
**Significance:** 4
**Originality:** 4
**Overall Recommendation:** 5
**Confidence:** 4

**Summary:**

The authors analyse how an AI’s social impact evaluation depends on who evaluates it - specifically, they compare first-party to different third-party evaluations, focusing on seven different dimensions, ranging from bias and harm, to performance disparity, to environmental impacts. To this end, the authors identify different evaluation reports of AI models from FHugging face, leadership boards, or technical reports while utilizing other resources like Paperfinder. After identification they categorized the models by based on available models, provider organization type or provider geographic origin. Then, 16 experts score to which extend each reports addresses each of the seven dimensions. Based on the results, a regression analysis to identifies systematic influence factors of the party type on the extent of the evolution per dimensions. Additionally, authors complemented these results with ten expert interviews. The final synthesis of the results shows intriguing differences between the type of evaluation party – specifically that first-party evaluation is generally less elaborate in all but one dimension (see Table 8). Additionally, authors find temporal difference in how first-party player evaluate social impact and thereby identify a drift in focus and commitment in the evolution of social impacts. For example, focus on bias decreased over time for google and meta (see Fig. 4)

**Compliance With Llm Reviewing Policy:**

Affirmed.

**Final Justification:**

I remain convinced that this paper addresses a timely and important topic. It will spark valuable discussion by drawing attention to who evaluates AI systems and how these evaluations engage with social dimensions, a view also shared by my fellow ICML reviewers.

I agree with reviewer aUgh that the result may be partly tautological, insofar as third-party academic papers tend to score higher because academic work conventionally includes greater methodological detail. However, I do not see this as a weakness per se. On the contrary, even if the finding is partly tautological, it still supports the paper’s main implication: first-party evaluations should either be held to the same transparency and methodological standards or be complemented by explicit third-party evaluations conducted to academic standards.

**Key Questions For Authors:**

- What are the precise takeaways for legislation?
- What was the methodological foundation for the work at hand?
- What are the more general emerging themes and aggregated dimensions?

**Limitations:**

I find that the authors do put a lot of emphasis on quotes from the interviews. While they address that the sample is rather small, they do not further comment on potential issues (e.g., selection bias) as we, for example, do not know whether all for-profit interviewees come from the same company and have also no additional information about the expertise (e.g., in years of working experience) of the interviewees.

**Strengths And Weaknesses:**

Strengths:
- I really appreciate the authors emphasis on the quantitative analysis of (e.g., testing different priors for the regression in Table 9 and 10). It increase the robustness of the results substantially.
- The additional visual representations of variance composition (Figure 13) and the interval of the posterior distribution of regression coefficients add additional transparency of the results.
- The provided source code. The provision of the approach for identifying models and evaluation reports (Fig. 12) as well as providing the interview guide and interviewee overview improve transparency and reproducibility.
- As far as I can judge, the specification of the regression model is sound. But please keep in mind, that this is not my specific area of expertise so I might have missed nuanced but important aspects.
- I find that the provided overview within the main parts of the paper provide a quick and easy way to grasp the general finding of the research
- In general, I find the paper well structured, the order and content Chapters and Sub-Chapters make sense and allow thready to easily follow the general though of the authors
- In my personal view, the paper addresses a highly relevant issue. That there seems to be a need to ensure that the evaluation of social impacts of AI models is not a nice-to-have if companies feel like it, but a must-have that must be “forced-onto” companies through proper legislations and accompanying frameworks. Therefore, it can provide a first starting point for creating an understanding of the problem to then creat solutions. The qualitative results provide some insights into possible reasoning but due to the small sample size I see the significance mostly in the mpirically demonstrated negletance of first-party evalautions of social impact factors and also showcases how the efforts drift over time creating at least a starting point for connecting legislation and how AI safety and responsible Ai and word like bias are seen in the political world and how this effects how companies evalute models
- As already touched on above, I find the insights very interesting and they highlight a central and important issue on our way towards effective AI regulations and where we could start: legistation to ensure a more elaborate evlaution of the social impact dimensions


Weaknesses:
- While the quantitative analysis is only complementary to the quantitative results, I would urge authors to utilize established methods and workflows for analysing and presenting quantitative results - like from literature reviews or semi-structured interview. The authors might find inspiration in the works of Wolfswinkel et al. (2013) or Gioia et al. (2012).
- Based on the coding results, authors could have identified emerging concepts, themes and aggregated dimensions. Providing an overviews of such themes and possible sentiments of the 10 interviewed experts could have strengthened the results and especially discussion in which quotes were used to explain found effects. In the current form, unfortunately, it is hard to completely judge whether the presented quotes where cherry-picked or if there were indeed common, emerging themes across interviewees.
- As there is so much information in this paper and a lot of buried in the Appendix, I would urge the authors so streamline the Appendix: For example, the heading of the Appendix J is disconnected from it content. Directly after the heading of Appendix J comes the heading of Appendix K. to clarify presentation and enable readers to more easily capture the vast information provide in the appendix authors should really focus on a clear appendix structure.
- Additionally, I would suggest that the authors add an additional average aggregation in their overview figures (e.g., Fig. 1) that shows the average coverage not only per provider but also across the social impact dimensions, as this connects better to the overall topic of the research, as the average score per provider.
- I do understand that the authors do not want to be too bold with their call for actions, but with such results, I would have hoped for a more concrete plan and/or call for action- especially with regard to necessary legislation.
- On the other hand, I do find some framing of potential reasons for companies not evaluating certain dimensions as too strong for such a small qualitative data set.
- I find the discussion really focused on trying to identify reasons from the interviews. Yet, I am missing a more detailed embedding with other literature from responsible AI literature and also more focus on how we can move towards ensuring that first-party evaluations are also thorough and holistic. Or draw on literature from organizational or political science to explain the seen systematic effects and behavior or on frameworks like https://nvlpubs.nist.gov/nistpubs/ai/nist.ai.100-1.pdf

Misc:

Other Papers that the authors might find interesting:
https://jolt.law.harvard.edu/assets/articlePDFs/v35/Selbst-An-Institutional-View-of-Algorithmic-Impact-Assessments.pdf
https://dl.acm.org/doi/10.1145/3442188.3445935

Final Remark:

A final remark on my side: I do come from a more socio-technological background which push me a bit more towards the underlying organisational and political mechanisms and theories as it might be expected for a more technical outlet as the ICML. Nevertheless, I find that the paper is methodologically sound and appropriate, and offers interesting and significant results that will spark discussions and might inform future research in the direction of legislations and frameworks to support companies to evaluate their model holistically while also holding them accountable.

I wish the authors all the best with this and their future research, congratulate them on a well-done research project, and hope they find my comments helpful.



Wolfswinkel, J. F., Furtmueller, E., & Wilderom, C. P. M. (2013). Using grounded theory as a method for rigorously reviewing literature. European Journal of Information Systems, 22(1), 45–55. https://doi.org/10.1057/ejis.2011.51

Gioia, D. A., Corley, K. G., & Hamilton, A. L. (2012). Seeking Qualitative Rigor in Inductive Research. Organizational Research Methods, 16(1), 15–31. https://doi.org/10.1177/1094428112452151

---

> ### Author Rebuttal · Authors · 2026-03-31
>
> **Strengths**
>
> We thank the reviewer for their thorough and constructive evaluation. We are glad the reviewer found our work to address a highly relevant issue and appreciated the quantitative rigor of our analysis, including the robustness checks with different priors. We are also pleased that the reviewer found our supplementary materials (such as the variance composition visualizations, posterior distribution intervals, source code, model identification approach, interview guide, and interviewee overview) to strengthen the transparency and reproducibility of our work. We appreciate the reviewer's recognition that the paper is well-structured and accessible, and that the empirical demonstration of the neglect of first-party social impact evaluations, alongside the qualitative insights, provides a meaningful starting point for connecting AI evaluation practices to effective legislation.
>
>
> **Key Questions**
>
> - **_Q1_** (see W4)
> - **_Q2_** (see W1)
> - **_Q3_** (see W1)
>
>
> **Limitations**
>
> - **_L1_** We appreciate the reviewer raising the concern about potential selection bias. To clarify, all for-profit interviewees work at different companies. We will state this explicitly in the revised paper and add a column to Table 12 (App. 9.13.2) indicating years of experience for each interviewee, providing additional context to assess the expertise represented in our sample. We acknowledge that the sample is small (N=10) and not representative of smaller organizations or global majority countries, as noted in Sec. 8. We view the interviews as providing contextualizing evidence for the quantitative patterns rather than standalone generalizable findings.
>
>
> **Weaknesses**
>
> - **_W1 and W2_** We agree with the reviewer that the interviews function as illustrative anecdotes rather than systematic qualitative evidence, and will revise the manuscript to reflect this consistently, replacing causal framing ("explains why") with contextualizing framing ("is consistent with"). All central claims rest on the quantitative scoring and regression. In addition, we will add a paragraph in Appendix M.1 describing the thematic coding to clarify how quotes were selected and to what extent they represent convergent themes but we do agree the sample is limited, which we acknowledged in our limitations. We thank the reviewer for the references to Wolfswinkel et al. and Gioia et al. and will draw on these to strengthen the presentation of the qualitative component.
>
> - **_W3_** We will streamline the appendix structure in the revision to ensure heading-content alignment and improve navigability across the supplementary material.
>
> - **_W4_** We will add a row showing per-category averages across providers in the revision, which we agree connects more directly to the paper's central research question about where social impact reporting gaps lie.
>
> - **_W5_** We appreciate this push. We outline three directional takeaways in Sec. 7: (1) safe-harbor provisions to reduce legal and reputational barriers to good-faith disclosure, (2) standardized reporting templates aligned with governance efforts to improve comparability and reduce documentation burden, and (3) publicly supported infrastructure (e.g., compute- or energy-use reporting APIs) to lower reporting costs across developers of all sizes. These are deliberately framed as directional rather than prescriptive, since effective legislative implementation will vary across jurisdictions. However, we will note in the camera-ready version how existing efforts such as SB-53 or the EU AI Act in the EU could be extended to address some of the concerns raised in the paper. We will also be more explicit about the role that policymakers can play in collectively advancing transparency infrastructure. We’re open to including other (even bolder! :)) calls to action here and would love any suggestions if the reviewer had specific action items in mind!
>
> - **_W6_** We agree and will revise the manuscript to consistently use contextualizing rather than causal framing for claims supported by interview evidence. For example, we will replace language like "gaps are attributed to" with "interviewees described gaps as consistent with," ensuring the text accurately reflects the illustrative role of the qualitative data.
>
> - **_W7_** We thank the reviewer for the insightful references. In the camera-ready, we will incorporate these to better situate our findings within the broader responsible AI and organizational literature, including work on financial and market motivations influencing first-party disclosure practices. Thank you for your suggestions here!
>
>
> **We are grateful for the reviewer's encouraging final remarks and constructive suggestions, which will substantially strengthen the paper. We look forward to incorporating these changes in the revision. Thank you so much for your time and work on this – and we also wish you all the best for your future research endeavors!**

---

> > ### Author Rebuttal · Reviewer_9bdA · 2026-04-04
> >
> > Thank you for the clarifications. I will stick to my "accept" rating.

---

### Official Review · Reviewer_1tGL · 2026-03-13

**Soundness:** 3
**Presentation:** 3
**Significance:** 3
**Originality:** 3
**Overall Recommendation:** 5
**Confidence:** 4

**Summary:**

This social analysis paper reports on an investigation into evaluations of AI systems, both from first and third parties. They look at reports, conduct interviews and score according to a fixed framework. The key findings are that first party reports lack among many reporting dimensions, third party evals often give better details, even though only first party developers have full insight into the processes. They also find reporting detail has dwindled in recent years compared to earlier releases.

**Compliance With Llm Reviewing Policy:**

Affirmed.

**Final Justification:**

I maintain that this is a good paper, recommend accept.

**Key Questions For Authors:**

1. The timeline breakdown in Figure 3 is given for two companies in Figure 4, but would it be possible to include a figure similar to figure 3 but broken down by developer? That would help to understand the dynamics better
2. Later papers often build on previous papers/development and re-use the dataset (sometimes in its entirety) and refer to the previous paper for dataset construction. In such cases, does the reporting have to be repeated, or is the reporting then scored against the ground reference? E.g. model 1.0 declares no manual labour used, then model 2.0 says "we use exactly the data from model 1.0". Does the model 2.0 paper get a low score or does it "link" to the model 1.0 reporting?

**Limitations:**

The paper does a pretty good job of outlining limitations, but a few cases could be clearer, for example they mention having low interview rates outside of north america and europe. Looking at the appendix it looks like there were none from outside US/Canada/UK/France. This could be stated more explicitly.

**Strengths And Weaknesses:**

Strengths:
- S1: A thorough analysis of societal impact reports covering industry, non-profits, academia, government AI system releases.
- S2: The approach is principled with a clear search framework and detailed description for reproducibility. The scoring framework consists of a previously published one, with multiple scorers and inter-agreement reported
- S3: The paper is well written, clear and easy to follow

Weaknesses:
- W1: The analysis is limited to a specific small group of model developers.
- W2: The paper would benefit from a description of the dimensions that are being scored against in the main text. For example, how is "Performance disparity" defined? Looking at the supplementary/appendix breaks the flow, as it is necessary to understand the paper.
- W3: The manual search on the company website is at risk of missing reports or published details. A more comprehensive and automated search might find things that missed
- W4: Some items are scored as zero when there are zero instances of it. However, it is not clear that this is missing and should be included or if a simple mention e.g. "No moderation labour used", would be sufficient to get good marks if it truly is not applicable in that case. This is slightly confusing and some clarification is needed.

---

> ### Author Rebuttal · Authors · 2026-03-31
>
> **Strengths**
>
> We thank the reviewer for their careful reading and positive assessment. We are glad the reviewer recognizes the thoroughness of our analysis across industry, non-profit, academic, and government AI system releases (S1), the principled and reproducible nature of our search and scoring framework (S2), and the clarity of the writing (S3).
>
>
> **Key Questions**
>
> - **_Q1_** We appreciate this suggestion. We will include per-provider temporal breakdowns for all 16 stratified providers in the camera-ready appendix, extending the analysis currently shown for Google and Meta in Figure 4 to the full stratified sample.
>
> - **_Q2_** Thank you for raising this important point about cross-referencing between model releases. When a report explicitly references prior documentation (e.g., "we use exactly the data from model 1.0"), we treat the linked information as part of the current document and score on the union of reported information across both sources. In the example given, model 2.0 would receive a score based on the combined manual labor information from both reports. We will clarify this scoring procedure in the revision.
>
>
> **Limitations**
>
> - **_L1_** We appreciate the reviewer's push for greater transparency here. The reviewer is correct that all interviews were conducted with participants based in the US, Canada, UK, and France. While our recruitment efforts did extend beyond these regions, some prospective participants were reluctant to engage, possibly due to concerns around discussing commercially sensitive information. We will state the geographic distribution of interviewees more explicitly in the main body and incorporate this context into the limitations section in the revision.
>
>
> **Weaknesses**
>
> - **_W1_** We recognize that our sampling favors more prominent providers. However, the foundation model ecosystem is itself highly concentrated, with a small number of providers accounting for the majority of deployed usage [1]. Our dataset of 50+ providers covers these alongside smaller and regional developers across sectors and geographies (see Appendix B). We will make sampling limitations more explicit in the revision by clarifying that our strategy of triangulating across public sources, including leaderboards, inherently favors models with greater visibility.
>
> - **_W2_** We agree that the social impact dimensions should be described in the main text. While they currently sit in the appendix due to space constraints, we will include concise descriptions along with examples in the camera-ready version to improve readability and flow.
>
> - **_W3_** We acknowledge that manual search is less scalable than fully automated alternatives and may miss relevant documents. To mitigate this, we triangulated across multiple sources and iteratively expanded searches when referenced materials were identified. We will explicitly note this limitation in the revision. That said, we would point out that if first-party social impact evaluation reports cannot be found through targeted manual search of a provider's website, this itself constitutes a transparency concern, as the information is functionally inaccessible to stakeholders. We discuss potential mitigations such as shared reporting infrastructure in Section 6.
>
> - **_W4_** While we have not encountered deliberate disclosures of the form "no moderation labor used" during annotation, such cases would still receive a score of 0 under our current scheme. We will revise the scoring description to clarify how absence-by-design versus absence-by-omission is handled.
>
> [1] https://arxiv.org/pdf/2601.10088
>
> **We want to thank the reviewer for their time and effort – your comments substantially improved our paper, thank you so much!**

---

> > ### Author Rebuttal · Reviewer_1tGL · 2026-04-04
> >
> > I thank the authors for their responses to my questions and concerns. I have no further questions and continue to recommend acceptance.

---

### Review · Ethics_Reviewer_7BP2 · 2026-03-28

**Recommendation:** Remediation action needed

**Ethics Issue:**

It is unclear whether this project was reviewed by an internal review board or equivalent. Also, the details of the informed consent are unclear.

**Remediation Action:**

The ethical issues here can be easily addressed with the addition of a few details in the methods section and appendix.

If the project was reviewed by an internal review board or equivalent, a statement to that effect should be included in the methods section. In some situations, IRB is not required by law or policy. (For example, in the U.S., researchers at universities or government agencies must receive IRB approve. But, researchers working for companies unfortunately do not.) If IRB or an equivalent review was not conducted, I recommend including details of how the researchers mitigated risk to the participants. Minimizing potential risks to participants is the primary purpose of IRB review. If that review was not required by law or policy, ethical researchers still consider any potential risk, whether those risks are justified by the benefits, and how to mitigate those risks. I discussion of these considerations and the choices the researchers made should be included.

The appendix contains a statement given to the participants. I would recommend adding a couple of sentences about the informed consent process - when did researchers get consent? how was that done (verbally, in writing, requiring a verbal affirmation from the participant, requiring a signature., etc.)? how was consent recorded?

---

### Decision · Program_Chairs · 2026-04-30

**Decision:**

Accept (regular)

**Comment:**

The reviews agree that the paper tackles an important and timely problem; analysis of social impact evaluation of AI from first and third party evaluators. The paper makes a novel, large-scale empirical contribution using a transparent and generally well-executed methodology (including Bayesian analysis and a reproducible pipeline). Its core empirical finding is that that first-party reporting is consistently less detailed than third-party evaluation and may be declining over time, which is a valuable contribution and potentially impactful for policy and governance discussions.

While the reviewers agreed that this is a strong and timely paper with meaningful insights into transparency and evaluation practices in AI, they also raised concerns around construct validity, interpretation and reframing or limiting claims concerning comparison of first and third party evaluations. Despite this and other surface level limitations outlined by reviewers, the methodology is broadly sound, the findings are robust at a high-level, and the weaknesses are either acknowledged or addressable without undermining the main contribution.